# Formation, evolution and drainage of short-lived glacial lakes in permafrost environments of the northern Teskey Range, Central Asia

Mirlan Daiyrov[1,2], Chiyuki Narama[3]

[1] Central-Asian Institute for Applied Geosciences (CAIAG), Bishkek, Kyrgyz Republic
[2] Niigata University, Graduate School of Science and Technology, Niigata, Japan
[3] Niigata University, Program of Field Research in the Environmental Sciences, Niigata, Japan

*Correspondence to*: Mirlan Daiyrov (mirlan085@gmail.com), Chiyuki Narama (narama@env.sc.niigata-u.ac.jp)

**Abstract.** In the Teskey Range of the Tien Shan (Kyrgyz Republic), four outburst flood disasters from short-lived glacial lakes in 2006, 2008, 2013, and 2014 caused severe damages in the downstream part. Short-lived glacial lakes in the Teskey Range grow rapidly and drain within a few months, due to closure and opening of an outlet ice-tunnel in an ice-cored moraine complex at the glacier front. In addition to these factors, summer meltwater from the glacier can cause rapid growth. Outburst floods of this lake type are a major hazard in this region and differ from the moraine-dam failures common to the eastern Himalaya. To clarify how short-lived glacial lakes store and drain water over short periods, we use results from a field survey and satellite data to analyse the water level, area, volume, and discharge of Korumdu lake (2017–2019) as well as satellite data to monitor the appearance of 160 other short-lived lakes (2013–2018). Except in 2016, Korumdu lake appeared and drained within about one month during all the summers. Water level data recorded by a data logger and time-lapse camera images show that the lake appeared and expanded suddenly from July to August in 2017–2019. The timing of lake appearance indicates that the lake formed when an outlet ice-tunnel (subsurface channel) drainage was blocked by depositions of ice-debris mixture due to ice melting, not by freezing of stored water. For 2017, we used unmanned aerial vehicle (UAV)-derived digital surface models (DSMs) and water levels, finding that the lake's volume reached 234,000 $m^3$ within 29 days, and then the water discharged for 17 days at a maximum rate of 0.66 $m^3$/s. This discharge rate is more than 20 times smaller than those found earlier (2006–2014) for four short-lived lakes in the region. We argue that this large variation in discharge rates is due to variation in the dimensions of the outlet ice-tunnels.  For the 160 other short-lived glacial lakes, we found that 117 formed during the ice-melt period from July to September. This timing and our findings for Korumdu lake show that these 117 lakes likely formed primarily because deposition of an ice-debris mixture blocked the outlet tunnel, though increased glacial melt would also have contributed. In the Teskey Range, the appearance of short-lived glacial lakes on the moraine complexes at glacier fronts is inevitable in summer when the melting rate is high. Similar behaviour of short-lived lakes may occur in other mountain regions of Central Asia, such as the Tien Shan and Pamir mountains, wherever ice-cored moraine complexes exist within mountain permafrost zone. Moreover, increasing temperatures may increase both tunnel size and lake-basin size (lake volume) leading to increased hazard potential from such lakes in the future.

# 1 Introduction

Compared to the large proglacial lakes in the eastern Himalayas (Ageta et al., 2000; Komori et al., 2004; Bajracharya et al., 2007; Nagai et al., 2017), glaciers in the northern Tien Shan (Central Asia) tend to have small glacial lakes near their termini (Janský et al., 2008; Narama et al., 2010a; 2015). Drainage events from these lakes often produce hazardous debris flows and floods. For example, debris flows in 2006, 2008, 2013, 2014, and 2019 in the Teskey Range of the northern Tien Shan caused severe damage (including casualties) and destroyed bridges, roads, houses, and crops (Narama et al., 2010a, 2018; Daiyrov et al., 2020).

Some of these small lakes are called short-lived as they grow rapidly and drain within a few months (Narama et al., 2010a, 2018; Daiyrov et al., 2018). Such short-lived lakes appear in depressions of ice-cored moraine complexes at glacier fronts. The lakes drain through an outlet ice-tunnel (subsurface channel) within the moraine complex (Popov, 1987; Narama et al., 2010a; 2018). Some authors call them nonstationary lakes (Erokhin et al., 2017), though this term also includes lakes with a long lifetime. Most short-lived glacial lakes fill periodically and within one year, though some may develop for 2–3 years before draining. The latter type can be more dangerous; for example, in the Tajik Pamir, drainage from a short-lived glacial lake that formed within 2 years resulted in 25 casualties (Mergili et al., 2012). In northern Tien Shan, short-lived glacial lakes can be a severe hazard for local residents because they appear suddenly yet can cause large debris flows. Outburst mechanism and damage potential of short-lived glacial lakes in the northern Tien Shan differ from those that are caused by moraine-dam failure in the Himalaya and Andes (Costa and Schuster, 1988; Richardson and Reynolds, 2000; Shreshta 2010; Emmer and Cochachin, 2013; Neupane et al.; 2019). A mass-movement such as an ice avalanche or landslide is often the main cause of dam failures of the glacial lakes in the Himalayas and Andes (Emmer and Cochachin, 2013; Neupane et al.; 2019).

Short-lived glacial lakes that are dammed by partially frozen moraine material (ice-cored moraine complex) drain through a subsurface outlet ice-tunnel. These lakes can expand rapidly when the outlet ice-tunnel is blocked due to either freezing of stored water or depositions of ice and debris (Narama et al., 2010a, 2018). Drainage then occurs when the outlet ice-tunnel opens during summer. Some of these short-lived glacial lakes reappear every year (Daiyrov et al., 2018), which is behavior they share with supraglacial lakes. Several studies reported that formation and drainage of supraglacial lakes are related to connectivity of englacial conduits on a debris-covered glacier (Benn et al., 2000, 2017; Miles et al., 2016; Watson et al., 2016; Narama et al., 2017). However, the variations of short-lived glacial lakes in northern Tien Shan arise from their ice-tunnel opening and closing as well as the increase in glacial melt during summer (Daiyrov et al., 2020).

Short-lived glacial lakes in northern Tien Shan appear at depressions that can be created when a glacier retreats, when an ice-cored moraine complex subsides (Narama et al., 2010a, 2018; Daiyrov et al., 2018). Narama et al. (2018) showed that such short-lived glacial lakes typically form where the following three conditions exist: 1) an ice-cored moraine complex (debris landform containing ice), 2) a depression with a water supply to an ice-cored moraine complex or glacier terminus, and 3) the absence of a visible surface outflow channel from the depression. The last condition indicates that the moraine complex has an outlet ice-tunnel to drain lake water.

The number and area of glacial lakes in the Tien Shan has recently increased, a trend that is linked to climatic warming and glacier shrinkage (Bolch et al., 2011; Wang et al., 2013; Kapitsa et al., 2017). In addition, Daiyrov et al. (2018) showed that the large variability in the number and distribution of glacial lake types in the Issyk-Kul Basin is not only related to the local climate conditions, but also to the three conditions in the glacier forefield described above.

Many short-lived glacial lakes have been observed in the northern Tien Shan in recent years (Daiyrov et al., 2018). They can

change in area and volume over a short period of time, making their drainage features and discharge rates unpredictable (Erokhin et al., 2017), but not all short-lived glacial lakes cause large-scale floods. A short-lived lake's fate depends on whether the dam contains ice (Mergili et al., 2013), and if so, how the outlet ice-tunnel closes and opens. However, the mechanisms of lake formation and drainage remain unclear. Hazards from an abruptly changing glacial-lake discharge can intensify dramatically and unexpectedly within weeks or even days (Huggel, 2004).

In this study, we investigate formation and drainage mechanisms of Korumdu lake in the Teskey Range, Tien Shan (Kyrgyz Republic) and the reason for different discharge (rates) from short-lived lakes based on field survey and satellite data analysis. To clarify how the other short-lived lakes in the Teskey Range form and drain water, we investigate their timing of appearance during summer months between 2013 and 2018 using Landsat-7/8, Sentinel-2, and PlanetScope satellite images. Finally, we discuss the causes of outlet ice-tunnel closure for Korumdu lake and other lakes of the same type in the study area. We also

examine the relationship between outlet tunnel size and lake drainage rate.

## 2 Study area

The study area is situated in the northern part of the Teskey Range, south of Lake Issyk-Kul (Fig. 1). The glacier distribution (3700–4200 m a.s.l) in the western part of the rangeis lower than the distribution (3800–4500 m a.s.l) in the eastern part due to the annual precipitation being higher in the eastern part than in the western part. For example, during 1998–2007, the average

annual precipitation at the Kara-Kujur station (2800 m a.s.l) of the western part was 255 mm, whereas at the Tien Shan station (3614 m a.s.l) of the central part it was 378 mm, and at the Chong-Ashu station (2788 m a.s.l) of the eastern part it was 550 mm (Podrezov and Ryskal, 2019; Fig. 1). Mean annual air temperature was 0.1°C (1961–1988) for Kara-Kujur, –6.28°C (1995–2011; Kuzmichenok, 2013) for Tien Shan, and 0.27°C (1995–2005) for Chong-Ashu. The western part of the range showed less glacier shrinkage than that in the eastern part (Aizen et al., 2006; Narama et al., 2006; Kutuzov and Shahgedanova,

2009).

In this area, the four large drainage events of Kashkasuu (2006), western Zyndan (2008), Jeruy (2013), and Karateke (2014) recently occurred from short-lived glacial lakes that formed on ice-cored moraine complexes (debris landforms including ice) (Narama et al., 2010a; 2018). The ice-cored moraine complexes here lie at 3200–4000 m a.s.l (Daiyrov et al., 2018) at the glacier fronts that developed during the Little Ice Age (Dikih, 1982; Shatravin, 2007; Narama et al., 2010b) due to ice and

95 debris stagnating during glacier shrinkage after several glacier advances (Iwata et al., 2005).

We ran a field survey at Korumdu lake at 3806 m a.s.l (Figs. 1, 2). The Korumdu catchment is source to the largest tributary in the Tong River Basin, and according to a Sentinel-2 satellite image of 2019, Korumdu glacier occupies an area of 2.35 km$^2$.

At the front of the Korumdu glacier lies Korumdu glacial lake (Fig. 2). The dam of this lake is an ice-cored moraine complex. The lake developed in a depression that formed during the retreat of the glacier and retains direct contact with the glacier. We selected this lake for field surveys because (i) the lake is a short-lived type that appears every year, (ii) it is easy to access, and (iii) it is located in the Tong region where four large outburst floods occurred in the past. According to data in Narama et al. (2018), outburst drainage from Korumdu lake is the flood-wave type in the downstream region because the water stream flows on a gentle slope which the flow hardly acquire any debris by erosion. In addition, we investigated the timing of appearance for 160 short-lived lakes in the northern Teskey Range during 2013–2018 (Fig. 1) using Landsat-7/8, Sentinel-2, and PlanetScope satellite images (Supplementary Table 1).

## 3 Methods

### 3.1 Field observations at Korumdu lake

The field survey at Korumdu lake (Figs. 1, 2) was run during the summers of 2015–2019. We installed water level and water temperature data loggers (Hobo U20) at lake bottom and ground surface on the moraine to collect measurements once per hour since 21 August 2015. Water level logger measurements (water pressure data) at lake bottom were converted to the water level (meter) using atmospheric pressure data at the adjacent ground surface on the moraine. A time-lapse camera (Brinto) was installed as well and took one oblique image of the area per day.

In addition, we obtained aerial images of the lake basin acquired by Phantom-4 (DJI) and JABO H601G (Medix) unmanned aerial vehicles (UAVs) with a mounted camera (Ricoh GR) on 21 August 2015, 12 Aug 2016, 6 August 2017, 20 July 2018, and 4 August 2019. High-resolution orthoimages and digital surface models (DSMs; resolution of 0.2 m) were made using the Pix4D mapper (Pix4D SA) of Structure from Motion (SfM) software and ground control points (GCPs). We collected the GCPs around the lake using a Trimble GeoExplore 6000 Global Navigation Satellite System (GNSS). The absolute positions of GCPs were corrected during post-processing using data from the Kyrgyz GNSS reference station and had an accuracy of 30–40 cm. Surface elevation changes of the moraine complex surrounding the lake were computed in ArcGIS 10.5 by comparing UAV-derived DSMs from 2015 and 2016.

The daily volume and discharge of the lake during the summers of 2017–2019 were calculated using the daily water level data and the 2017–2019 UAV-derived DSMs combined with the 2016 UAV-derived DSM (without water) in ArcGIS 10.5. For the water volume of the lake's bottom layer, we used the 2016 DSM because the 2017–2019 DSMs had water at the lake bottom. In addition, we investigated whether satellite remote sensing data could replace in-situ water level logger data to calculate lake water levels using the combined DSMs. We found that the water level logger measurements agreed with the derived water levels based on UAV-derived DSMs combined with satellite imagery. For example, we confirmed the position of the water level by comparing a UAV orthorectify image or satellite data with 1-m contour lines from the combined UAV-derived DSMs. Finally, we obtained the water level and lake area from satellite data. Using this method, we reconstructed the water level data between August 4 and 31 2019 based on 10 satellite images from PlanetScope, Landsat-8/Operational Land Imager (OLI), and Sentinel-2 because we do not have water level data after our last field survey on 4 August 2019. We also investigated the

changes in lake area during 2017–2019 by comparing lake polygons that had been digitized manually using PlanetScope images.

Finally, we examined the meteorological and thermal conditions using air and ground temperature data loggers (TR-52i; T&D; accuracy ± 0.3°C) to log data at 1-hour intervals around the lake (Fig. 2). Mean annual air temperature (MAAT) between 2015 and 2017 and mean annual ground surface temperature (MAGST) during 2015–2019 were calculated.

## 3.2 Timing of appearance of short-lived lakes using satellite data

Short-lived glacial lakes in the northern Teskey Range were identified in ArcGIS 10.5 using satellite images (Landsat-7/Enhanced Thematic Mapper Plus (ETM+, SLC-off), Landsat-8/OLI), acquired during 2013–2018 (Supplementary Table 1). We used the definition by Daiyrov et al. (2018) for short-lived lakes, which is based on seasonal changes in lake area over the summer months of each year. Specifically, a short-lived lake is a temporary lake, lasting just one or two years, that suddenly appears or increases substantially in area, then disappears or shrinks within the same year. We counted the number of lakes that appeared from June to September each year. In addition, the number of lakes was tracked in each given year to examine how it changed from one year to the next. Polygon shapefiles of lakes were digitized manually from the images using ArcGIS 10.5. We also investigated the area changes of short-lived glacial lakes during summer months in a given year.

## 4 Results

### 4.1 Areal variability of Korumdu lake

ALOS/AVNIR-2 data taken on 17 September 2007 indicated that most of the lake basin had been covered by Korumdu glacier. Thus, the lake basin developed in a depression that formed during the retreat of the glacier in recent years. The UAV ortho-images of 2019 indicated a basin length of 360 m, a width of 110 m, with a total area of 0.062 km$^2$. The lake basin volume increased from 264,000 to 330,000 m$^3$ from 2017 to 2019 (Fig. 2) due to retreat of the glacier terminus. In the field, we observed ice-exposed ridge and debris sliding on the basin's slope, indicating that the ice was melting around the shore, thus increasing the basin's width.

The lake had no discernable surface drainage channel, but we found an outlet point where meltwater from the lake emerges from a subsurface ice-tunnel within the ice-cored moraine complex that connects to the lake (Fig. 2). The length of the outlet ice-tunnel is 60 m from the entrance of the lake basin. During the fieldwork, we observed melt water draining at the outlet point on 30 July 2015, 6 August 2017, and 4 August 2019, but not on 12 August 2016 and 20 July 2018.

Concerning lake size changes, in 2015, the lake appeared sometime before 30 July, then shrank significantly by 21 August (Fig. 3). In 2016, according to the water level data and on-site time-lapse camera images, the lake area did not form. For 2017–2019, we had more images of the area could be acquired and thus a more detailed evolution of changes in lake size is shown with a sequence of PlanetScope satellite images in Fig. 4. The images show that the lake appears suddenly at the end of July

to the beginning of August and then shrinks and vanishes by the end of August (Fig. 4). Although the timing of lake expansion differs slightly over the years 2017–2019, the lake always appears in summer. The time-lapse on-site images show the same behavior from a different view (Fig. 5). These images also indicate that the lake began to expand from mid-July and reached its maximum size at some time between late July and early August. Thus, the satellite data demonstrate that the lake is a short-lived glacial lake.

## 4.2 Changes in water level, area, volume, and discharge of Korumdu lake

Changes in water level, area, volume and discharge of Korumdu lake were studied in detail during the three summers of 2017–2019. For 2017, Fig. 6a shows the water level increasing from 6 July, reaching a maximum on 3 August, and then the lake is empty on 19 August. Within 29 days, the water level increases 13 m, the area reaches 0.36 km$^2$ (Fig. 6b), and the volume reaches 234,000 m$^3$ (Fig. 6c). The resulting rate of lake volume increase is 8,070 m$^3$ per day. During the emptying of the lake, 234,000 m$^3$ of water drain in 17 days, with half of the volume draining from 3 to 7 August 2017 (Fig. 6c), resulting in a maximum net outflow discharge of 0.66 m$^3$/s (Fig. 6d). Although the water level increases intermittently before 3 August, the net outflow is relatively low. The lake water temperature averages about 1°C (Fig. 6a). The water temperature fluctuates more when the lake is shallower because the heating of shallower water by solar irradiance is stronger than cooling from inflowing ice meltwater.

In 2018, the water level peaks three times, though reaches only about half that of 2017 (Fig. 6a). The first peak, on 25 July, occurs with a lake depth of 3.5 m and a volume of 21,000 m$^3$ (Fig. 6a, c). The second, and maximum peak, occurs on 11 August with a lake depth of 6 m and a volume of 53,000 m$^3$. The third peak occurs on 17 August with a lake depth of 5 m and a volume of 39,000 m$^3$. The maximum net discharge occurs after the second peak, reaching 0.32 m$^3$/s (Fig. 6d). Similar to 2017, the net inflow rate also clearly varies over time in 2018. The three peaks in water level, area, and volume indicate that closure of the ice-tunnel occurred several times during a one-month period.

In 2019, the lake water level rises and falls before 22 July, when it rises sharply (Fig. 6a). Then, the water level shows a maximum around 30–31 July, reaching a lake depth of 5 m and a volume of 53,000 m$^3$. The 2019 maximum level occurs on 11 August, with a lake depth of 6.5 m and a corresponding volume of 74,000 m$^3$ (Fig. 6a, c). The maximum discharge occurs right after the second peak, reaching 0.24 m$^3$/s (Fig. 6d).

Over all three years, the highest water level is in 2017 (Fig. 6a). In general, each year differs in the timing of lake-level increase, number of peaks, and maximum water volume. All three years have relatively small net discharge rates (maxima of 0.66, 0.32, and 0.24 m$^3$/s in 2017, 2018, and 2019), which is consistent with the absence of reported flooding.

During each of these years, the lake level rose and fell several times, indicating repeated storage-drainage cycles. In the field, we observed sudden small increases in water level in 2016 and 2017, with the lake level increasing tens of centimeters within 3 h (Fig. 7). These results indicate that water level fluctuations occurred frequently due to closing and opening of the outlet ice-tunnel.

During the fieldwork, we observed lake water draining out an outlet point in 2015, 2017, and 2019, but not in 2016 and 2018. In years that drainage occurred, the lake elevation exceeded that of the outlet ice-tunnel entrance. For example, the water levels were at 3,810 m a.s.l on 21 August 2015, 3,816 m a.s.l on 6 August 2017, and 3,810 m a.s.l on 4 August 2019, all above that of the outlet ice-tunnel entrance at approximately 3,807.5 m a.s.l. In contrast, in 2016 and 2018, lake water levels were lower than 3,807.5 m a.s.l, the outlet tunnel elevation (Fig. 8a, c). Therefore, the field-survey results indicate that the key factor

determing where drainage occurs is the relative elevations of the lake surface and outlet ice-tunnel entrance.

### 4.3 Surface changes on ice-cored moraine complex around Korumdu lake

Between 2015 and 2016, debris sliding and horizontal backwasting around the lake exposed an ice ridge of up to 7 m height (Fig. 9). The backwasting indicates that melting of debris-covered ice occurred, which is supported by comparing the UAV-

derived DSMs from both years (Fig. 9c). For instance, along the cross-sectional profile a–a' in Fig. 9b, the surface elevation decreased by about 5 m (Fig. 9c). These results are consistent with closure in the outlet ice-tunnel during ice-melt period due to surface motion and ice-debris deposition. During our fieldwork in 2016, we observed water flow at the entrance of an ice-tunnel. After two or three hours, the lake level increased (Fig. 7), consistent with the cause being closure of the ice-tunnel. In the northern part of the Teskey Range, the discontinuous mountain permafrost zone lies above 3,100–3,200 m a.s.l (Daiyrov

et al., 2018). Around Korumdu lake (3,806 m a.s.l), the mean annual air temperature (MAAT) during 2015–2017 was –4.8°C and the mean annual ground surface temperature (MAGST) during 2015–2019 was –2.9°C. Thus, the buried ice of the ice-cored moraine complex at Korumdu lake is maintained under a permafrost environment. Melting of buried ice causes surface changes including expansion of the lake basin, expansion and deposition (closure) in the outlet ice-tunnel.

**4.4 Comparison to other short-lived glacial lakes of the Teskey Range**

To determine when other short-lived glacial lakes in the northern Teskey Range formed, we used satellite images of 2013–2018 (cf. Supplementary Table 1). Based on the satellite imagery, a total of 160 short-lived glacial lakes could be identified. A classification of these lakes by month of appearance is shown in Fig. 10. Most lakes appeared in June (43 lakes) during the snow-melt period, and in July (90 lakes) during the ice-melt period. The total numbers and the proportions of the numbers for

these two periods varied during the 6 years. The number of lakes vary greatly by year and by appearance date, indicating that the formation of these short-lived glacial lakes can not be explained solely by an increase of melt water during summer. Such variability has been argued to be related to geomorphological conditions such as drainage through ice tunnel inside of ice-cored moraine complex (Daiyrov et al., 2018).

Concerning reappearances, 81 lakes appeared only once during six years. Of the remaining, 19 lakes appeared twice, 7 lakes

appeared three times, 2 lakes appeared four times, and 2 lakes appeared all 6 years. These results are consistent with tunnel closure being the main cause of formation. Short-lived glacial lakes that reappear during many years likely have an environment that either favors tunnel closure and hence lake formation or an increase in meltwater from glacier during summer (Daiyrov et al., 2020).

## 5 Discussion

### 5.1 Causes of outlet ice-tunnel closure in the northern Teskey Range

We first consider four previously studied short-lived lakes in the area. The Kashkasuu (2006), western Zyndan (2008), Jeruy (2013), and Karateke (2014) lakes appeared in May–June and expanded in area until June–July, then all had relatively large drainage events leading to serious damages (Narama et al., 2010a, 2018). This timing of lake appearance suggests an ice-tunnel closure that is caused by the freezing of stored water during winter or deposition of ice-debris mixture as sketched in Fig. 11a (Popov, 1987; Narama et al., 2010a, 2018). We call this the deposition–freezing type of ice-tunnel closure.

In contrast, Korumdu lake appeared during July–August (except in 2016) and produced relatively little drainage during emptying. This different appearance time might reflect a different formation process. As we observed subsidence and downwasting changing the lake basin (Fig. 9), the blockages of the outlet ice-tunnel at its entrance or interior were likely caused by deposition of ice-debris mixture from thermal erosion. This type of blockage (deposition–collapse type) is sketched in Fig. 11b. The water level fluctuations support this mechanism. The fluctuations of lake water level and discharge spikes reveal changes in the ice-tunnel morphology (Fig. 6d). A sudden blockage of an outlet ice-tunnel can cause a rapid increase in water level within a few weeks. Also, the water level increase was sporadic, indicating that the outlet ice-tunnel was not completely closed, the blockage was temporary, and the size of the ice-tunnel is small. As a result, lake drainages can occur any time in summer, depending on how the outlet ice-tunnel responds to changes in water pressure or deposition of ice-debris mixture through melting processes.

In the northern Teskey Range, the Toguz-Bulak glacial lake appeared in June and disappeared in September every year from 2010 through 2019 due to the inflow of glacier meltwater (Daiyrov et al., 2020). This lake has a surface drainage channel from the lake, but its incoming glacial runoff controls its behaviour, such as its area. Thus, as for short-lived glacial lakes with surface drainage channels like Toguz-Bulak, the evolution of the glacier mass balance during summer (amount of snow and ice melt flowing into the lake) also plays an important role for the formation and evolution of short-lived glacial lakes having a subsurface outlet ice-tunnel.

In 2017, there were two trends in water volume of Korumdu lake (Fig. 6b). The first period (5–25 July) involved sporadic fluctuations superimposed on an increase in water volume, indicating incomplete closure of the ice-tunnel. Then, in the second period of 26 July to 3 August, the volume continuously and rapidly increases, indicating complete closure of the ice-tunnel. Hence, we argue that the main factor of these rapid lake-area changes is tunnel closure. The lake area reached its maximum in 2017. This indicates that the tunnel closure was longer in 2017 than in 2018 or 2019. Longer periods of tunnel closure are associated with the formation of larger short-lived glacial lakes (Narama et al., 2018). Thus, the period of closure is likely determined by the morphology of the ice-tunnel and deposition condition of tunnel-closure point (e.g., when melting can open the blocked region).

Many of the other short-lived glacial lakes in the northern Teskey Range, observed via satellite imagery, are likely to belong to the deposition–collapse type as well. However, some likely have a larger influence from the water balance between drainage

and storage, related to increasing glacial meltwater and tunnel size. Consider Jeruy glacial lake between 2014 and 2016 (Fig. 12). Ice melting caused distinct changes and rapid deposition within the outlet ice-tunnel, which likely led to tunnel closure.

Thus, morphology and surface characteristics of an ice-cored moraine complex within the mountain permafrost zone are prone to frequent changes, and the deposition–collapse type is likely the main type for the short-lived glacial lakes in the northern Teskey Range. If the deposition–collapse processes occur in summer when the melting rate is high, the formation of a short-lived glacial lake is highly likely.

## 5.2 Relationship between outlet tunnel size and lake drainage

Of the 160 short-lived lakes we identified in 2013–2018, only Jeruy lake (in 2013) and Karateke lake (in 2014) showed considerable drainage. The estimated maximum discharges from Jeruy (182,000 $m^3$) and Karateke (123,000 $m^3$) lakes were 14.9 and 11.5 $m^3$/s, respectively (Narama et al., 2018). These lakes had relatively large outlet tunnels, with one at Jeruy, as well as one at Karateke, having a cross-section of about 8 $m^2$ (Fig. 12a,b). Earlier, in 2008, the w-Zyndan lake (437,000 $m^3$)

emptied at a higher discharge rate of 27 $m^3$/s (Narama et al., 2010a). Most of the water in these three cases drained over a period of several hours. In contrast, Korumdu lake did not show such high drainage rates during 2014–2019, draining at a maximum rate of 0.66 $m^3$/s in 2017, taking 17 days to drain 234,000 $m^3$, and its tunnel cross-section was much smaller than those at Jeruy or Karateke lakes.

In addition, Korumdu lake exhibited sudden fluctuations of water level over several hours, which we argue was related to

280 closure of the small outlet ice-tunnel caused by deposition of and blockage by debris. The relatively small tunnel size of this lake resulted in slower lake discharge even when lake volume reached its maximum (330,000 $m^3$). During 2017–2019, the lake size was largest in 2017, yet discharge rates were in the same order of magnitude every year. These results show that, at least for Korumdu lake, the dimensions of the outlet ice-tunnel were the dominant factor controlling lake discharge rates.

However, tunnel dimensions could increase in the future due to thermal erosion and flowing water, allowing greater discharge

rates. Meltwater and increasing temperature can accelerate thermokarst processes enlarging the outlet ice-tunnel (Sakai et al., 2000; Kääb et al., 2001; Miles et al., 2018). In addition, although lake basin size changes on ice-cored moraine complexes depend on the details of the thermal erosion, the basin area of Korumdu lake has increased each year due to glacier retreat. If these conditions and evolution also apply to other short-lived glacier lakes in the Teskey Range, large-scale flooding events during their discharge may become more frequent in the future due to increasing temperature.

## 6 Conclusions

From our field survey (2015–2019), we found that Korumdu lake, located in the Tong region of Teskey Range, northern Tien Shan, appeared and expanded from July to August and then drained over a period of 2–3 weeks. The lake formed when its outlet ice-tunnel closed, which we argue was due to deposition of ice-debris mixture during summer. The lake drainage was

295 always relatively slow. We argue that predicting drainage rates requires knowing the dimensions of the outlet ice-tunnel and the size of the lake basin. By combining water level data and UAV-derived DSMs from consecutive years, we were able to

study the temporal evolution (lake area, volume) and daily lake discharge and approximate the tunnel dimensions to much less than 8 m$^2$. Four lakes that appeared a month earlier (May–June) showed drainage rates significantly higher compared to Korumdu lake. Based on satellite images from 2013–2018, 160 short-lived glacial lakes were detected in the northern Teskey Range, many of which had a similar timing of appearance as Korumdu lake with average 27% forming in June, average 73% in July–September. This result shows the deposition–collapse type is likely the main type for the short-lived glacial lakes in the northern Teskey Range.

Although short-lived glacial lakes in the northern Teskey Range rarely drain through moraine-dam failure, they can be nevertheless a major flood hazard. Moreover, the warming climate may result in larger outlet ice-tunnels and lake basin sizes that could cause large flood events. Therefore, short-lived lakes should be monitored using satellite data and field observations to better understand their characteristics and behaviour. Such monitoring may help mitigate glacier-related hazards in permafrost zones of high-mountain areas of Central Asia.

**Acknowledgements**

Special thanks are due to O. Mordovekov, S. Usupbaev, A. Osmonov of Central-Asian Institute for Applied Geosciences (CAIAG), A. Aitaliev of the Ministry of Emergency Situations of the Kyrgyz Republic, S. Erokhin of Geology Institute of the Kyrgyz Republic, Y. Mori, H. Takadama, N. Sakurai, H. Sugiyama, S. Okuyama, N. Yamada of Niigata University to support field survey and satellite data analysis. Thanks to the editor (Margreth Keiler) and two reviewers (Mauro Fisher and anonymous) who gave us many valuable comments and improvements for rephrasing our manuscript. This study was supported by Japanese Government (Monbukagakusho: MEXT) Scholarships at Graduate School of Science and Technology, Niigata University in 2015–2017, the Sasakawa Scientific Research Grant from the Japan Science Society, Grant-in-Aid for Scientific Research (B) 16H05642 and 19H01372 of the Ministry of Education, Science, Sports and Culture, and Heiwa Nakajima Research Foundation.

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

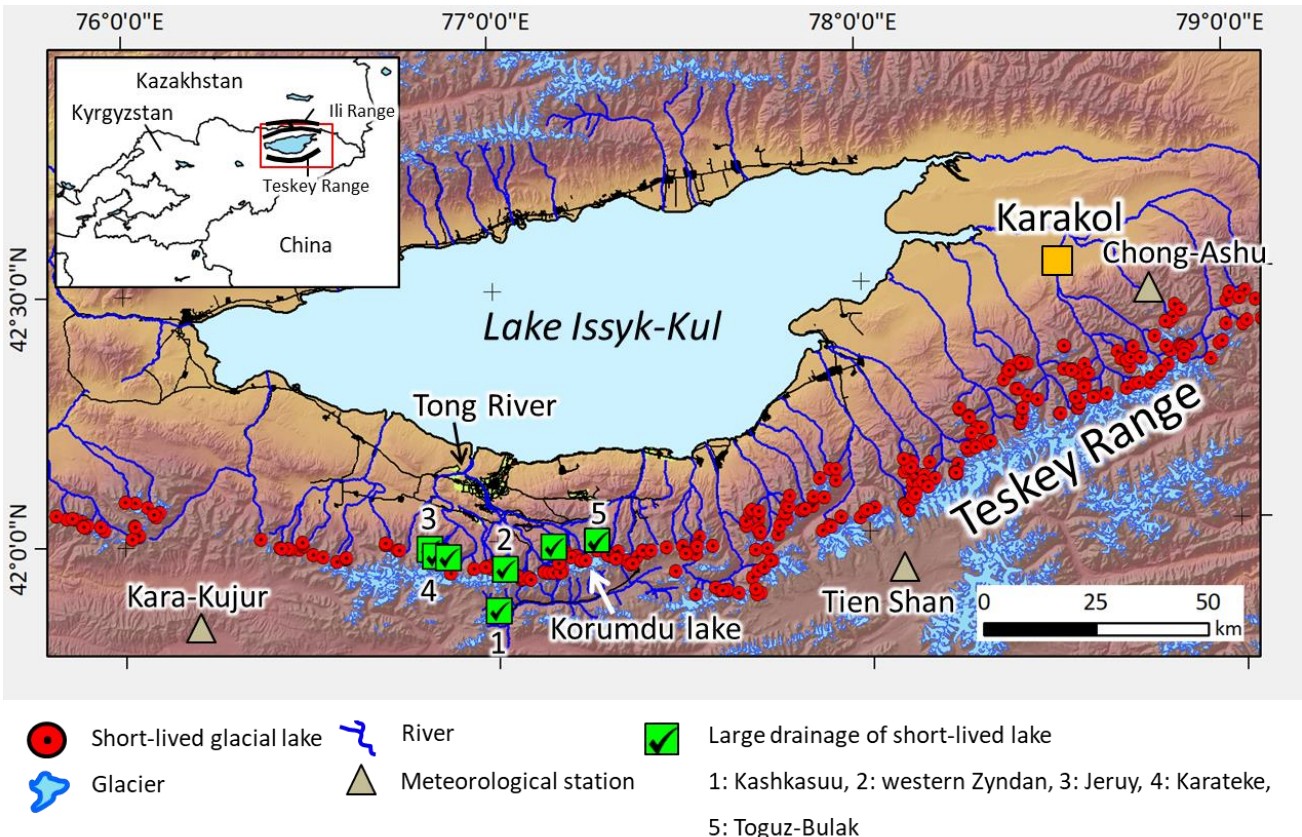

Figure 1: Study area in the northern part of the Teskey Range located on the south of Lake Issyk-Kul, Kyrgyz Republic. Red
circles indicate locations of short-lived glacial lakes that appeared in 2013–2018. Green squares with checks show short-lived
glacial lakes that have caused large drainage events since the 1970s. The shaded relief map was created using SRTM DEM.

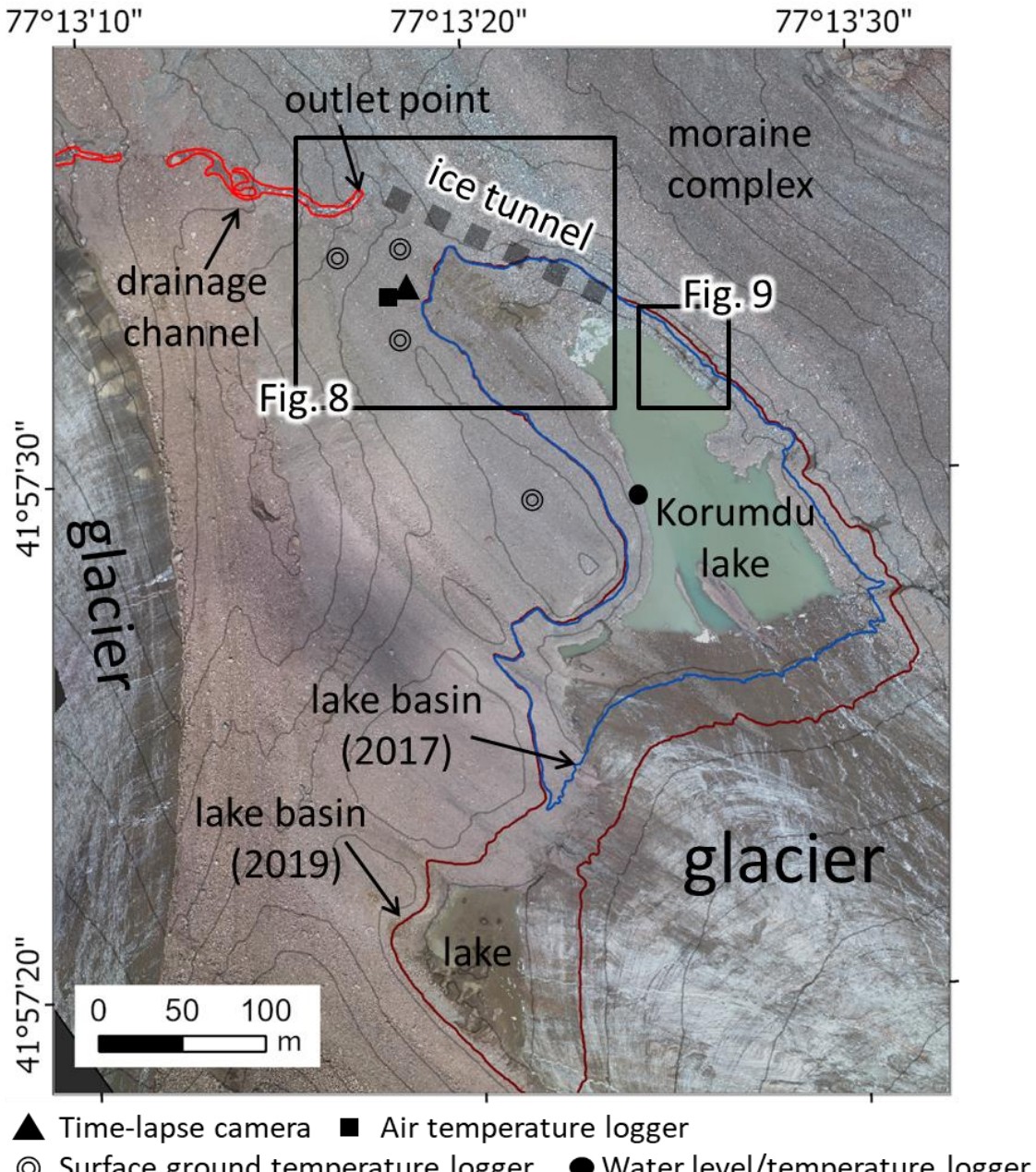

▲ Time-lapse camera ■ Air temperature logger
◎ Surface ground temperature logger ● Water level/temperature logger

Figure 2: Overview of the Korumdu glacier front. The location of the glacier is shown in Fig. 1. Orthoimages were acquired by our UAV imagery in 2019. Contour spacing is 10 m.

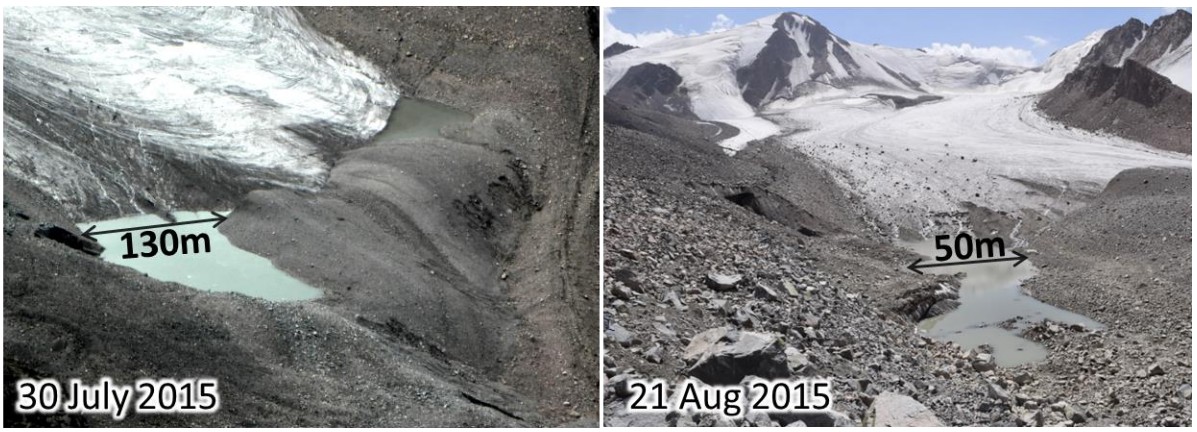

Figure 3: Korumdu glacial lake on 30 July 2015 (from a helicopter) and 21 August 2015 (from field observation). Maximum lake width is about 130 m (left image) and 50 m (right image).

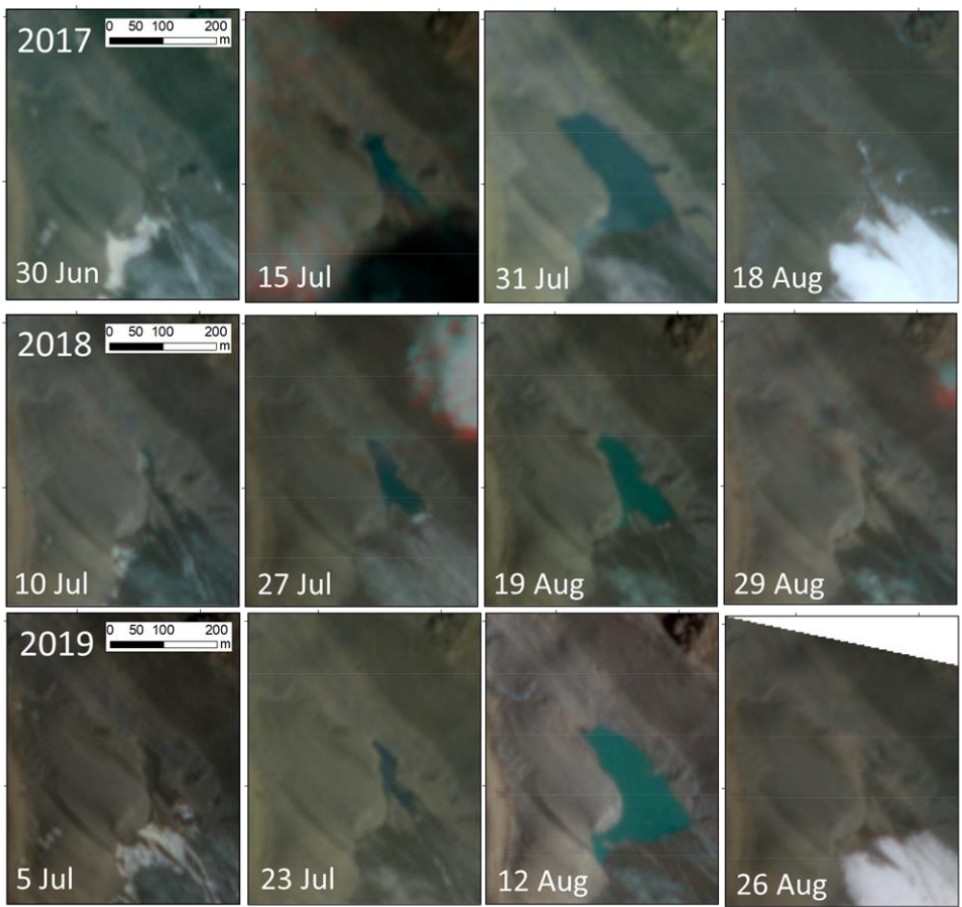

Figure 4: Time sequence of satellite images (PlanetScope) of Korumdu lake in 2017, 2018, and 2019.

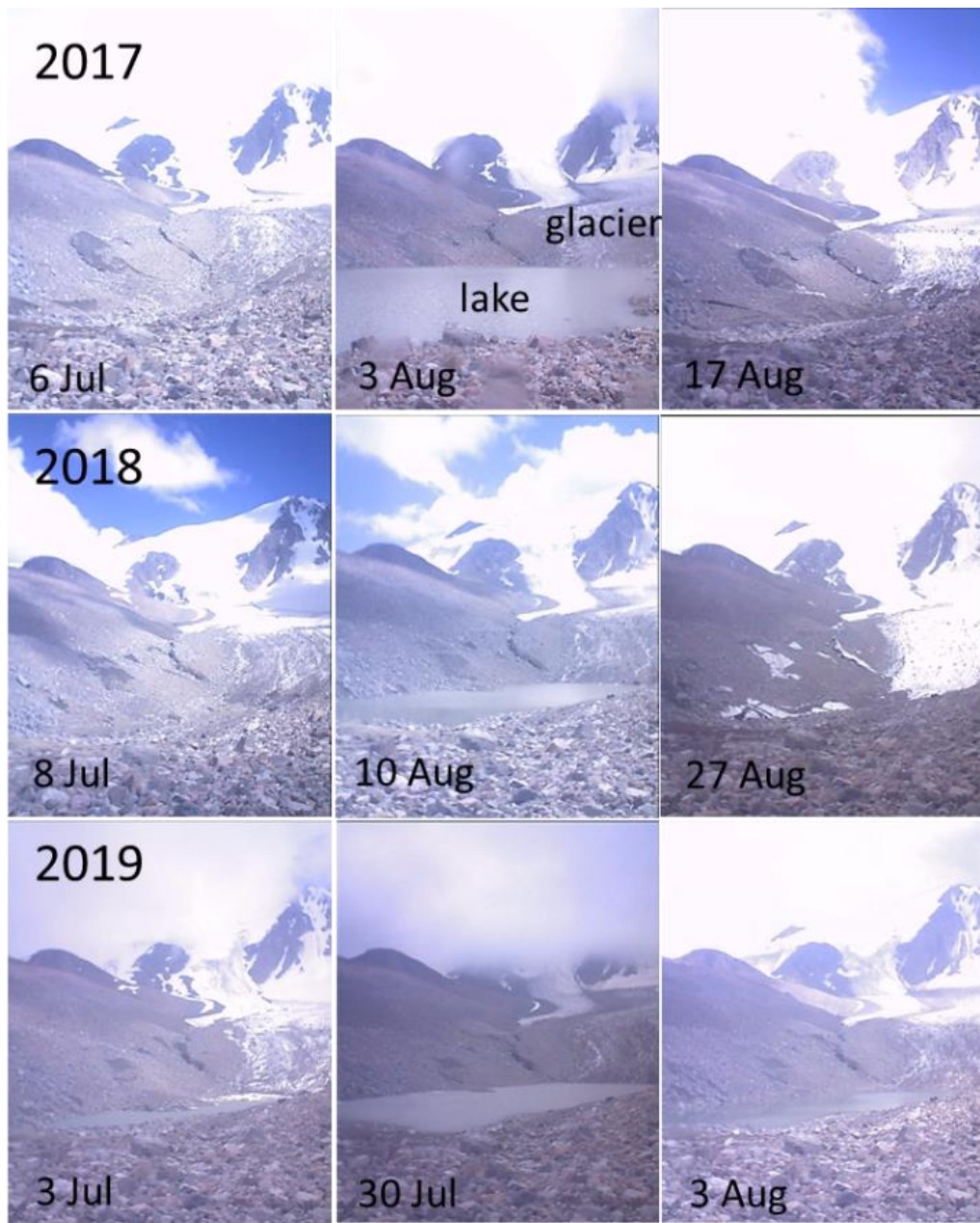

Figure 5: Korumdu lake during 2017–2019 from on-site on time-lapse camera images acquired in the field.

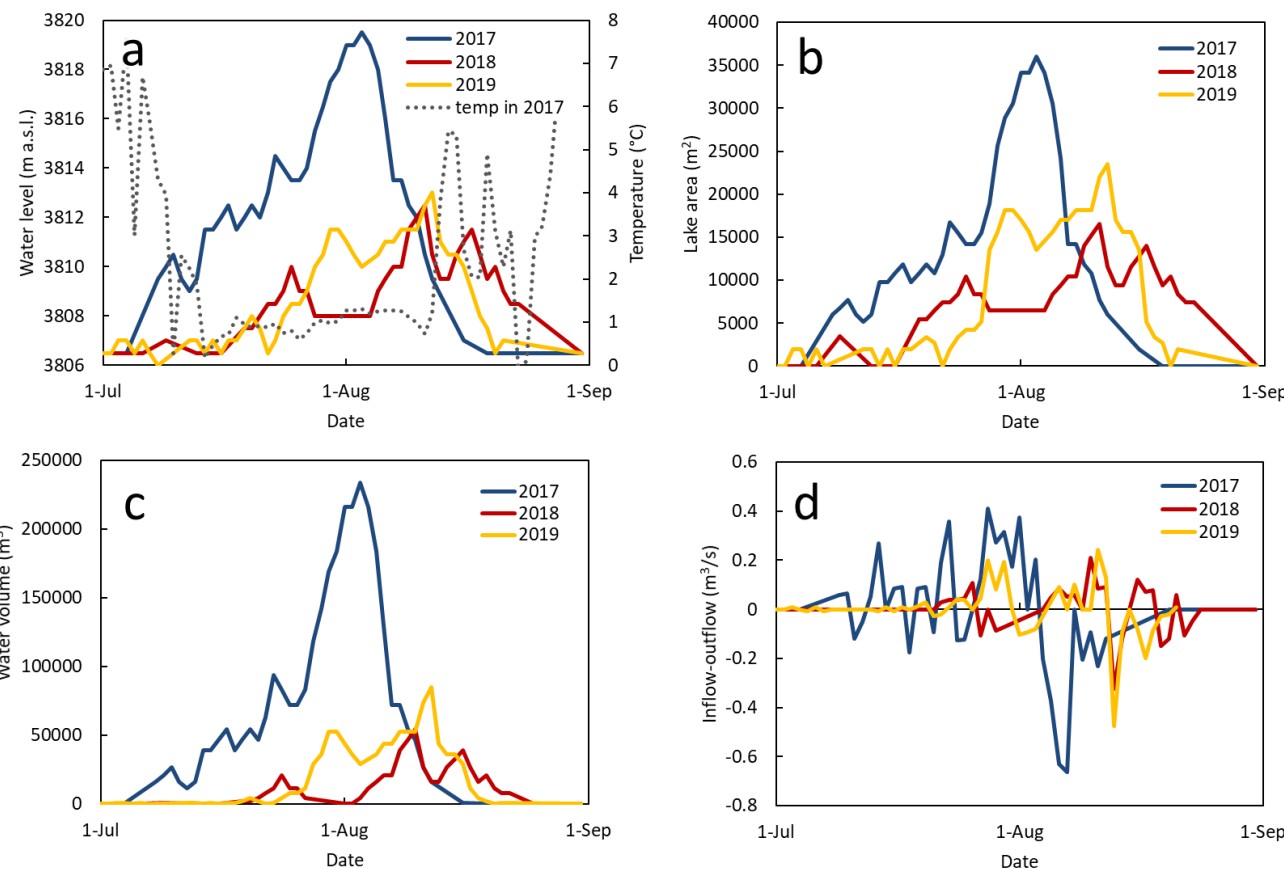

Figure 6: (a) Water levels and water temperature, (b) lake volume, (c) lake area, and (d) inflow-outflow rate of Korumdu lake during summer months of 2017–2019. Data based on water level logger data, UAV-derived DSMs, time-lapse camera images, and PlanetScope satellite images.

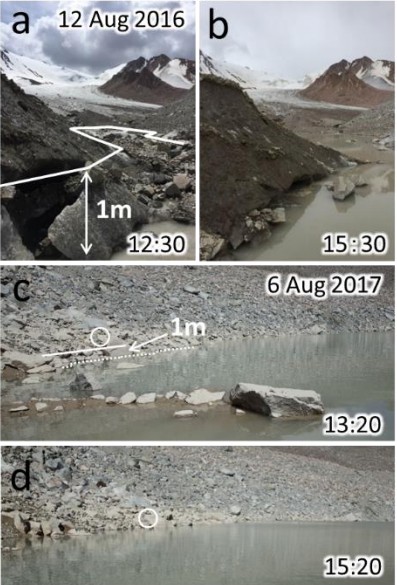

Figure 7: Two examples of a sudden increase in water level of Korumdu lake. (a) On 12 August 2016, (b) Same as (a) except
3 hours later, (c) On 6 August 2017, (d) Same as (c) except 2 hours later. Images taken in the field.

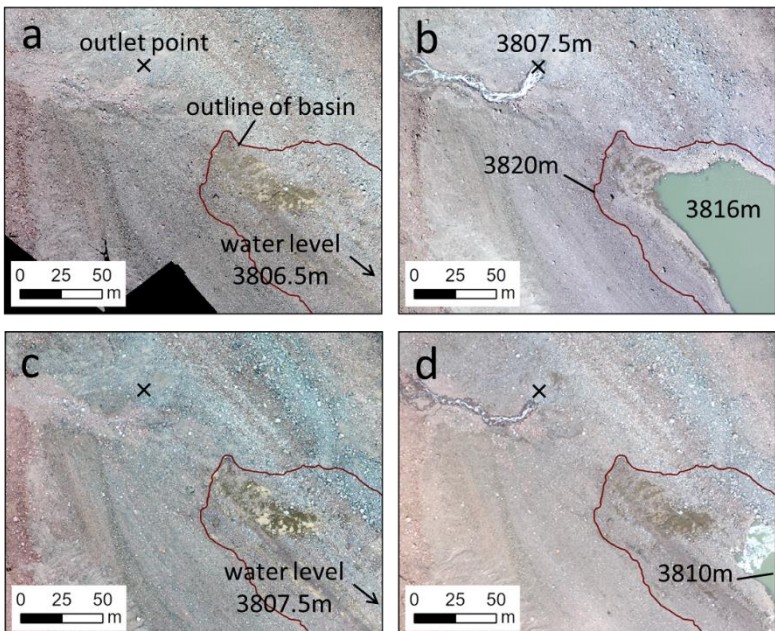

Figure 8: One-day drainage events from Korumdu lake. (a) On 12 August 2016. (b) On 6 August 2017. (c) On 20 July 2018.
(d) On 4 August 2019. Images from UAV surveys.

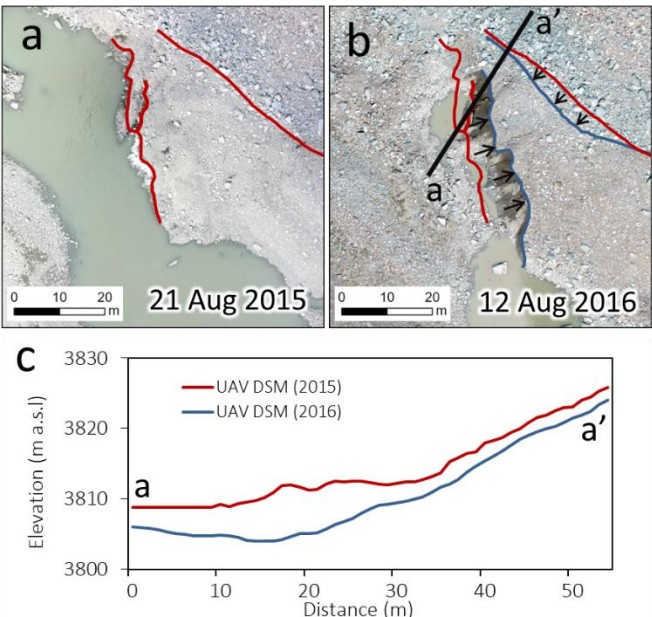

Figure 9: Surface features and elevation profiles of the debris-covered stagnant ice at the entrance of the outlet ice-tunnel based on UAV orthoimages. (a) On 21 August 2015. Left red line shows the position of the exposed ice edge of the debris surface before the ice-cliff underwent backwasting and melting. Right red line shows the deposition line of boulders on the slope. (b) Same as (a) except 12 August 2016. The blue lines show the new positions of the respective surface features after one year. (c) Elevation profile of the surface along line a–a′ in (b).

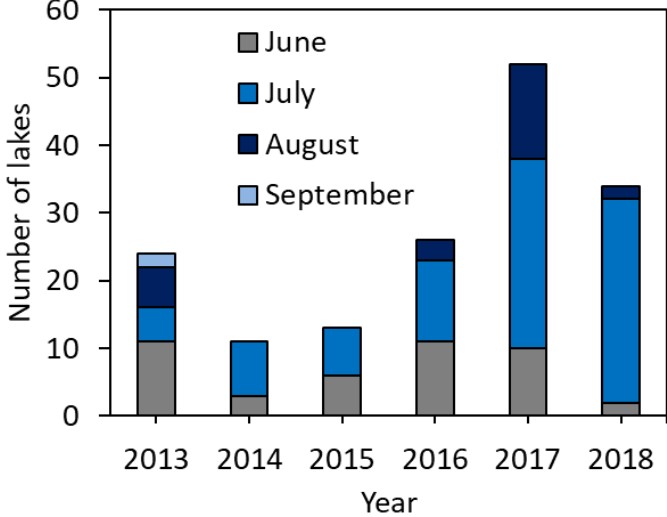

Figure 10: Total number of short-lived lakes in the months of June–September during 2013–2018 in the northern part of the Teskey Range derived by Landsat-7/8, Sentinel-2, and PlanetScope satellite images.

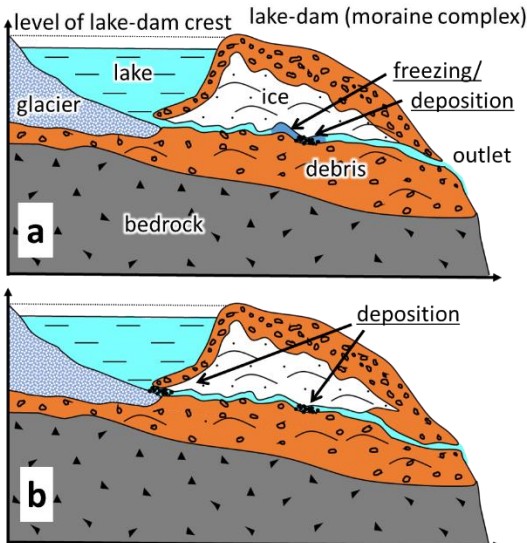

Figure 11: The two types of ice-tunnel closure occurring in the northern Teskey Range. Sketches show cross-sections through a glacier, lake basin, and ice-cored moraine complex in the case of a short-lived lake (based on Popov, 1987). (a) Deposition–freezing type of closure in case of an outlet ice-tunnel being blocked by freezing of storage water or deposition of debris and ice. Dark blue in the tunnel is frozen. (b) Deposition–collapse type of closure in case of an outlet ice-tunnel being blocked by deposition of debris and ice by thermal erosion (ice melt).

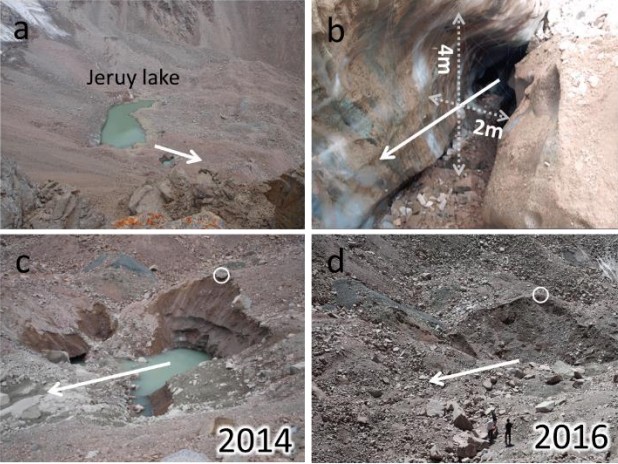

Figure 12: Basin and outlet ice-tunnel of Jeruy lake, which drained on 15 August 2013. (a) Lake basin of Jeruy glacial lake on 9 August 2014. The white arrow shows the direction of lake drainage. (b) Insight into the outlet ice-tunnel on 9 Aug 2014. (c) The outlet ice-tunnel area on 9 August 2014. The white circles in (c) and (d) show the same location. (d) Same as (c) except on 9 August 2016.

## Supplementary Table 1: Satellite images source dates

### Landsat-7 and 8

| ID | Satellite | Acquisition date | Resolution (m) | Lake |
|---|---|---|---|---|
| 1 | Landsat-8/OLI | 12-Jun-13 | 15 (pan) | other short-lived glacial lakes |
| 2 | Landsat-8/OLI | 19-Jun-13 | 15 (pan) | other short-lived glacial lakes |
| 3 | Landsat-8/OLI | 5-Jul-13 | 15 (pan) | other short-lived glacial lakes |
| 4 | Landsat-8/OLI | 28-Jul-13 | 15 (pan) | other short-lived glacial lakes |
| 5 | Landsat-7 ETM+/SLC-off | 29-Jul-13 | 15 (pan) | other short-lived glacial lakes |
| 6 | Landsat-8/OLI | 30-Jul-13 | 15 (pan) | other short-lived glacial lakes |
| 7 | Landsat-8/OLI | 6-Aug-13 | 15 (pan) | other short-lived glacial lakes |
| 8 | Landsat-8/OLI | 15-Aug-13 | 15 (pan) | other short-lived glacial lakes |
| 9 | Landsat-8/OLI | 29-Aug-13 | 15 (pan) | other short-lived glacial lakes |
| 10 | Landsat-8/OLI | 31-Aug-13 | 15 (pan) | other short-lived glacial lakes |
| 11 | Landsat-8/OLI | 1-Sep-13 | 15 (pan) | other short-lived glacial lakes |
| 12 | Landsat-8/OLI | 7-Sep-13 | 15 (pan) | other short-lived glacial lakes |
| 13 | Landsat-7 ETM+/SLC-off | 8-Sep-13 | 15 (pan) | other short-lived glacial lakes |
| 14 | Landsat-8/OLI | 23-Sep-13 | 15 (pan) | other short-lived glacial lakes |
| 15 | Landsat-7 ETM+/SLC-off | 24-Sep-13 | 15 (pan) | other short-lived glacial lakes |
| 16 | Landsat-7 ETM+/SLC-off | 1-Oct-13 | 15 (pan) | other short-lived glacial lakes |
| 17 | Landsat-8/OLI | 9-Oct-13 | 15 (pan) | other short-lived glacial lakes |
| 18 | Landsat-7 ETM+/SLC-off | 10-Oct-13 | 15 (pan) | other short-lived glacial lakes |
| 19 | Landsat-8/OLI | 5-May-14 | 15 (pan) | other short-lived glacial lakes |
| 20 | Landsat-7 ETM+/SLC-off | 29-May-14 | 15 (pan) | other short-lived glacial lakes |
| 21 | Landsat-7 ETM+/SLC-off | 14-Jun-14 | 15 (pan) | other short-lived glacial lakes |
| 22 | Landsat-8/OLI | 15-Jun-14 | 15 (pan) | other short-lived glacial lakes |
| 23 | Landsat-8/OLI | 27-Jun-14 | 15 (pan) | other short-lived glacial lakes |
| 24 | Landsat-7 ETM+/SLC-off | 30-Jun-14 | 15 (pan) | other short-lived glacial lakes |
| 25 | Landsat-8/OLI | 1-Jul-14 | 15 (pan) | other short-lived glacial lakes |
| 26 | Landsat-8/OLI | 8-Jul-14 | 15 (pan) | other short-lived glacial lakes |
| 27 | Landsat-7 ETM+/SLC-off | 9-Jul-14 | 15 (pan) | other short-lived glacial lakes |
| 28 | Landsat-7 ETM+/SLC-off | 16-Jul-14 | 15 (pan) | other short-lived glacial lakes |
| 29 | Landsat-7 ETM+/SLC-off | 25-Jul-14 | 15 (pan) | other short-lived glacial lakes |
| 30 | Landsat-7 ETM+/SLC-off | 1-Aug-14 | 15 (pan) | other short-lived glacial lakes |
| 31 | Landsat-8/OLI | 2-Aug-14 | 15 (pan) | other short-lived glacial lakes |
| 32 | Landsat-7 ETM+/SLC-off | 10-Aug-14 | 15 (pan) | other short-lived glacial lakes |
| 33 | Landsat-8/OLI | 25-Aug-14 | 15 (pan) | other short-lived glacial lakes |
| 34 | Landsat-8/OLI | 1-Sep-14 | 15 (pan) | other short-lived glacial lakes |
| 35 | Landsat-7 ETM+/SLC-off | 2-Sep-14 | 15 (pan) | other short-lived glacial lakes |
| 36 | Landsat-8/OLI | 3-Sep-14 | 15 (pan) | other short-lived glacial lakes |
| 37 | Landsat-8/OLI | 10-Sep-14 | 15 (pan) | other short-lived glacial lakes |
| 38 | Landsat-7 ETM+/SLC-off | 18-Sep-14 | 15 (pan) | other short-lived glacial lakes |
| 39 | Landsat-8/OLI | 19-Sep-14 | 15 (pan) | other short-lived glacial lakes |
| 40 | Landsat-8/OLI | 26-Sep-14 | 15 (pan) | other short-lived glacial lakes |
| 41 | Landsat-7 ETM+/SLC-off | 17-Jun-15 | 15 (pan) | other short-lived glacial lakes |
| 42 | Landsat-8/OLI | 18-Jun-15 | 15 (pan) | other short-lived glacial lakes |
| 43 | Landsat-8/OLI | 2-Jul-15 | 15 (pan) | other short-lived glacial lakes |
| 44 | Landsat-7 ETM+/SLC-off | 3-Jul-15 | 15 (pan) | other short-lived glacial lakes |
| 45 | Landsat-8/OLI | 4-Jul-15 | 15 (pan) | other short-lived glacial lakes |
| 46 | Landsat-8/OLI | 11-Jul-15 | 15 (pan) | other short-lived glacial lakes |
| 47 | Landsat-7 ETM+/SLC-off | 12-Jul-15 | 15 (pan) | other short-lived glacial lakes |
| 48 | Landsat-8/OLI | 18-Jul-15 | 15 (pan) | other short-lived glacial lakes |
| 49 | Landsat-8/OLI | 30-Jul-15 | 15 (pan) | other short-lived glacial lakes |
| 50 | Landsat-8/OLI | 12-Aug-15 | 15 (pan) | other short-lived glacial lakes |
| 51 | Landsat-8/OLI | 19-Aug-15 | 15 (pan) | other short-lived glacial lakes |
| 52 | Landsat-7 ETM+/SLC-off | 20-Aug-15 | 15 (pan) | other short-lived glacial lakes |
| 53 | Landsat-8/OLI | 21-Aug-15 | 15 (pan) | other short-lived glacial lakes |
| 54 | Landsat-7 ETM+/SLC-off | 4-Sep-15 | 15 (pan) | other short-lived glacial lakes |
| 55 | Landsat-7 ETM+/SLC-off | 5-Sep-15 | 15 (pan) | other short-lived glacial lakes |
| 56 | Landsat-8/OLI | 6-Sep-15 | 15 (pan) | other short-lived glacial lakes |
| 57 | Landsat-8/OLI | 13-Sep-15 | 15 (pan) | other short-lived glacial lakes |
| 58 | Landsat-8/OLI | 29-Sep-15 | 15 (pan) | other short-lived glacial lakes |
| 59 | Landsat-8/OLI | 8-Oct-15 | 15 (pan) | other short-lived glacial lakes |
| 60 | Landsat-7 ETM+/SLC-off | 10-Jun-16 | 15 (pan) | other short-lived glacial lakes |
| 61 | Landsat-8/OLI | 11-Jun-16 | 15 (pan) | other short-lived glacial lakes |
| 62 | Landsat-7 ETM+/SLC-off | 26-Jun-16 | 15 (pan) | other short-lived glacial lakes |
| 63 | Landsat-8/OLI | 20-Jun-16 | 15 (pan) | other short-lived glacial lakes |
| 64 | Landsat-7 ETM+/SLC-off | 12-Jul-16 | 15 (pan) | other short-lived glacial lakes |
| 65 | Landsat-8/OLI | 13-Jul-16 | 15 (pan) | other short-lived glacial lakes |
| 66 | Landsat-8/OLI | 20-Jul-16 | 15 (pan) | other short-lived glacial lakes |
| 67 | Landsat-7 ETM+/SLC-off | 21-Jul-16 | 15 (pan) | other short-lived glacial lakes |
| 68 | Landsat-8/OLI | 22-Jul-16 | 15 (pan) | other short-lived glacial lakes |
| 69 | Landsat-8/OLI | 29-Jul-16 | 15 (pan) | other short-lived glacial lakes |
| 70 | Landsat-7 ETM+/SLC-off | 30-Jul-16 | 15 (pan) | other short-lived glacial lakes |
| 71 | Landsat-8/OLI | 29-Jul-16 | 15 (pan) | other short-lived glacial lakes |
| 72 | Landsat-8/OLI | 7-Aug-16 | 15 (pan) | other short-lived glacial lakes |
| 73 | Landsat-7 ETM+/SLC-off | 13-Aug-16 | 15 (pan) | other short-lived glacial lakes |
| 74 | Landsat-8/OLI | 14-Aug-16 | 15 (pan) | other short-lived glacial lakes |
| 75 | Landsat-8/OLI | 21-Aug-16 | 15 (pan) | other short-lived glacial lakes |
| 76 | Landsat-7 ETM+/SLC-off | 22-Aug-16 | 15 (pan) | other short-lived glacial lakes |
| 77 | Landsat-8/OLI | 23-Aug-16 | 15 (pan) | other short-lived glacial lakes |
| 78 | Landsat-7 ETM+/SLC-off | 7-Sep-16 | 15 (pan) | other short-lived glacial lakes |
| 79 | Landsat-8/OLI | 8-Sep-16 | 15 (pan) | other short-lived glacial lakes |
| 80 | Landsat-7 ETM+/SLC-off | 14-Sep-16 | 15 (pan) | other short-lived glacial lakes |
| 81 | Landsat-8/OLI | 15-Sep-16 | 15 (pan) | other short-lived glacial lakes |
| 82 | Landsat-8/OLI | 22-Sep-16 | 15 (pan) | other short-lived glacial lakes |
| 83 | Landsat-7 ETM+/SLC-off | 23-Sep-16 | 15 (pan) | other short-lived glacial lakes |
| 84 | Landsat-8/OLI | 24-Sep-16 | 15 (pan) | other short-lived glacial lakes |
| 85 | Landsat-8/OLI | 14-Jun-17 | 15 (pan) | other short-lived glacial lakes |
| 86 | Landsat-8/OLI | 30-Jun-17 | 15 (pan) | other short-lived glacial lakes |
| 87 | Landsat-8/OLI | 1-Jul-17 | 15 (pan) | other short-lived glacial lakes |
| 88 | Landsat-8/OLI | 7-Jul-17 | 15 (pan) | other short-lived glacial lakes |
| 89 | Landsat-8/OLI | 9-Jul-17 | 15 (pan) | other short-lived glacial lakes |
| 90 | Landsat-8/OLI | 16-Jul-17 | 15 (pan) | other short-lived glacial lakes |
| 91 | Landsat-8/OLI | 25-Jul-17 | 15 (pan) | other short-lived glacial lakes |
| 92 | Landsat-8/OLI | 25-Jul-17 | 15 (pan) | other short-lived glacial lakes |
| 93 | Landsat-8/OLI | 1-Aug-17 | 15 (pan) | other short-lived glacial lakes |
| 94 | Landsat-8/OLI | 8-Aug-17 | 15 (pan) | other short-lived glacial lakes |
| 95 | Landsat-8/OLI | 10-Aug-17 | 15 (pan) | other short-lived glacial lakes |
| 96 | Landsat-8/OLI | 26-Aug-17 | 15 (pan) | other short-lived glacial lakes |
| 97 | Landsat-8/OLI | 2-Sep-17 | 15 (pan) | other short-lived glacial lakes |
| 98 | Landsat-7 ETM+/SLC-off | 17-Sep-17 | 15 (pan) | other short-lived glacial lakes |
| 99 | Landsat-8/OLI | 11-Oct-17 | 15 (pan) | other short-lived glacial lakes |
| 100 | Landsat-8/OLI | 25-Jun-18 | 15 (pan) | other short-lived glacial lakes |
| 101 | Landsat-8/OLI | 10-Jul-18 | 15 (pan) | other short-lived glacial lakes |
| 102 | Landsat-8/OLI | 19-Jul-18 | 15 (pan) | other short-lived glacial lakes |
| 103 | Landsat-8/OLI | 19-Jul-18 | 15 (pan) | other short-lived glacial lakes |
| 104 | Landsat-8/OLI | 5-Sep-19 | 15 (pan) | other short-lived glacial lakes |
| 105 | Landsat-8/OLI | 7-Aug-19 | 15 (pan) | other short-lived glacial lakes |

### Sentinel-2

| ID | Satellite | Acquisition date | Resolution (m) | Lake |
|---|---|---|---|---|
| 1 | Sentinel-2 | 10-Jun-16 | 10 | other short-lived glacial lakes |
| 2 | Sentinel-2 | 30-Jun-16 | 10 | other short-lived glacial lakes |
| 3 | Sentinel-2 | 17-Jul-16 | 10 | other short-lived glacial lakes |
| 4 | Sentinel-2 | 9-Aug-16 | 10 | other short-lived glacial lakes |
| 5 | Sentinel-2 | 29-Aug-16 | 10 | other short-lived glacial lakes |
| 6 | Sentinel-2 | 18-Sep-16 | 10 | other short-lived glacial lakes |
| 7 | Sentinel-2 | 8-Oct-16 | 10 | other short-lived glacial lakes |
| 8 | Sentinel-2 | 12-Jun-17 | 10 | other short-lived glacial lakes |
| 9 | Sentinel-2 | 12-Jun-17 | 10 | other short-lived glacial lakes |
| 10 | Sentinel-2 | 12-Jun-17 | 10 | other short-lived glacial lakes |
| 11 | Sentinel-2 | 12-Jun-17 | 10 | other short-lived glacial lakes |
| 12 | Sentinel-2 | 15-Jun-17 | 10 | other short-lived glacial lakes |
| 13 | Sentinel-2 | 17-Jun-17 | 10 | other short-lived glacial lakes |
| 14 | Sentinel-2 | 21-Jun-17 | 10 | other short-lived glacial lakes |
| 15 | Sentinel-2 | 30-Jun-17 | 10 | other short-lived glacial lakes |
| 16 | Sentinel-2 | 2-Jul-17 | 10 | other short-lived glacial lakes |
| 17 | Sentinel-2 | 7-Jul-17 | 10 | other short-lived glacial lakes |
| 18 | Sentinel-2 | 7-Jul-17 | 10 | other short-lived glacial lakes |
| 19 | Sentinel-2 | 7-Jul-17 | 10 | other short-lived glacial lakes |
| 20 | Sentinel-2 | 9-Jul-17 | 10 | other short-lived glacial lakes |
| 21 | Sentinel-2 | 20-Jul-17 | 10 | other short-lived glacial lakes |
| 22 | Sentinel-2 | 22-Jul-17 | 10 | other short-lived glacial lakes |
| 23 | Sentinel-2 | 27-Jul-17 | 10 | other short-lived glacial lakes |
| 24 | Sentinel-2 | 27-Jul-17 | 10 | other short-lived glacial lakes |
| 25 | Sentinel-2 | 27-Jul-17 | 10 | other short-lived glacial lakes |
| 26 | Sentinel-2 | 27-Jul-17 | 10 | other short-lived glacial lakes |
| 27 | Sentinel-2 | 27-Jul-17 | 10 | other short-lived glacial lakes |
| 28 | Sentinel-2 | 9-Aug-17 | 10 | other short-lived glacial lakes |
| 29 | Sentinel-2 | 14-Aug-17 | 10 | other short-lived glacial lakes |
| 30 | Sentinel-2 | 29-Aug-17 | 10 | other short-lived glacial lakes |
| 31 | Sentinel-2 | 29-Aug-17 | 10 | other short-lived glacial lakes |
| 32 | Sentinel-2 | 31-Aug-17 | 10 | other short-lived glacial lakes |
| 33 | Sentinel-2 | 31-Aug-17 | 10 | other short-lived glacial lakes |
| 34 | Sentinel-2 | 31-Aug-17 | 10 | other short-lived glacial lakes |
| 35 | Sentinel-2 | 31-Aug-17 | 10 | other short-lived glacial lakes |
| 36 | Sentinel-2 | 18-Sep-17 | 10 | other short-lived glacial lakes |
| 37 | Sentinel-2 | 20-Sep-17 | 10 | other short-lived glacial lakes |
| 38 | Sentinel-2 | 20-Sep-17 | 10 | other short-lived glacial lakes |
| 39 | Sentinel-2 | 20-Sep-17 | 10 | other short-lived glacial lakes |
| 40 | Sentinel-2 | 20-Sep-17 | 10 | other short-lived glacial lakes |
| 41 | Sentinel-2 | 23-Sep-17 | 10 | other short-lived glacial lakes |
| 42 | Sentinel-2 | 10-Oct-17 | 10 | other short-lived glacial lakes |
| 43 | Sentinel-2 | 10-Jun-18 | 10 | other short-lived glacial lakes |
| 44 | Sentinel-2 | 17-Jun-18 | 10 | other short-lived glacial lakes |
| 45 | Sentinel-2 | 22-Jun-18 | 10 | other short-lived glacial lakes |
| 46 | Sentinel-2 | 22-Jun-18 | 10 | other short-lived glacial lakes |
| 47 | Sentinel-2 | 27-Jun-18 | 10 | other short-lived glacial lakes |
| 48 | Sentinel-2 | 27-Jun-18 | 10 | other short-lived glacial lakes |
| 49 | Sentinel-2 | 27-Jun-18 | 10 | other short-lived glacial lakes |
| 50 | Sentinel-2 | 30-Jun-18 | 10 | other short-lived glacial lakes |
| 51 | Sentinel-2 | 7-Jul-18 | 10 | other short-lived glacial lakes |
| 52 | Sentinel-2 | 7-Jul-18 | 10 | other short-lived glacial lakes |
| 53 | Sentinel-2 | 7-Jul-18 | 10 | other short-lived glacial lakes |
| 54 | Sentinel-2 | 10-Jul-18 | 10 | other short-lived glacial lakes |
| 55 | Sentinel-2 | 22-Jul-18 | 10 | other short-lived glacial lakes |
| 56 | Sentinel-2 | 25-Jul-18 | 10 | other short-lived glacial lakes |
| 57 | Sentinel-2 | 25-Jul-18 | 10 | other short-lived glacial lakes |
| 58 | Sentinel-2 | 27-Jul-18 | 10 | other short-lived glacial lakes |
| 59 | Sentinel-2 | 30-Jul-18 | 10 | other short-lived glacial lakes |
| 60 | Sentinel-2 | 30-Jul-18 | 10 | other short-lived glacial lakes |
| 61 | Sentinel-2 | 30-Jul-18 | 10 | other short-lived glacial lakes |
| 62 | Sentinel-2 | 1-Aug-18 | 10 | other short-lived glacial lakes |
| 63 | Sentinel-2 | 1-Aug-18 | 10 | other short-lived glacial lakes |
| 64 | Sentinel-2 | 1-Aug-18 | 10 | other short-lived glacial lakes |
| 65 | Sentinel-2 | 1-Aug-18 | 10 | other short-lived glacial lakes |
| 66 | Sentinel-2 | 1-Aug-18 | 10 | other short-lived glacial lakes |
| 67 | Sentinel-2 | 4-Aug-18 | 10 | other short-lived glacial lakes |
| 68 | Sentinel-2 | 4-Aug-18 | 10 | other short-lived glacial lakes |
| 69 | Sentinel-2 | 4-Aug-18 | 10 | other short-lived glacial lakes |
| 70 | Sentinel-2 | 9-Aug-18 | 10 | other short-lived glacial lakes |
| 71 | Sentinel-2 | 11-Aug-18 | 10 | other short-lived glacial lakes |
| 72 | Sentinel-2 | 11-Aug-18 | 10 | other short-lived glacial lakes |
| 73 | Sentinel-2 | 11-Aug-18 | 10 | other short-lived glacial lakes |
| 74 | Sentinel-2 | 11-Aug-18 | 10 | other short-lived glacial lakes |
| 75 | Sentinel-2 | 14-Aug-18 | 10 | other short-lived glacial lakes |
| 76 | Sentinel-2 | 14-Aug-18 | 10 | other short-lived glacial lakes |
| 77 | Sentinel-2 | 14-Aug-18 | 10 | other short-lived glacial lakes |
| 78 | Sentinel-2 | 16-Aug-18 | 10 | other short-lived glacial lakes |
| 79 | Sentinel-2 | 16-Aug-18 | 10 | other short-lived glacial lakes |
| 80 | Sentinel-2 | 16-Aug-18 | 10 | other short-lived glacial lakes |
| 81 | Sentinel-2 | 16-Aug-18 | 10 | other short-lived glacial lakes |
| 82 | Sentinel-2 | 20-Aug-18 | 10 | other short-lived glacial lakes |
| 83 | Sentinel-2 | 29-Aug-18 | 10 | other short-lived glacial lakes |
| 84 | Sentinel-2 | 31-Aug-18 | 10 | other short-lived glacial lakes |
| 85 | Sentinel-2 | 31-Aug-18 | 10 | other short-lived glacial lakes |
| 86 | Sentinel-2 | 31-Aug-18 | 10 | other short-lived glacial lakes |
| 87 | Sentinel-2 | 31-Aug-18 | 10 | other short-lived glacial lakes |
| 88 | Sentinel-2 | 3-Sep-18 | 10 | other short-lived glacial lakes |
| 89 | Sentinel-2 | 5-Sep-18 | 10 | other short-lived glacial lakes |
| 90 | Sentinel-2 | 5-Sep-18 | 10 | other short-lived glacial lakes |
| 91 | Sentinel-2 | 5-Sep-18 | 10 | other short-lived glacial lakes |
| 92 | Sentinel-2 | 8-Sep-18 | 10 | other short-lived glacial lakes |
| 93 | Sentinel-2 | 8-Sep-18 | 10 | other short-lived glacial lakes |
| 94 | Sentinel-2 | 13-Sep-18 | 10 | other short-lived glacial lakes |
| 95 | Sentinel-2 | 30-Sep-18 | 10 | other short-lived glacial lakes |
| 96 | Sentinel-2 | 30-Sep-18 | 10 | other short-lived glacial lakes |
| 97 | Sentinel-2 | 27-Jun-19 | 10 | other short-lived glacial lakes |
| 98 | Sentinel-2 | 27-Jun-19 | 10 | other short-lived glacial lakes |
| 99 | Sentinel-2 | 27-Jun-19 | 10 | other short-lived glacial lakes |
| 100 | Sentinel-2 | 30-Jun-19 | 10 | other short-lived glacial lakes |
| 101 | Sentinel-2 | 5-Jul-19 | 10 | other short-lived glacial lakes |
| 102 | Sentinel-2 | 12-Jul-19 | 10 | other short-lived glacial lakes |
| 103 | Sentinel-2 | 15-Jul-19 | 10 | other short-lived glacial lakes |
| 104 | Sentinel-2 | 30-Jul-19 | 10 | other short-lived glacial lakes |
| 105 | Sentinel-2 | 1-Aug-19 | 10 | other short-lived glacial lakes |
| 106 | Sentinel-2 | 4-Aug-19 | 10 | other short-lived glacial lakes |
| 107 | Sentinel-2 | 4-Aug-19 | 10 | other short-lived glacial lakes |
| 108 | Sentinel-2 | 9-Aug-19 | 10 | other short-lived glacial lakes |
| 109 | Sentinel-2 | 11-Aug-19 | 10 | other short-lived glacial lakes |
| 110 | Sentinel-2 | 11-Aug-19 | 10 | other short-lived glacial lakes |
| 111 | Sentinel-2 | 11-Aug-19 | 10 | other short-lived glacial lakes |
| 112 | Sentinel-2 | 11-Aug-19 | 10 | other short-lived glacial lakes |
| 113 | Sentinel-2 | 29-Aug-19 | 10 | other short-lived glacial lakes |
| 114 | Sentinel-2 | 29-Aug-19 | 10 | other short-lived glacial lakes |
| 115 | Sentinel-2 | 31-Aug-19 | 10 | other short-lived glacial lakes |
| 116 | Sentinel-2 | 31-Aug-19 | 10 | other short-lived glacial lakes |
| 117 | Sentinel-2 | 31-Aug-19 | 10 | other short-lived glacial lakes |
| 118 | Sentinel-2 | 3-Sep-19 | 10 | other short-lived glacial lakes |
| 119 | Sentinel-2 | 3-Sep-19 | 10 | other short-lived glacial lakes |
| 120 | Sentinel-2 | 23-Sep-19 | 10 | other short-lived glacial lakes |
| 121 | Sentinel-2 | 23-Sep-19 | 10 | other short-lived glacial lakes |
| 122 | Sentinel-2 | 25-Sep-19 | 10 | other short-lived glacial lakes |
| 123 | Sentinel-2 | 25-Sep-19 | 10 | other short-lived glacial lakes |
| 124 | Sentinel-2 | 25-Sep-19 | 10 | other short-lived glacial lakes |

### PlanetScope

| ID | Satellite | Acquisition date | Resolution (m) | lake basin extraction |
|---|---|---|---|---|
| 1 | Planet Scope | 28-Jul-13 | 3 | analysis short-lived glacial lakes |
| 2 | Planet Scope | 18-Aug-13 | 3 | analysis short-lived glacial lakes |
| 3 | Planet Scope | 27-Jun-14 | 3 | analysis short-lived glacial lakes |
| 4 | Planet Scope | 30-Jun-17 | 3 | Korumdu lake |
| 5 | Planet Scope | 15-Jul-17 | 3 | Korumdu lake |
| 6 | Planet Scope | 31-Jul-17 | 3 | Korumdu lake |
| 7 | Planet Scope | 18-Aug-17 | 3 | Korumdu lake |
| 8 | Planet Scope | 10-Jul-18 | 3 | Korumdu lake |
| 9 | Planet Scope | 27-Jul-18 | 3 | Korumdu lake |
| 10 | Planet Scope | 19-Aug-18 | 3 | Korumdu lake |
| 11 | Planet Scope | 23-Aug-18 | 3 | Korumdu lake |
| 12 | Planet Scope | 29-Aug-18 | 3 | Korumdu lake |
| 13 | Planet Scope | 30-Aug-18 | 3 | Korumdu lake |
| 14 | Planet Scope | 5-Jul-19 | 3 | Korumdu lake |
| 15 | Planet Scope | 23-Jul-19 | 3 | Korumdu lake |
| 16 | Planet Scope | 8-Aug-19 | 3 | Korumdu lake |
| 17 | Planet Scope | 12-Aug-19 | 3 | Korumdu lake |
| 18 | Planet Scope | 26-Aug-19 | 3 | Korumdu lake |

### ALOS/AVNIR-2

| ID | Satellite | Acquisition date | Resolution (m) | Used for… |
|---|---|---|---|---|
| 1 | ALOS/AVNIR-2 | 2007/9/17 | 2.5 | lake basin extraction |