# Peer review of "Formation, evolution and drainage of short-lived glacial lakes in permafrost environments of the northern Teskey Range, Central Asia"

_Natural Hazards and Earth System Sciences, 2020_

## Referee Comment (RC1) · Mauro Fischer (Referee) · 19 Nov 2020

**Review by Mauro Fischer (Glaciologist at the Institute of Geography, University of Bern, Switzerland, mauro.fischer@giub.unibe.ch) of the manuscript „Water storage and drainage of short-lived lakes in the Teskey Range, Central Asia" submitted by Mirlan Daiyrov and Chiyuki Narama to Natural Hazards and Earth System Sciences (NHESS)**

**Summary**

In their manuscript submitted to Natural Hazards and Earth System Sciences (NHESS), Mirlan Daiyrov and Chiyuki Narama present their findings about the formation and drainage of short-lived glacial lakes occurring on or due to the presence of ice-cored moraine complexes (i. e. in permafrost zones) of the northern Teskey Range, Kyrgyz Republic (Central Asia). Based on extensive field surveys carried out during the summer months of 2014–2019 and use of various satellite imagery data, they show interesting results about the formation, evolution and drainage of Korumdu lake in the western part of the northern Teskey Range. They found that Korumdu lake appeared each summer (except in 2016) due to the blockage of a subsurface outlet ice-tunnel by deposition of debris and ice caused by thermal erosion (ice melt). Korumdu lake rapidly increased in volume and area each summer, but always drained within about one month due to the opening of its outlet ice-tunnel. Drainage rates from Korumdu lake were rather small, but could increase in future according to the authors due to climate change, increasing temperatures and enlarging of the outlet ice-tunnel. The behavior of and drainage from Korumdu lake was compared to four other short-lived glacial lakes of the northern Teskey Range, for which large drainage events and downstream flooding were reported in recent years. In addition, based on a series of satellite images from various sources, the authors detected 160 short-lived glacial lakes spread over the entire northern Teskey Range, and analyzed their timing of appearance during the summer months of 2013–2018. The authors show that insights from monitoring short-lived glacial lakes in permafrost zones are useful to better understand their characteristics and behavior, and therefore important for mitigation of glacier-related hazards in high-mountain areas of Central Asia.

**General comments**

I want to thank the authors for their work and valuable scientific contribution. The study contains (partly) novel and and interesting findings, the presented results seem solid overall and are of interest to the scientific community. I think that this work deserves to get published in Natural Hazards and Earth Syste Sciences. However, in my opinion, the authors still need to put quite some effort into improvement of their manuscript. As it is, it cannot be published. There are some important major and an extensive number of minor issues which definitely need to be addressed, corrected, clarified, extended and implemented prior to publication. I guess that – even if there are quite a few – the majority of the specific comments listed below are easy to implement, whereas some specific comments will maybe need some additional work and time. I hope that my work will help improving the paper, and I encourage the authors to implement and reply to all my comments as far as possible. Thanks a lot, all the best and kind regards.

*List of some general comments:*

- In general, linguistic and content clarity and correctness need to be substantially improved (see my introduction to the specific comments below)

- For some parts, the manuscript also needs major improvement in terms of content (see various specific comments thereupon below). For instance, the conclusions have rather "discussion character". You need to add an introductory paragraph to the conclusions, elaborating the research questions and what you did in order to answer them (including a short summary about your study area and methods). Then you can summarize your results and contextualize the meanings of your findings

- You have used so many different satellite imagery data from various sources and for many different things. I think for the reader of your paper it would be beneficial to add a table containing sources or platforms (i. e. Landsat, Sentinel-2 etc.) of the satellite images used, also listing source dates, resolution and purpose (i. e. which satellite imagery you used for what). You could then refer to this table in the manuscript wherever needed instead of having to write long and rather choppy sentences

- Is there a reason why Korumdu lake was chosen for your extensive fieldwork?

- In 2018, there was no water leakage at the outlet point close to Korumdu lake, but the lake emptied anyway. Do you know how, through which processes, and why? – Would have been interesting to add in the discussion

- In my opinion, it would have been interesting to know why, in the Teskey Range, there are "so many" short-lived glacial lakes… is this mainly due to the presence of permafrost (ice-cored moraines)? It would have been interesting to elaborate "a list of geomorphological prerequisites or conditions" for the formation of the type of short-lived glacial lakes occurring in the northern Teskey Range (i. e. presence of sufficient debris, location in permafrost zone, retreating (debris-covered) glacier etc.). If possible, it would have been interesting to compare the spatial density or number of short-lived glacial lakes in the northern Teskey Range to other high-mountain areas…

**Specific comments and technical corrections**

To facilitate the author's correction of the manuscript I combined specific comments and technical corrections (including language and comprehensibility issues). Sometimes comments contain both specific comments and technical corrections, sometimes just one or the other. I took a lot of time to rephrase beforehand rather unclear text passages and tried to improve a lot of the manuscript. I hope that helps and that the authors appreciate my efforts in this regard. I would ask the authors to implement my comments and suggestions as far as possible.

*Title*

I think it would be more attractive, preciser and more transparent to change the title of the manuscript into "Formation, evolution and drainage of short-lived glacial lakes in permafrost environments of the northern Teskey Range, Central Asia"

*Abstract*

Ln 11f: Not clear to me if you refer to short-lived lakes in general here or if this sentence already refers to processes of drainage of the short-lived Korumdu lake. If the former applies, you have to write "Short-lived lakes grow rapidly and drain within a few months, due to…"; if you already refer to Korumdu lake here you would have to state this, e.g. "Korumdu lake, a short-lived glacier lake in the Teskey range surveyed in detail for this study, grows rapidly and drains within a few months, due to…"

Ln 12: "… in moraine complex…" → "…in **a** moraine complex at **the** glacier front."

Lns 12f: This is a general statement → you have to write "Outburst floods of this lake type are a major hazard in this region and differ from many cases…"

Lns 14f: This sentence starts with a statement on the drainage of short-lived glacier lakes in general, i.e. it's not clear if you then write "…, we examined its recent changes…" → be precise and write "…, we examined recent changes of Korumdu lake in water level, area, volume and discharge…", or if you refer to the whole sample of short-lived glacier lakes you analysed in the Teskey Range you have to write "…, we examined recent changes in water level, area, volume and discharge of short-lived glacier lakes in the Teskey Range with a field survey and satellite data analysis."

Ln 16: "… during all summers during…" → "… during all summers between 2014 and 2019 except in 2016."

Ln 14 vs. Ln 16: "water level" vs. "water-level" → be consistent and write "water level" everywhere in the manuscript

Lns 16f: How this sentence is written one would think that the sudden appearance and expansion of Korumdu lake took only place 2017-2019, not before, correct? Below in the methods section you write that you carried out field surveys from 2015 to 2019. So, during summers 2015 and 2016 there was no sudden appearance and expansion (and drainage) of Korumdu lake? I think it would be good to clarify/be more precise here.

Ln 17: "The timing…" which timing? – be precise in order to be clear → "The timing of lake appearance/lake formation indicates…"

Ln 18: "drain" → "drainage"

Ln 19: Here, the abbreviations "UAV" and "DSM" appear for the first time. I would write them out and put the abbreviations in parentheses, unless it's ok to do that in the abstract (but then you would have to write the terms out when they appear for the first time below in the manuscript. Moreover, a lake is per definition made of water → in my opinion you can just write "…, the lake's volume reached…"

Ln 22: "…that caused large drainages…" → "…that showed larger drainage rates…"

Lns 22f: This is a general statement → write "As a result, the dimensions of outlet ice-tunnels of short-lived glacial lakes…". Moreover (for the entire manuscript), in my opinion "flooding scale" is a somewhat misleading and not very precise term… here you refer to discharge (rates) and of course smaller discharge over longer time means less hazard potential for flooding downstream… Be preciser/clarify!

Ln 24: "basin volumes" is again a bit misleading, I would just write "… both tunnel size and lake volume…"

Ln 24: I would replace "greater hazard" with "increased hazard potential"

Lns 24f: I thinks it's more clear to write "In addition to our field surveys of Korumdu lake, we investigated…

Ln 25: It's clearer to replace "in this region" with "in the Teskey Range"

Ln 27: This is a general statement → write "The appearance of short-lived glacial lakes is inevitable in summer when the melting rate is high." Moreover, I think this general statement is not clear enough yet! For which region, for which glaciers, for which type and composition of forefield characteristics does this statement/sentence apply? In my opinion you need to clarify this, otherwise you cannot just leave this as a general statement which could be interpreted as being "valid everywhere"…

Lns 27f: This sentence is not yet clear enough to me. Do you mean something like "Similar characteristics of short-lived lake formation and drainage through blockage and opening of subsurface channels might also be found in other mountain regions of (Central) Asia"? Moreover, this is "just a guess" and I think as a concluding sentence of your abstract I would try to be a bit preciser and clearer. You also mention the term "permafrost" for the first time here. Of course this is important for the short-lived lakes you're looking at (in terms of the frozen material, i.e. moraine complexes in the glacier forefield). So again, try "not just to guess" but try to say why and where in high-mountain (Central) Asia you would expect to find similar processes, lake types and characteristics of formation and drainage! Seems important and interesting to me…

*Introduction*

Lns 32f: clearer to write "…, rather small glacial lakes can be found close to the present termini of glaciers in the northern Tien Shan (Central Asia) (References)."

Lns 33f: I would write "…often produce hazardous debris flows and floods." Like this, the link/logical connection to the subsequent sentence is much better…

Ln 35: I would insert commas before and after "including casualties" or put "including casualties" in parentheses for better/smoother readability

Ln 36: Reference "Daiyrov et al., under review" → Please do only cite research articles that are either already accepted for publication or already published!

Ln 36: "Such short-lived glacial lakes…": Here you mention "short-lived" for the first time in the introduction. Above it's just about small glacial lakes. So I guess one could assume that all small glacial lakes are short-lived, which of course is not true. In order to be a bit clearer I would therefore add a sentence or a subordinate clause here to precisely state that some/a certain number of these small lakes are "short-lived" or "unstable" (or if you are sure about the percentage of small glacial lakes in the Tien Shan that are "short-lived" you can write "many/the majority of small lakes are short-lived"…), then you can add that "they grow rapidly and drain within a few months"

Lns 38f: "Such lakes drain through…" here you refer again to small and short-lived proglacial lakes dammed by ice-cored moraine complex, ok, but I think the clarity of the introduction (and the manuscript in general) would benefit from explaining your focus a bit in more detail (including further references), i.e. explaining in more detail which types of short-lived glacial lakes you're looking at/focusing on. – I mean, supraglacial lakes are often also "short-lived", i.e. appear and disappear over the summer season, but processes are different from short-lived lakes in the glacier forefield where frozen moraine material plays a role as well… Not all small glacier lakes in the glacier forefield are short-lived and not all short-lived glacier lakes show the same formation and drainage processes as the ones you're specifically focusing on… I would appreciate if you could elaborate these issues a bit more clear, precise and complete/holistic in the introduction (this would for instance also include an explanation which nonstationary lakes are short-lived and which have a longer lifetime and why (cf. Lns 39f). Partly this is already done further below in the introduction… maybe you could rearrange the different sections of the introduction a bit in order to avoid confusion about what you mean by "small lakes" or "short-lived lakes"…

Lns 40f: "A short-lived lake can be a severe hazard…", next sentence "The short-lived lakes are a major hazard…" → somehow a repetition, I think you could easily merge these two statements to avoid this

Lns 40f: About the use of the term "hazard" or "natural hazard". Please be aware that there is a difference between "hazard" and "risk"! As far as I understand these terms, "hazards" or "natural hazards" are just geomorphological processes which take place (naturally) and which can potentially be dangerous for people, infrastructure and goods. Important factors defining "hazards" are, for instance, duration, intensity/magnitude, spatial extent and return period of the respective geomorphological processes. "Risk" refers to the combination of the probability of occurrence and the damage potential of an event (whereas damage potential is a function of exposition and vulnerability). I was not always quite sure if you actually refer to "hazard" or "risk" whenever you use the term "hazard" in your manuscript. Please check if you use these terms correctly everywhere.

Ln 41: If you write "The short-lived lakes are a major hazard in this region…", this means "all short-lived lakes in the area" are concerned. I would just write "Short-lived lakes are a major hazard…"Moreover I would be precise again and replace "in this region" with "in the Teskey Range" or with "in northern Tien Shan".

Lns 41f: "…, and differs from the outburst which caused by…" → sentence ist not fully correct (English). I don't know if you refer to the differences in characteristics of lake drainage (i.e. processes) here or if you refer to the different damage potential/risk of the different lake types and outburst mechanisms here… Please clarify, write more precisely what you mean

Ln 44: "As such glacial lakes" → which glacial lakes? I know which ones you mean but I think it's important here to precisely write which types of glacial lakes you're talking about (→ "small and short-lived proglacial lakes which are dammed by (partially) frozen moraine material/sediments" would be clear, wouldn't it?)

Ln 44: "Small and short-lived proglacial lakes which are dammed by (partially) frozen moraine material/sediments drain through a subsurface outlet ice-tunnel. These lakes can expand rapidly when the outlet ice-tunnel…." I think it's better and clearer like this

Ln 46: "Some short-lived glacial lakes" → clearer if you write "Some of these/the aforementioned short-lived glacial lakes…"

Lns 46f: "…, which is behavior they share with supraglacial lakes on a debris-covered glacier" → not only the case for supraglacial lakes on debris-coverd glaciers. See for instance "Gornersee" or "Lac de Faverges" on the (quasi debris-free) "Gorner-/Grenzgletscher" or "Glacier de la Plaine Morte" in the Swiss Alps in recent years… (e.g. Huss et al. 2007 in Journal of Glaciology (doi: 10.3189/172756507782202784) or Huss et al. 2013 in Geographica Helvetica (doi:10.5194/gh-68-227-2013) → I would just delete the "on a debris-covered glacier"

Lns 47ff: "Several studies have examined the relationship between supraglacial lakes and englacial conduit on a debris-covered glacier (Benn et al., 2000, 2017; Miles et al., 2016; Watson et al., 2016; Narama et al., 2017), but this relationship has seen little study for glacial lakes." → To me it's not clear what you want to say with this sentence… Please rephrase the sentence in order to be clear

Ln 50: "Short-lived glacial lakes" → again, it has to be clear which types of short-lived glacial lakes you're talking about… You could refer to the definition above (it's about "Small and short-lived proglacial lakes which are dammed by (partially) frozen moraine material/sediments"), maybe you could introduce an abbreviation for the types of short-lived glacial lakes you're investigating and use this abbreviation after having precisely introduced it in order to facilitate smooth reading and avoid misunderstandings, increase clarity…

Ln 50: delete "either"

Ln 51: better "… or on a depression formed by a surging glacier"?

Lns 52f: better and clearer → "Narama et al. (2018) showed that such short-lived glacial lakes typically exist where the three following geomorphological conditions apply: 1)…, 2)…, 3)…"

Ln 55: maybe more precise if you write "…the existence of a subsurface outlet ice-tunnel."

Ln 56: "…the recent expansion…" do you refer to the increase in number of glacial lakes or the growth of already existing lakes? – I guess rather the former, right? – If so you have to reformulate in order to be clear

Ln 58: "the large variability of glacial lakes" → large variability in terms of what? – Lake type? size? Formation/evolution/drainage? → please clarify

Ln 58: "…was not only related to…" → "…is not only related to…

Ln 58: I would write "… of glacial lakes in the Issyk-Kul basin (Tien Shan) is not only related to the local climate conditions, but also…"

Ln 59: In my opinion it is less misleading if you delete the "regional" here

Ln 59: As you stated this before referring to the three geomorphological conditions for the existence of these short-lived glacial lakes (cf. Narama et al., 2018), I would rephrase the sentence as follows: "…, but also to geomorphological conditions in the glacier forefield as described above (cf. Narama et al., 2018)", I think this is easier to understand and clearer because it's not only about the closure and opening of an outlet ice-tunnel…

Ln 60: "such complex" → "such complexes"

Lns 60f: "Ice degradation within such complexes results in moraine formation"??? – I am not sure whether I agree here, I mean, you write about "ice-cored moraine complexes", which are also already moraine structures in my understanding (the word "moraine" is even already included…), just that they contain ice… I would delete this sentence and instead write what changes in terms of surface dynamics and landform processes when there is no more ice in the morainic material…

Lns 61f: "…were confirmed in the Jeruy Glacier front…" → you mean "…were observed in the forefield of Jeruy Glacier…" → change accordingly

Lns 62f: "…, and such changes likely affect the outlet ice tunnel and formation of the depressions." → ok, can you briefly state/write how? Seems important to me here…

Ln 64: "As changes can occur…" → new section, please specify what changes you're referring to here!

Ln 64: See comment above, I think using the term "flood scale" is not very precise here… do you refer to discharge (rates), range of the flood, or what? → please rephrase in order to be clear

Lns 65f: "…are confirmed in recent years in the northern Tien Shan" → "…have been observed in the northern Tien Shan in recent years, …"

Ln 66: "…difference of…" → "…difference in…" or "…difference between…"

Lns 66f: "flood scale" → see comment just above

Lns 67f: "A lake's fate depends on…" → this is a very general statement and not true as such → you again have to be precise about which lakes you mean, because, for instance, a rock dammed lake doesn't depend on the existence of ice or permafrost when it comes to GLOF's at all…., here again you talk about a very specific glacial lake and dam type!

Ln 68: "Such hazards" → "Hazards from abruptly changing discharge of glacial lakes can.. ."

Ln 69: I think "investigate" is more suited than "predict" here

Ln 70: I would add "…at the Korumdu lake (Teskey Range, Tien Shan, Kyrgyz Republic)…"
Ln 70: "…reason of…" → "…reason for…"

Ln 70: "flood scales" → see comment above

Ln 70: "shot lived lakes" → "short lived lakes"

Ln 71: "These new knowledges are important for glacier disaster mitigation" → "Findings from our study are relevant for glacier-related hazard mitigation."

Ln 72: "The paper is organized as following" → "The paper is organized as follows: …"

Ln 72: "To understand the closure and drainage…" → The lake per se doesn't close! → "To understand the formation and drainage mechanism of…"

Ln 74: "To clear the reason how…" → "To find out how…/To investigate how…"

Ln 75: The outlet ice-tunnel is not "in" the Korumdu lake → rephrase

Ln 75: "…we examined the surface changes around the Korumdu lake in field survey." → "… we surveyed surface elevation changes around Korumdu lake in the field"

Lns 75f: "To clarify how the other short-lived lakes…" → which other short-lived lakes/where → be clear, precise and rephrase accordingly

Lns 76f: "…we investigated the timing of appearance of short-lived lakes for the other lakes in this region were studied in 2015–2019" → "…we investigated their timing of appearance during summer months between 2015 and 2019 using…"

Lns 78f: "…we discussed…" → "…we discuss…"; "…the reason of…" → "…the causes of…"; "…at Korumdu lake including other lakes…" → "…for Korumdu lake and other lakes of the same type in the study area.", then start with a new sentence and rephrase as follows: "We also examine the relationship between outlet tunnel size and lake drainage rate under the influence of increasing air temperature."

***Study area***

Lns 81f: "The study area is in the northern part of the Teskey Range and near the south shoreline of the Issyk-Kul Basin, Kyrgyz Republic (Fig. 1)." → "The study area is situated in the northern part of the Teskey Range south of Lake Issyk-Kul (Fig. 1)."

Lns 82f: "The glacier distribution in the western part of the range (3700–4200 m) is lower than the distribution in the eastern part (3800–4500 m)." → "There are less glaciers in the western part of the Teskey Range (3700–4200 m) than in the eastern part (3800–4500 m)."

Ln 83: "The difference is…" → "This difference is…"

Ln 85: "…of the western part is 255 mm, …" → "…of the western part was 255 mm, …"

Ln 85: "…whereas that at the…" → delete "that"

Ln 85: "…of the central part is 378 mm, …" → "…of the central part it was 378 mm, …"

Lns 85f: "…, and that at the…" → "…, and at the Cong-Ashu station (2788 m) of the eastern part it was 550 mm (Podrezov and Ryskal, 2019; Fig. 1)."

Lns 86f: "Their annual average temperatures are 0.1°C (1961–1988), –6.28°C (1995–2011; Kuzmichenok, 2013), and 0.27°C (1995–2005), respectively." → "Mean annual air temperature was 0.1°C (1961–1988) for Kara Kujur, –6.28°C (1995–2011; Kuzmichenok, 2013) for Tien Shan, and 0.27°C (1995–2005) for Chong-Ashu."

Moreover, unlike for mean annual precipitation, which you compare for the same reference period for the three weather stations (but unfortunately only over 9 years), you compare mean annual air temperatures for three completely different time periods for the three weather stations. In my opinion this makes not much sense (climate change and increasing temperatures!). If possible, you should take (both for mean annual precipitation and air temperature) reference periods of ca. 30 years (statistically significant) and you should compare values of mean annual precipitation and mean annual air temperature over the same reference periods for all weather stations!

Ln 88: "…has been smaller in the western than the eastern part…" → "…was less pronounced in the western than in the eastern part…"

Ln 90: "The glacier-moraine zones…" → This is not a technical term, or at least I haven't heard or read about "glacier-moraine zones"… What do you exactly mean here? Please rephrase and clarify, using correct technical terms

Lns 91f: "…the ice-cored moraine complex (debris landform including ice) at the glacier front developed during the Little Ice Age (Dikih, 1982; Shatravin, 2007)." → "…ice-cored moraine complexes (debris landforms including ice) at the glacier front developed during the Little Ice Age (Dikih, 1982; Shatravin, 2007)."

Ln 93: "recessions" → "retreat"

Lns 93ff: "Four large drainages occurred from short-lived glacial lakes that appeared on the ice-cored moraine complex; specifically, from Kashkasuu (2006), west Zyndan (2008), Jeruy (2013), and Karateke (2014) (Narama et al., 2010, 2018)." → "Four large drainage events occurred from short-lived glacial lakes that formed on ice-cored moraine complexes (Kashkasuu (2006), west Zyndan (2008), Jeruy (2013), and Karateke (2014) (Narama et al., 2010; 2018))."

Lns 96f: "The Korumdu catchment forms the largest tributary in the Tong River Basin." → "The Korumdu catchment gives source to the largest tributary in the Tong River Basin."
Moreover, from your figures I cannot really distinguish Tong River… If you name it (if this is important at all), you should indicate it on the map in Fig. 1…

Ln 97: "The Korumdu glacier occupies an area of 2.35 km$^2$" → can you add in parentheses when the glacier covered 2.35 km$^2$? Does this number come from a glacier inventory? – Source/Reference?

Lns 98f: "In addition, we investigated the timing of appearance for 160 short-lived lakes in this region (Fig. 1)." → see similar comments above, you should be more clear here and write if all these lakes are of the same type as Korumdu and also say how you chose these lakes (data source?, criteria?)

**Methods**
Ln 100: "Method" → "Methods"

Ln 102: I would place "(Fig. 1,2)" after "…at Korumdu lake…"

Ln 103: "Aug" → "August"

Ln 104: "set" → "placed/installed": "water-level" → "water level" (everywhere in the manuscript, be consistent how you write it, see respective comment above); "…and ground levels…" → I guess you mean "at water table level" here? – "ground levels" is not very clear to me…, maybe you can combine that in a short and easily understandable way with information given in the following sentence (Ln 105).
Moreover, I think it's a bit misleading to start the sentence with "We also placed/installed water level data loggers…" because just above you already mention that, and then it's confusing to read "We also placed/installed…" → please rephrase accordingly

Ln 105: "Water-level" → "Water level";
Moreover, technically it is not very clear to me how you did this correction, because – depending on daily weather conditions – atmospheric pressure which you measured and used to correct your water level data varies as well (i.e. is not a constant)… maybe you can add some information oder reference here

Ln 106: "…was also set with an interval of 1 day." → "…was installed as well and took one oblique image of the area per day"

Lns 108f: "Aug" → "August" (change everywhere)

Ln 110: "…with ground control points…" → "… and ground control points…" (because GCPs are logically independent of the Software you use but of course necessary to produce orthoimages)

Lns 110f: "We obtained the GCPs…" → "We collected/surveyed the GCPs…"

Ln 111: "…using the Trimble GeoExplore 6000… → "…using a Trimble GeoExplore 6000"

Lns 111f: "The absolute positions were accurate to 30-40 cm at GCPs positions by post-processing with data from the Kyrgyz GNSS reference station…" → "The absolute positions of GCPs were corrected during post-processing using data from the Kyrgyz GNSS reference station and had an accuracy of 30–40 cm."

Lns 112ff: "We also investigated the surface changes in an ice-cored moraine complex around the lake by comparing DSMs obtained in 2015 and 2016 on ArcGIS 10.5." → "Surface elevation changes of an ice-cored moraine complex surrounding the lake were computed in ArcGIS 10.5 by comparing DSMs from 2015 and 2016."

Ln 115: "…in the summers…" → "…during the summers…"

Ln 116: "…on ArcGIS 10.5" → "…in ArcGIS 10.5"

Ln 116: "The daily volume…" → "The daily lake volume…"; "…(without water), combined with…" → delete the comma here

Lns 116ff: It is not clear to me how daily time-lapse images were processed and combined with DSMs of different summers/years to compute daily lake volumes. Did you somehow orthorectify the (oblique) time-lapse images? This is important in my opinion but not clear at all from your explanations here… → please add/write how you combined DSMs with daily time-lapse images to compute daily lake volumes

Ln 117: You write "(including amount of glacier recession)" → It is clear to me that you can compute glacier surface elevation changes and terminus retreat from DEM/DSM differencing, fair enough, but it is not clear to me how you included (annual, i.e. from summer to summer, i.e. from one UAV/SfM-derived DSM to the next) glacier surface elevation changes into daily lake volume calculations… or do you mean you just looked at glacier surface elevation

changes as well and just wanted to mention that? – This is not 100% clear to me, please rephrase accordingly

Ln 117: "water-level" → "water level"

Ln 118: "…time-lapse camera data based on UAV DSMs" → don't you mean "…time-lapse camera data and/combined with UAV DSMs"?

Lns 118ff: "Using the same method, we also reconstructed…" → again, it's not clear at all in my opinion how satellite images were "combined" with UAV DSMs to reconstruct water levels of Korumdu lake (see respective comment above) → you have to write how you did this
Moreover, what is the benefit of using satellite imagery (of different resolution and quality) to reconstruct lake water levels compared to using your time-lapse camera images? – is it just to compare two or more different data sources to get the same results and compare the latter? – or is it to extend the temporal resolution of you lake water level data? – or does it have something to do with what you write below ("because we visited at the lake on 4 August 2019")? – or is it to investigate whether satellite remote sensing data could (completely) replace in situ time-lapse camera data to calculate lake water levels using DSMs? – interesting, but not really clear to me here → can you add something on that, be clearer on that point?

Ln 120: "…because we visited at the lake on 4 August 2019." → delete "at"

Ln 121: "We also investigated the changes in lake area during 2017–2019 using PlanetScope images." → ok, I guess You just manually digitized lake areas from the satellite imagery and then compared lake areas? – maybe it would be clearer to write this
Moreover, I think it would be really good and more transparent for people who read your paper if you could add a table containing all different sorts and sources of satellite imagery you used, including columns with "resolution", "acquisition date", and information on how the different satellite imagery were used (i.e. for which type of analyses described in your methodology section you used which satellite images)

Ln 122: I think "meteorological" is the right term to use here instead of "climatic"

Ln 123: delete "resolution"

Ln 124: "…were calculated for 2016–2017" → whole year round or only during summer months? → please clarify

Ln 126: "…were identified using satellite images on ArcGIS 10.5" → "were identified in ArcGIS 10.5 using satellite images."

Lns 126ff: "In particular, 91 images from Landsat- 7/Enhanced Thematic Mapper Plus (ETM+, SLC-off) and Landsat-8/OLI, 31 images from Sentinel-2, and 16 images from PlanetScope acquired during 2013–2018. The resolutions of these images are 15 m (pan-sharpened images of Landsat-7/8), 10 m (Sentinel-2), and 3 m (PlanetScope)." → referring to my comment above, I think it would be very good to make a table with all satellite imagery data you used (including columns with "resolution", "acquisition date", and information on how the different satellite imagery were used (i.e. for which type of analyses described in your methodology section you used which satellite images)); because you worked with many different satellite images from different source dates, with different resolution, for various analyses etc. If you do that then you don't have to write all this information in Lns 126ff, but you can refer to the table and can rephrase these lines in a more descriptive manner…

Ln 129f: "As a definition of short-lived lake, we use that in Daiyrov et al. (2018), which is based…" → "We used the definition by Daiyrov et al. (2018) for short-lived lakes, which is based on seasonal changes in lake area over the summer months of each year"

Ln 131: "…that appears or doubles in area" → "…that suddenly appears and/or increases substantially in area" (I would write this like this because these lakes don't necessarily have to double in area, you just want to express that they often increase substantially in area)

Lns 131f: "We counted the number that appeared…" → "We counted the number of lakes that appeared…"

Ln 132: "In addition, the number was tracked…" → "In addition, the number of lakes was tracked…"

Ln 133: "extracted" → "digitized"

**_Results_**
Ln 136: "Areal variations of Korumdu lake" → better: "Changes in area of Korumdu lake" or "Areal variability of Korumdu lake"

Ln 137: "It sits in a basin formed during glacier recession." → better: "It developed in a depression that formed during the retreat of the glacier."

Ln 138: "The basin developed…" → "The lake basin developed…"

Lns 137f: "At the front of the Korumdu glacier lies the Korumdu glacial lake (Fig. 2). It developed in a depression that formed during the retreat of the glacier. The lake basin developed inside an ice-cored moraine complex." → In my opinion these three sentences belong to the "Study area" section and it is not necessary to write that again here at the beginning of the results section…

Ln 138: Why "Although"? – Makes more sense to delete this

Ln 138: "…most of the basin area…" → "…most of the lake basin…"?

Ln 139: "…glacier, based on…" → delete the comma there

Ln 139: "…a basin length…" → "…a lake basin length…"

Ln 140: "…with total area of…" → "…and a total area of the lake basin of…"

Ln 140: "The basin volume…" → "The lake basin volume…"; "…from 264,000 m$^3$ in 2017 to 330,000…" → add "m$^3$" after "330,000"

Ln 142: "basin" → "lake basin" (2x)

Ln 143: "…, but we found an outlet point…" → "…, but we found an outlet point where meltwater from the lake emerges from a subsurface ice-tunnel within the frozen moraine complex which is connected to the lake (Fig. 2)." → Then you can delete the whole next sentence as it's clear enough (i.e. delete "The existence of the outlet shows that lake water flows through an outlet ice-tunnel from the lake.")

Ln 145: "basin" → "lake basin"

Ln 145: "Drainage water was observed at the outlet point in 2015, 2017 and 2019, but not 2016 and 2018." → "Leakage of meltwater was observed at the outlet point in 2015, 2017 and 2019, but not in 2016 and 2018."

Ln 146: "…becoming large on 30 July" → do you mean "…reached its maximum on 30 July…"?

Lns 147f: "…of the changes appears in the images in Fig. 4…" → better "…of changes in lake size is shown with a sequence of PlanetScope satellite images in Fig. 4." (then you can delete "…, which are based on PlanetScope satellite data."

Ln 152: "…and reached its maximum level.." → "…and reached its maximum size…"

Lns 152f: "In contrast, the lake area did not change dynamically in 2016." → ok, but what does that exactly mean? – the lake appeared but did not expand substantially? – not clear to me at all. Moreover, do you know this from field surveys, the on-site time-lapse cameras or satellite data? – not clear, either… → please clarify here

Lns 153f: "Based on Landsat-8/OLI data, we also found that the lake appeared in 2014 (May 5, June 27, and September 10)." → ok, does that mean it appeared (and thus also drained) three times or does that mean that you just found the lake on May 5, June 27, and September 10 2014 on the satellite imagery but do not know what happened in between? – this is not clear here → please rephrase and clarify, see also my comment just below

Lns 154f: "Although these images show rapid drainage, we did not find evidence that the drainage caused flooding during the survey period" → my comment here goes a bit in the same direction as my comment just above: How can you see on three single (and "point in time") satellite images that the lake showed rapid drainage? – in my opinion you cannot! – If you have field evidence or know from other sources that the lake drained ok, but then you have to write this more clearly → please rephrase and clarify
Also, you write "…we did not find evidence that the drainage caused flooding…" → evidence like what? – can you maybe be more concrete/precise here?

Lns 155f: "According to data in Narama et al. (2018), drainage from Korumdu lake is the flood-wave type in the downstream region because the water stream flows on a gentle slope." → ok, can you please connect this sentence a bit better to the precedent one in terms of logic and context?
Also, "…drainage from Korumdu lake is the flood-wave type…" → ok, but from the sentence just above (when you write "…we did not find evidence that the drainage caused flooding…") it seems that the lake can also drain without flooding, so these two statements are a bit contradictory… → can you please rephrase and clarify?

Ln 159: I would delete the first sentence ("Consider the properties of Korumdu lake from 2017 to 2019"), and then write…. (see comment just below)

Lns 159f: … "Figure 6 shows the measured water level, lake area and volume, and inflow-outflow rate of Korumdu lake from 2017 to 2019" ($\rightarrow$ I would not use the term "inflow-outflow discharge" because "inflow is not a discharge" $\rightarrow$ I recommend using "inflow-outflow rate", you would also need to change this accordingly in Fig. 6d)

Lns 160f: "For 2017, we also show the water temperature (Fig. 6a). We also reconstructed the water level data between August 4 and 31 based on 10 satellite images (yellow points in Fig. 6a)." $\rightarrow$ better and clearer to shorten and write everything in one sentence: "For 2017, water temperature data were also recorded (Fig. 6a), and water level data between 4 and 31 August were reconstructed based on 10 satellite images (yellow points in Fig. 6a)."

Lns 161f: "We calculated volume and discharge using the water levels and the UAV DSMs." $\rightarrow$ "Lake volume and discharge were calculated based on the water level data and the UAV DSMs."

Ln 163: "…August 3…" $\rightarrow$ "…3 August…"

Lns 163f: "…, and then vanishes on 19 August (Fig. 6a)." $\rightarrow$ "…, and then the lake is empty on 19 August (Fig. 6a)."

Ln 164: Why "In the first 29 days…"? $\rightarrow$ write "Within 29 days, …"

Ln 165: "The resulting rate of volume increase was 8,070 m$^3$/day." $\rightarrow$ "The resulting rate of lake volume increase was 8,1 m$^3$ per day." (I think the uncertainty in the applied methodology to derive daily lake volumes is too high to write three decimal places $\rightarrow$ round to one decimal place)

Ln 165: "During discharge, 234,000 m$^3$ of water drains in 17 days, …" $\rightarrow$ "During the emptying of the lake, 234,000 m$^3$ of water drain in 17 days, …"

Ln 166: insert comma after "(Fig. 6b)"

Ln 167: "August 3" $\rightarrow$ "3 August"

Lns 167f: In my opinion it's a bit a pitty to only mention the average recorded lake temperature here (referring to Fig. 6a). From your graph one can clearly see that if the water level is low, lake temperature is high, and vice versa, which makes absolutely sense (heating of shallower lake with less water by solar irradiance is stronger than cooling from inflowing ice meltwater) → I think it would be worth it to add some words about the observed variability in recorded lake temperatures (as well as about the reasons for these measured lake temperatures)

Ln 168: Insert "(Fig. 6a)" after "1°C"

Ln 169: "The first, on 25 July, reaches 3.5 m and a volume of 21,000 $m^3$ (Fig. 6a, b)." → "The first peak on 25 July, showing a lake depth of 3.5 m and a volume of 21,000 $m^3$ (Fig. 6a, b)."

Ln 170: "The second, the yearly maximum, on 11 August, reaches 6 m and a volume of 53,000 $m^3$." → "The second on 11 August, with a lake depth of 6 m and a volume of 53,000 $m^3$, which corresponds to the maximum values in 2018."

Lns 170f: "Finally, the third peak, on 17 August, reaches a level of 5 m and a volume of 39,000 $m^3$." → "The third peak occurs on 17 August, showing a lake depth of 5 m and a volume of 39,000 $m^3$."

Ln 172: "Compared to the case in 2017, …" → "Compared to 2018, …"

Lns 172f: "However, like that in 2017, the inflow rate is also intermittent in 2018." → "Similar to 2017, the inflow rate also clearly varies over time in 2018."

Lns 173f: "The three peaks indicate that closure of the tunnel occurred several times during the 1-month period." → "The three peaks in water level, area and volume of Korumdu lake indicate that closure of the ice-tunnel occurred several times during the one-month period."

Ln 175: "In 2019, the water level goes up and down until 22 July, when it rises sharply (Fig. 6a)." → "In 2019, the lake water level rises and falls before 22 July, when it rises sharply (Fig. 6a)."

Lns 175ff: "Then the level has a local maximum on 30–31 July, reaching 5 m and a volume of 53,000 m3, followed by a yearly maximum on 11 August, reaching 6.5 m and 74,000 m3

(Fig. 6 a, b).” → "Then, the water level shows an intermittent maximum around 30 – 31 July, reaching a lake depth of 5 m and a volume of 53,000 m3. 2019 maximum values were recorded on 11 August, with a lake depth of 6.5 m and a corresponding volume of 74,000 m3 (Fig. 6a, b)."

Ln 178: Delete "Over these years, …" and start with "Other differences include…"

Lns 179f: "…small discharge rates…" → "…small lake discharge rates…"

Ln 180: "…, consistent with the lack of reported flooding." → "…, which is consistent with the absence of reported flooding."

Ln 181f: "Concerning fluctuations, according to the water level data for 2017–2019, the level increased with repeated storage-drainage cycles." → clearer: "According to the water level data of 2017–2019, the lake level rose and fell several times, indicating repeated storage-drainage cycles."

Ln 182: "…small increases of water level…" → "…small increases in water level…"; "…with the level…" → "…with the lake level…"

Ln 185: "We observed drainage water at an outlet point…" → "We observed lake water leakage at an outlet point…"

Lns 185f: "The reason we argue is due to the relative elevations." → "We argue that this might be due to the difference in relative elevations between the lake level and the outlet ice-tunnel entrance."

Lns 186f: "The water levels were 3,810 m on 21 Aug 2015, 3,816 m on 6 Aug 2017, and 3,810 m on 4 Aug 2019, all of which are higher than the outlet point at the basin." → "The water levels were at 3,810 m a.s.l on 21 August 2015, 3,816 m a.s.l on 6 August 2017, and 3,810 m a.s.l on 4 August 2019, thus always higher than the outlet ice-tunnel entrance at approximately 3,807.5 m a.s.l."

Lns 187f: "However, we did not observe water drainage in 2016 and 2018 because the water levels were 3,806.5 and 3,807.5 m, respectively (Fig. 8a, c)." → rephrase to avoid repetition of what is written in Ln 185 → "In 2016 and 2018, lake water levels were at 3,806.5 m a.s.l and 3,807.5 m a.s.l, respectively, thus always lower than the outlet ice-tunnel entrance at

approximately 3,807.5 m a.s.l (Fig 8a, c). Therefore, no lake water leakage was observed at the outlet point of the ice-tunnel in 2016 and 2018."

Ln 188f: "These results indicate that…" → following my rephrasing of the paragraph above, you can delete this whole last sentence!

Ln 191: "4.3 Surface changes around Korumdu lake" → more correct to write "4.3 Surface elevation changes around Korumdu lake"

Lns 192f: "Over a period of one year, how does the region near the entrance of the outlet ice-tunnel change? To answer this question, we compared UAV orthoimages with DSM data in 2015 and 2016 (Fig. 9)." → "To investigate annual surface elevation changes near the entrance of the outlet ice-tunnel, we compared UAV-derived orthoimages with DSMs from 2015 and 2016 (Fig. 9)."

Lns 193f: "A comparison of Fig. 9a, b shows debris sliding, with horizontal backwasting of an exposed ice ridge by 7m." → "Debris sliding and horizontal backwasting by 7 m of an exposed ice ridge between 2015 and 2016 appear from the comparison of the orthophotos."

Lns 194f: "The backwasting indicates melting occurred, which is supported by the UAV-derived DSMs in Fig. 9c)." → "The backwasting indicates that melting of debris-covered ice occurred, which is supported by comparing the UAV-derived DSMs from both years (Fig. 9c)."

Lns 195f: "In particular, along the profile (a–a'; Fig. 9b) of the landform between 2015 and 2016, the surface elevation decreases by about 5 m." → "For instance, along a cross-sectional profile (see a–a' in Fig. 9b), the surface elevation decreased by about 5 m (Fig. 9c)."

Lns 196f: "These results indicate that the surface motion and deposition of debris can cause closure of the outlet ice-tunnel during summer." → I don't think that this is really clear here. Do you have real evidence that debris from the surface of melting ice blocked the entrance of the outlet ice-tunnel? What is the diameter of the ice-tunnel at its entrance (i.e. can it easily/quickly be blocked by mobilized sediments?) And how are these sediments transported from the ice margin to the entrance of the outlet-ice tunnel where they cause blocking of the ice-tunnel? How close are the features shown in Fig. 9 to the entrance of the ice-tunnel? Are these processes really directly linked? And what is the grain size distribution

of the debris (I mean to block the tunnel entrance you need to have a mix of finer and coarser material so that lake water doesn't leak through the deposited debris material anymore)? And how does reopening of the tunnel entrance work after closure by debris deposition? I think you could very well be right with what you're saying but this is all not very clear from the information I have now from the text and the figures of the manuscript… → Please clarify, add on this

Ln 198: delete "mountain" or write "…discontinuous mountain permafrost…"; and write "3,100–3,200 m a.s.l"

Ln 199: "…in 2015–2017…" → "…between 2015 and 2017…"

Ln 200: "…in 2015–2019…" → "… during 2015–2019…"

Lns 201f: "…such as that for a supraglacial lake on a debris-covered glacier…" → "…comparable to supraglacial lakes on debris-covered glaciers…"
Moreover, I think that this sentence is kind of a repetition from the paragraph just above in the manuscript (Lns 192-197) → Maybe better to delete the sentence here and include the references in the paragraph above (Lns 192-197), would fit better…

Ln 205: "4.4 Comparison to other short-lived lakes in the area" → "4.4 Comparison to other short-lived glacial lakes of the Teskey Range"

Ln 206: "…had relatively little drainage…" → "…showed relatively little drainage during emptying…"

Ln 206: "…, whereas four other short-lived lakes…" → I would add the names of these lakes in parentheses here

Ln 207: "…caused large drainages…" → "…caused larger drainage…"

Ln 208: "…these appearance times…" → do you mean the (compared to Korumdu lake) earlier appearance times here → rephrase in order to be clear

Ln 209: "…short-lived lakes…" → "…short-lived glacial lakes…"

Lns 209f: You had a look at the months from June to September 2013–2018 to determine the appearance times of other short-lived glacial lakes, ok, but just above you refer to four lakes described by Narama et al. (2010, 2018) appeared from May to June… So why didn't you also consider May 2013–2018?

Lns 211f: "…such short-lived lakes during 2013–2018 (the total includes re-appearances of the same lake in different years) in the study area." → "…such short-lived glacial lakes in the northern Teskey Range during 2013–2018 (the total number of lakes includes reappearances of the same lake in different years)."

Ln 212: "In Fig. 10, we classify these by month of appearance." → "A classification of these lakes by month of appearance is shown in Fig. 10 for the six year period."

Lns 212f: "The appearance months with the most lakes are June, the snow-melt period, and July, the ice-melt period; specifically, 43 lakes in June and 90 in July." → "Most lakes appeared in June (43 lakes), the snow-melt period, and July (90 lakes), the ice-melt period." Moreover, you can also have snow melt earlier or later than June, and ice melt earlier or later than July. Would it make sense to write "…June (43 lakes), the period of maximum snow-melt, and July (90 lakes), the period of maximum ice-melt."?

Ln 214: "…in these two periods…" → "…for these two periods…"

Lns 214f: Would it be right to write "This large variability is related to different meteorological conditions during summer months of 2013–2018."? I think this would be clearer.

Lns 216f: "Concerning re-appearances, 81 lakes appeared only once for 6 years. Of the remaining, 19 appeared twice, 7 appeared 3 times, 2 appeared 4 times, and 2 lakes appeared all 6 years." → "Concerning reappearances, 81 lakes appeared only once during six years. Of the remaining, 19 lakes appeared twice, 7 lakes appeared three times, 2 lakes appeared four times, and 2 lakes appeared every year."

Ln 217: "indicating that tunnel closure occurred with a different month each year." → Does this refer to the two lakes that appeared every year? – not very clearly written, please rephrase

Ln 218: "Short-lived lakes that reappear many years likely have a tunnel condition in which closure occurs easily." → "Short-lived glacial lakes that reappear during many years likely

show geomorphological settings at the drainage tunnel entrance which favor tunnel closure and hence lake formation."

*Discussion*

Ln 221: "5.1 Cause of outlet ice-tunnel closure at Korumdu lake" → "5.1 Causes of outlet ice-tunnel closure at Korumdu lake"

Lns 222f: "In the case of ice tunnel closure, the supraglacial lakes on the debris-covered Inylchek Glacier in April–May are likely to appear due to the closure of englacial conduits when stored water freezes (Narama et al., 2017)." → "In the context of ice-tunnel closure, Narama et al. (2017) report that the supraglacial lakes on the debris-covered Inylcheck Glacier appear in April–May due to the closure of englacial conduits by freezing of stored water."

Ln 228: I would replace "…in the study region…" with "…in the northern Teskey Range…"

Ln 229: "…or by blockage by collapsing with deposition of mixed debris and ice…" → "…or by blockage with depositions of ice-debris mixture after roof collapsing…"

Lns 231f: "The short-lived lakes here that caused the four large drainages (2006, 2008, 2013, and 2014) appeared in May–June and expanded in June–July (Narama et al., 2010, 2018)." → "Four short-lived glacial lakes of the Teskey Range that caused four large drainage events (2006, 2008, 2013, and 2014) appeared between May and June and expanded in area until June–July (Narama et al., 2010; 2018)."

Ln 232: "The timing suggests a closure that…" → "The timing of lake appearance suggests an ice-tunnel closure that…"

Ln 233: Do you mean "We call this the deposition-freezing type of ice-tunnel closure."? → I would rephrase this accordingly…

Ln 234: Do you mean "However, for none of the case studies investigated by Narama et al. (2010, 2018), neither geomorphological behavior of the ice-tunnel nor water level fluctuations were studied in detail." → I would rephrase this accordingly…

Ln 235: "…excluding the case of no expansion…" → "…excluding the case of no lake expansion…"

Ln 236: I would replace "based on water-level of a data logger and time-lapse camera images." by "based on our field surveys." (smoother and the reader knows your survey methods from the parts of the manuscript further above)

Ln 236: "…changes in the basin…" → "…changes in the lake basin…"

Ln 238: "…likely was caused by…" → "…were likely caused by…" (plural because you write about "the blockages")

Ln 239: "Further evidence that Korumdu lake forms by the deposition process comes from consideration of water-level fluctuations." → "Looking at water level fluctuations of Korumdu lake gives further evidence for lake formation by deposition of ice and debris."

Ln 240: "The fluctuations of water level, such as spikes, reveal changes in the tunnel condition (Fig. 6d)." → I'd suggest "The fluctuations of lake water level and discharge spikes reveal changes in the ice-tunnel morphology (Fig. 6d)."

Ln 241: "…the water increase was…" → "…the water level increase was…"

Ln 242f: "…ice tunnel…" → "…ice-tunnel…"

Ln 243: delete "also"

Ln 244: "…to the water pressure…" → "…to changes in water pressure…"; I suggest replacing "…or thermal erosion." with "…or deposition of ice-debris mixture through melting processes."

Lns 245ff: "In 2017, the trend of water volume increase consisted of two parts: 5 to 25 July and 26 July to 3 August (Fig. 6b). The first period had sporadic fluctuations, indicating incomplete closure of the tunnel, but the second period had a smooth increase, indicating complete closure." → "In 2017, there were two periods of varying patterns of lake water volume increase (Fig. 6b). The first period (5 to 25 July) revealed sporadic fluctuations in increasing water volume, indicating incomplete closure of the ice-tunnel. However, the

second period (26 July to 3 August) showed a continuous and rapid increase in water volume, indicating complete closure of the ice-tunnel."

Ln 247: delete "value"

Ln 248: "Longer closure periods…" → "Longer periods of tunnel closure…"; "…larger short-lived lakes…" → "…larger short-lived glacial lakes…"

Ln 249: "Thus, the period of closure might be determined by the condition of the tunnel." → "Thus, the period of closure is likely determined by the morphology of the ice-tunnel."

Lns 250f: "Many of the other short-lived lakes that also appear in the ice-melting period are likely to be the deposition-closure type, for the same reasons we applied to Korumdu lake." → "As for Korumdu lake, many of the other short-lived glacial lakes in the northern Teskey Range which were detected based on satellite imagery are likely to belong to the deposition-closure type as well."

Lns 251f: "For example, in Fig. 12, we show surface changes in the outlet ice-tunnel at the Jeruy glacial lake between 2014 and 2016." → the observed surface changes are rather around the lake or above the ice-outlet channel I think, and not in the ice-tunnel itself → I would rephrase as follows: "For example, Figure 12 shows changes in surface elevation and the outlet ice-tunnel of the Jeruy glacial lake between 2014 and 2016."

Ln 252: I would replace "large" with "distinct"

Ln 253: "…making closure likely" → "…, which likely led to tunnel closure."

Lns 253: "Thus, the surface condition always changes…" → better: "Thus, morphology and surface characteristics of an ice-cored moraine complex within the mountain permafrost zone are prone to frequent changes, and…"

Ln 254: "…and the deposition-closure type is the major type in this region." → clearer to write: "…and the deposition-closure type is likely the main type for drainage tunnel blockage and hence formation of short-lived glacial lakes in the northern Teskey Range."

Lns 254f: "Thus, the appearance of a short-lived glacial lake is inevitable in summer when the melting rate is high." → following my suggestions to rephrase the subsequent sentence, I

would rephrase this one as follows to make things clear: "If deposition-closure processes occur in summer when the melting rate is high, the formation of a short-lived glacial lake is highly likely."

Lns 255f: "The characteristics of this lake disaster might be shown in another Asian mountain permafrost region." → This sentence makes not much sense to me and is quite misleading, as you primarily write about geomorphological processes linked to short-lived glacial lake formation and drainage! You hardly say anything about risks or disasters related to the emptying of the studied lake type! I really think it's better to delete this whole sentence.

Ln 258: "5.2 Relationship between outlet tunnel size and drainage scale" → "5.2 Relationship between outlet tunnel size and lake drainage"

Lns 259f: "…had large drainages." → "…showed considerable drainage."

Ln 261: "…with Jeruy's outlet cross-section being…" → "…with a cross-section measuring 8m$^2$ at Jeruy…"; "…4 x 2 m$^2$…" → was the tunnel width 2 m and the height 4 m? So a cross-section of 8 m$^2$ in area? → clarify

Ln 262: "…and Karateke's about the same or larger (not shown)." → "…and a cross-section of about the same size or larger at Karateke."

Lns 262f: "Earlier, back in 2008, the w-Zyndan lake of 437,000 m$^3$ had a discharge rate of…" → "In 2008, the w-Zyndan lake (437,000 m$^3$) emptied at a discharge rate of…"

Ln 264: "…did not have a large drainage during…" → "…did not show as high drainage rates during…"

Ln 265: "…than those of…" → "…than those at…"

Ln 266: "…of the two large drainages…" → "…of the two large drainage events…"

Ln 267: "…with Korumdu lake…" → "…for Korumdu lake…"

Lns 267f: "…, which is behavior consistent with closure of a small channel caused by deposition." → "…, which was related to closure of the small outlet ice-tunnel caused by deposition of and blockage by debris."

Lns 268f: "…ensured a slower discharge even when it became full (300,000 m$^3$)." → "…resulted in slower lake discharge even when lake volume reached its maximum (300,000 m$^3$)."

Lns 270f: "These results show that the lake size and the dimensions of the outlet ice-tunnel are related to the scale of discharge." → But above you just say that there was no clear relation between lake size and discharge rate for Korumdu lake during 2017-2019! → "These results show that, at least for Korumdu lake, the dimensions of the outlet ice-tunnel were the dominant factor controlling lake discharge rates."

Lns 273f: I would place "…enlarging the outlet ice-tunnel…" before the references here

Ln 274: "…although basin-size changes depend on the particular glacier landforms, the basin area in the case of Korumdu lake has increased each year due to glacier recession." → "…although lake basin-size changes depend on the particular glacial landforms, the basin area of Korumdu lake has increased each year due to glacier retreat."
Moreover, I am not very sure what you exactly mean here… how do which glacial landforms exactly influence lake basin-size? → please clarify

Ln 276: "…in this region,…" → "…in the Teskey Range,…"; "…during their discharge may increase in the future…" → "…during their discharge may become more frequent in the future…"

**Conclusions**
Ln 278: "6 Conclusion" → "6 Conclusions"?!

Ln 279: "Our field survey found…" → "From our field survey we found that…"

Ln 280: delete "Later, …"

Ln 281: "…the draining process was relatively slow…" → "The lake drainage was always relatively slow…"; "outlet ice tunnel" → "outlet ice-tunnel"; "…scale of discharge…" → "…discharge rate…"

Ln 282: "…sizes…" → "…size…"

Lns 282f: "We argued that predicting the scale of a drainage requires knowledge of the outlet ice-tunnel dimensions and the lake's depression size." → "We argue that predicting drainage rates requires knowledge about the dimensions of the outlet ice-tunnel and the size of the lake basin."

Lns 283f: "Our research method of combination between water-level data and UAV DSMs could estimate the discharge and the approximate dimensions of the tunnel." → "By combining water level data and UAV-derived DSMs from consecutive years, we were able to estimate daily lake discharge and approximate dimensions of the outlet tunnel."
I am not sure whether it is clear enough from the manuscript that (and how) you could estimate the dimensions of the outlet ice-tunnel from DSM differencing, i.e. from interpreting surface elevation changes. – In my opinion you rather wrote about ice melt, changing surfaces of the ice-cored moraine, backwasting and debris sliding… I did not read really much about how you estimated approximate dimensions of the outlet tunnel in the results section (4.3). If you write this in the conclusions, you would have to elaborate and talk about that in the results chapter… You cannot bring up "new results" only in the conclusions section! → please add some text in section 4.3 accordingly

Ln 285: "During 2013–2018, satellite data showed this region to have 160 short-lived glacial lakes, …" → "Based on satellite images from 2013–2018, 160 short-lived glacial lakes were detected in the northern Teskey Range, …"

Lns 286f: "Four lakes that appeared a month earlier had large drainages, the only cases of large drainage in the study." → "Four lakes that appeared a month earlier showed drainage rates which were significantly higher compared to the rest of the lakes."

Lns 287f: "Nevertheless, with a warming climate, any short-lived glacial lake might cause large flooding if the outlet ice tunnel and basin size sufficiently enlarge." → better and more specific: "However, with a warming climate resulting in enlarging outlet ice-tunnels and lake basin sizes, also other short-lived glacial lakes of the northern Teskey Range might cause large flood events."

Lns 289f: "The glacial lake outburst floods (GLOFs) which caused by moraine-dam failure such as Himalaya and Andes are minor cases in this region." → "Glacial lake outburst floods (GLOFs) caused by moraine-dam failure, as frequently observed in the Himalayas or the Andes, rather rarely occur in the northern Teskey Range."

Lns 290ff: "Short-lived lakes which caused by closure and opening of an outlet ice-tunnel in moraine complex are a major hazard in this region, because the short-lived lake exists on an ice-cored moraine complex within geomorphological and climate conditions of the mountain permafrost zone." → "Short-lived glacial lakes that form on ice-cored moraine complexes within the mountain permafrost zone through closure and opening of subsurface outlet ice-tunnels are a major hazard in the northern Teskey Range."

Lns 293f: "These new knowledges are useful to understand the phenomena and behavior of the short-lived lakes and consider glacier hazard mitigation in the mountain permafrost regions of Asian high mountains." → "Insights from monitoring short-lived glacial lakes in permafrost zones are useful to better understand their characteristics and behavior, and therefore important for mitigation of glacier-related hazards in high-mountain areas of Central Asia."

Lns 294f: "A threat of the short-lived lakes increases for the residents since 2000s. This hazard case might be major in Asian high mountains in present." → Apart from the fact that those two sentences would have to be rephrased in order to be clear, I would directly delete them because in your study you don't really address risks from glacier-related hazards and you have not shown that such threats have increased for residents since 2000. Hence, this would be something totally new at the end of the paper, and in the conclusions you should not come up with something new!

***Figures***
Figure 1: I would enlarge the figure if possible and also add names of specific glacial lakes or rivers that you mention by name in the manuscript (see also corresponding comments above). Figure Caption: "…located on the south shoreline of Issyk-Kul Lake,…" → "…south of Lake Issyk-Kul,…"; "Red circles are…" → "Red circles indicate locations of…"; "Green squares with checks are…" → "Green squares with checks show short-lived glacial lakes…"; "…caused large drainages…" → "…caused large drainage events…"

Figure 2: I think it would be beneficial for the reader of the paper to show locations of your time-lapse camera, water level measurement logger, water temperature logger, air and ground temperature loggers in Fig. 2 (see respective comments above referring to the methods section). Moreover, in my opinion, this is not really a "geomorphological map" (i.e. not really a map of landforms and processes) → I would just write "Overview of the Korumdu glacier front" in the figure caption instead of "Geomorphological map of the Korumdu glacier front".

Figure 4, figure caption: "…of the Korumdu lake area during 2017–2019." → "…showing the evolution of Korumdu lake in (a) 2017, (b) 2018, and (c) 2019."

Figure 5, figure caption: I would rephrase as follows: "Evolution of Korumdu lake during 2017–2019 based on time-lapse camera images acquired in the field."

Figure 6: I would write "air temperature" for the labeling of the right y-axe in Fig. 6a; Labeling of the y-axes: write "lake volume ($m^3$)" and "lake area ($m^2$)" (Fig. 6b, c); labeling of the x-axes; Fig. 6d: labeling of the y-axe: better write "inflow-outflow rate ($m^3$/s)"? See my comments thereupon further above…
Figure caption: I would delete the first sentence and rephrase the figure caption as follows: "(a) Water level of Korumdu lake in 2017–2019 and air temperature in 2017. (b) Lake volume. (c) Lake surface area. (d) Inflow-outflow rate. These data from 2017 to 2019 are based on water level logger data, UAV DSMs, time-lapse camera images, and PlanetScope satellite images."

Figure 7: You have to add "(c)" and "(d)" to the right images and "(b)" has to be replaced to the second image I guess (otherwise it doesn't make sense with what you write in the figure caption.
Figure caption: I would write "…increase in water level of Korumdu lake."

Figure 8: Better write "maximum lake extension/area" in Fig. 8a instead of "basin line"; I would write "m a.s.l" instead of "m" wherever you refer to "elevation values"
Figure caption: write out the name of the months ("August", "July"); in addition, write "Orthoimages were acquired from UAV surveys."

Figure 9: write "Elevation (m a.s.l)" in the labeling of the y-axis of Fig. 9c;
Figure caption: better write "…of the debris-covered stagnant ice/dead ice at the entrance of…"?; write "…based on UAV orthoimages."; "(a) On 21 August 2015."; delete "line" after "exposed ice edge"; "(b) Same as (a) except on 12 August 2016."; "…show the new positions of the respective surface features after one year."

Figure 10: I would write out the names of the months in the legend of the graph.
Figure caption: I would add "…derived by Landsat-7/8, Sentinel-2, and PlanetScope satellite images." at the end of the sentence!

Figure 11: With the dotted black line you show the level of the lake-dam crest; I would write "level of lake-dam crest" in Fig. 11a instead of just "lake-dam"

Figure caption: write "The two types of ice-tunnel closure occurring in the northern Teskey Range."; better: "(a) Deposition-freezing type of closure in case of an outlet ice-tunnel being blocked by freezing of stored water or deposition of debris and ice."; "(b) Deposition-closure type of closure in case of an outlet ice-tunnel being blocked by deposition of debris and ice by thermal erosion (ice melt)."

Figure 12: I would add information about the width and height of the ice tunnel shown in Fig. 12b (2 x 4 m).

Figure caption: write "…which drained on 15 August 2013."; write "(a) Lake basin of Jeruy glacial lake on 9 August 2014. The white arrow shows the direction of lake drainage. (b) Insight into the outlet ice-tunnel on 9 August 2014. (c) The outlet ice-tunnel area on 9 August 2014. The white circles in (c) and (d) show the same location. (d) Same as (c) except on 9 August 2016."

---

## Referee Comment (RC2) · Anonymous Referee #2 · 8 Dec 2020

Water storage and drainage of short-lived lakes in the Teskey Range, Central Asia

by Mirlan Daiyrov and Chiyuki Narama

General comments:

=============

This is an interesting study about ice tunnels of short-lived lakes in parts of the Tien Shan. A main problem with the paper is that it becomes not very clear over large parts what its focus is: Is it ice tunnels, is it Korumdu lake? Is Korumdu lake an example, or a main focus? Why Korumdu lake? The authors should at the beginning develop and

explain the purpose of the paper, and then relate to this purpose throughout the paper more clearly.

Specific comments:

============

Abstract:

————-

The abstract is unclear. Needs to be rewritten thoroughly. What is the relation of Korumdu lake with respect to the other lakes, not mentioned by name? Only later in the text it becomes clear that the paper is about Korumdu lake.

Introduction:

————

Well written, but I recommend an additional paragraph summarizing the previous findings from a number of papers of the authors about short-lived lakes, and how this paper relates and adds to these previous papers. Is it a new outburst, not covered in the previous papers? Why was it not covered in the previous overview papers. Something special with this lake? What new knowledge is expected compared to the previous papers? Is there a special focus of this study (on ice tunnels?), not covered in the other studies? Etc.

Study area:

————

This section suffers from the lack of clarity in the paper focus. You introduce the study region, not the lake Korumdu, but then you start investigating one specific lake. You need to introduce the region and the specific lake, and make clear why you investigate in detail lake Korumdu. What makes this lake particularly useful or necessary for ice

tunnel investigations in addition to the ones studied earlier?

Results:
———-

L189: too low for drainage? Do you mean the lake did not run over in 2016 and 2018? Where did the melt water from the basin go then?

Discussion:
————

L250: deposition-closure type? This term/type was not introduced before. What is the difference to the deposition-freezing type? L254-256: I don't understand these sentences. "inevitable"? Do you say every moraine complex will lead to a short-lived lake? I don't get the purpose of the last sentence. L270: But what causes what? Does more discharge lead to larger tunnels, or do larger tunnels enable more discharge. I would expect that large discharge melts the tunnel walls and enlarges the tunnel. But you seem to argue that other way round? Further: what influence has the drainage catchment size and the amount of melt water available? Could it be that larger catchments produce more water which then causes larger tunnels?

Conclusions:
—————-

Why don't you summarize your two types of tunnel closures? I think these are important to understand your conclusions of paragraphs 1 and 2. Last paragraph needs rewriting. Especially the conclusions regarding hazards are not well discussed and backed-up in the text.

Technical corrections:
=============

Line 49: "but this relationship was so far little studied with regard to proglacial lakes as of concern in this study." Or something like that. L54: supply "from"? L55: from the "lake" depression L64: As changes "related to short-lived lakes" can occur... L74: clear -> clarify L110: SA) Structure from Motion. Remove "of" L131: does the lake need to "double"??? or just increase in area? L155: lake is of flood-wave type... Do you want to say that the slope is too low that the flood would incorporate debris and become a debris flow?

Fig 5: can you remove the blue tone of the photos by improving the colour balance? Fig 7: there are no panels c and d as indicated in the caption, and I guess b is placed wrong. Fig 11: indicate that dark blue in the tunnel is frozen

—

---

## Author Comment (AC1) · 19 Jan 2021

We thank you for their valuable comments, all of which helped to improve the manuscript. Our response to each comment is below. We showed our replay for reviewer comments.

[General comments] I want to thank the authors for their work and valuable scientific contribution. The study contains (partly) novel and and interesting findings, the presented results seem solid overall and are of interest to the scientific community. I think that this work deserves to get published in Natural Hazards and Earth Syste Sciences. However, in my opinion, the authors still need to put quite some effort into improvement

of their manuscript. As it is, it cannot be published. There are some important major and an extensive number of minor issues which definitely need to be addressed, corrected, clarified, extended and implemented prior to publication. I guess that – even if there are quite a few – the majority of the specific comments listed below are easy to implement, whereas some specific comments will maybe need some additional work and time. I hope that my work will help improving the paper, and I encourage the authors to implement and reply to all my comments as far as possible. Thanks a lot, all the best and kind regards.

[Response] We thanks a lot for your careful reading and valuable comments! We appreciate your efforts.

[List of some general comments]:

[Comment 1] In general, linguistic and content clarity and correctness need to be substantially improved (see my introduction to the specific comments below)

[Response] Thanks for your careful reading and comments. We attempted to improve the contents of the manuscript based on comments.

[Comment 2] For some parts, the manuscript also needs major improvement in terms of content (see various specific comments thereupon below). For instance, the conclusions have rather "discussion character". You need to add an introductory paragraph to the conclusions, elaborating the research questions and what you did in order to answer them (including a short summary about your study area and methods). Then you can summarize your results and contextualize the meanings of your findings

[Response] Thanks for your suggestion. We revised our manuscript and have added introductory paragraph to the conclusions.

[Comment 3] You have used so many different satellite imagery data from various sources and for many different things. I think for the reader of your paper it would be beneficial to add a table containing sources or platforms (i. e. Landsat, Sentinel-2

etc.) of the satellite images used, also listing source dates, resolution and purpose (i. e. which satellite imagery you used for what). You could then refer to this table in the manuscript wherever needed instead of having to write long and rather choppy sentences

[Response] We added "Table 1".

[Comment 4] Is there a reason why Korumdu lake was chosen for your extensive fieldwork?

[Response] We added these reasons in the Study area. "As the reason why this lake was selected as a research site, (i) the lake is a short-lived type which appears every year, (ii) it is easy to access the field, and (iii) this lake is located at the Tong region where four large outburst floods occurred in the past. "

[Comment 5] In 2018, there was no water leakage at the outlet point close to Korumdu lake, but the lake emptied anyway. Do you know how, through which processes, and why? – Would have been interesting to add in the discussion

[Response] Lake was empty at our visiting in 2018, but lake appeared in this year as shown by lake level data and time-lapse camera images (Figs. 4-6).

[Comment 6] In my opinion, it would have been interesting to know why, in the Teskey Range, there are "so many" short-lived glacial lakes... is this mainly due to the presence of permafrost (ice-cored moraines)? It would have been interesting to elaborate "a list of geomorphological prerequisites or conditions" for the formation of the type of short-lived glacial lakes occurring in the northern Teskey Range (i. e. presence of sufficient debris, location in permafrost zone, retreating (debris-covered) glacier etc.). If possible, it would have been interesting to compare the spatial density or number of short-lived glacial lakes in the northern Teskey Range to other high-mountain areas...

[Response] In previous study (Daiyrov et al., 2018), we found many ice-cored moraine complexes in the Teskey Range, because this region is mountain permafrost zone.

Short-lived lakes appeared at depressions on ice-cored moraine complex due to ice melting.

[Specific comments and technical corrections] To facilitate the author's correction of the manuscript I combined specific comments and technical corrections (including language and comprehensibility issues). Sometimes comments contain both specific comments and technical corrections, sometimes just one or the other. I took a lot of time to rephrase beforehand rather unclear text passages and tried to improve a lot of the manuscript. I hope that helps and that the authors appreciate my efforts in this regard. I would ask the authors to implement my comments and suggestions as far as possible.

[Response] We thanks a lot. We also highly appreciate your efforts and time that took a lot for rephrasing our manuscript. Your comments and improvements are very helpful! Thanks again!

Title [Comment 1] I think it would be more attractive, preciser and more transparent to change the title of the manuscript into "Formation, evolution and drainage of short-lived glacial lakes in permafrost environments of the northern Teskey Range, Central Asia".

[Response] We changed the title.

Abstract [Comment 2] Ln 11f: Not clear to me if you refer to short-lived lakes in general here or if this sentence already refers to processes of drainage of the short-lived Korumdu lake. If the former applies, you have to write "Short-lived lakes grow rapidly and drain within a few months, due to..."; if you already refer to Korumdu lake here you would have to state this, e.g. "Korumdu lake, a short-lived glacier lake in the Teskey range surveyed in detail for this study, grows rapidly and drains within a few months, due to..."

[Response] We changed it.

[Comment 3] Ln 12: "... in moraine complex..." → "...in a moraine complex at the

glacier front."

[Response] We changed it.

[Comment 4] Lns 12f: This is a general statement → you have to write "Outburst floods of this lake type are a major hazard in this region and differ from many cases..."

[Response] We changed it.

[Comment 5] Lns 14f: This sentence starts with a statement on the drainage of short-lived glacier lakes in general, i.e. it's not clear if you then write "..., we examined its recent changes..." → be precise and write "..., we examined recent changes of Korumdu lake in water level, area, volume and discharge...", or if you refer to the whole sample of short-lived glacier lakes you analysed in the Teskey Range you have to write "..., we examined recent changes in water level, area, volume and discharge of short-lived glacier lakes in the Teskey Range with a field survey and satellite data analysis."

[Response] We changed it.

[Comment 6] Ln 16: "... during all summers during..."→ "... during all summers between 2014 and 2019 except in 2016."

[Response] We changed it.

[Comment 7] Ln 14 vs. Ln 16: "water level" vs. "water-level" → be consistent and write "water level" everywhere in the manuscript

Response] We changed all.

[Comment 8] Lns 16f: How this sentence is written one would think that the sudden appearance and expansion of Korumdu lake took only place 2017-2019, not before, correct? Below in the methods section you write that you carried out field surveys from 2015 to 2019. So, during summers 2015 and 2016 there was no sudden appearance and expansion (and drainage) of Korumdu lake? I think it would be good to clarify/be

more precise here.

Response] We changed it. This is for only 2016. "During summer 2016 there was no sudden appearance and expansion (and drainage) of Korumdu lake."

[Comment 9] Ln 17: "The timing..." which timing? → be precise in order to be clear "The timing of lake appearance/lake formation indicates..."

[Response] We changed it.

[Comment 10] Ln 18: "drain" → "drainage"

Response] We changed it.

[Comment 11] Ln 19: Here, the abbreviations "UAV" and "DSM" appear for the first time. I would write them out and put the abbreviations in parentheses, unless it's ok to do that in the abstract (but then you would have to write the terms out when they appear for the first time below in the manuscript. Moreover, a lake is per definition made of water in my opinion you can just write "..., the lake's volume reached..."

[Response] We added the abbreviations of UAV and DSM in Abstract.

[Comment 12] Ln 22: "...that caused large drainages..." → "...that showed larger drainage rates..."

[Response] We changed it.

[Comment 13] Lns 22f: This is a general statement → write "As a result, the dimensions of outlet ice-tunnels of short-lived glacial lakes...". Moreover (for the entire manuscript), in my opinion "flooding scale" is a somewhat misleading and not very precise term... here you refer to discharge (rates) and of course smaller discharge over longer time means less hazard potential for flooding downstream... Be preciser/clarify!

[Response] We changed it.

[Comment 14] Ln 24: "basin volumes" is again a bit misleading, I would just write "...

both tunnel size and lake volume. . ."

[Response] We changed it.

[Comment 15] Ln 24: I would replace "greater hazard" with "increased hazard potential"

[Response] We changed it.

[Comment 16] Lns 24f: I thinks it's more clear to write "In addition to our field surveys of Korumdu lake, we investigated. . .

[Response] We changed it.

[Comment 17] Ln 25: It's clearer to replace "in this region" with "in the Teskey Range"

[Response] We changed it.

[Comment 18] Ln 27: This is a general statement → write "The appearance of short-lived glacial lakes is inevitable in summer when the melting rate is high." Moreover, I think this general statement is not clear enough yet! For which region, for which glaciers, for which type and composition of forefield characteristics does this statement/sentence apply? In my opinion you need to clarify this, otherwise you cannot just leave this as a general statement which could be interpreted as being "valid everywhere". . .

[Response] We improved it. "In the Teskey Range, the appearance of short-lived glacial lakes on the moraine complexes at glacier fronts is inevitable in summer when the melting rate is high."

[Comment 19] Lns 27f: This sentence is not yet clear enough to me. Do you mean something like "Similar characteristics of short-lived lake formation and drainage through blockage and opening of subsurface channels might also be found in other mountain regions of (Central) Asia"? Moreover, this is "just a guess" and I think as a concluding sentence of your abstract I would try to be a bit preciser and clearer. You also mention the term "permafrost" for the first time here. Of course this is important for

the short-lived lakes you're looking at (in terms of the frozen material, i.e. moraine complexes in the glacier forefield). So again, try "not just to guess" but try to say why and where in high-mountain (Central) Asia you would expect to find similar processes, lake types and characteristics of formation and drainage! Seems important and interesting to me...

[Response] We improved it. "Similar characteristics of short-lived lake formation and drainage through blockage and opening of subsurface channels might also be found in other mountain regions of Central Asia such as the Tien Shan and Pamir mountains. These mountain regions have many ice-cored moraine complexes in mountain permafrost zones."

Introduction [Comment 20] Lns 32f: clearer to write "..., rather small glacial lakes can be found close to the present termini of glaciers in the northern Tien Shan (Central Asia) (References)."

[Response] We changed it.

[Comment 21] Lns 33f: I would write "...often produce hazardous debris flows and floods." Like this, the link/logical connection to the subsequent sentence is much better...

[Response] We changed it.

[Comment 22] Ln 35: I would insert commas before and after "including casualties" or put "including casualties" in parentheses for better/smoother readability

[Response] We changed it.

[Comment 23] Ln 36: Reference "Daiyrov et al., under review" → Please do only cite research articles that are either already accepted for publication or already published!

[Response] We changed it.

[Comment 24] Ln 36: "Such short-lived glacial lakes...": Here you mention "short-lived"

for the first time in the introduction. Above it's just about small glacial lakes. So I guess one could assume that all small glacial lakes are short-lived, which of course is not true. In order to be a bit clearer I would therefore add a sentence or a subordinate clause here to precisely state that some/a certain number of these small lakes are "short-lived" or "unstable" (or if you are sure about the percentage of small glacial lakes in the Tien Shan that are "short-lived" you can write "many/the majority of small lakes are short-lived"...), then you can add that "they grow rapidly and drain within a few months"

[Response] We changed it. "A certain number of these small lakes are "short-lived" or "unstable" that they grow rapidly and drain within a few months (Narama et al., 2010, 2018; Daiyrov et al., 2018)."

[Comment 25] Lns 38f: "Such lakes drain through..." here you refer again to small and short-lived proglacial lakes dammed by ice-cored moraine complex, ok, but I think the clarity of the introduction (and the manuscript in general) would benefit from explaining your focus a bit in more detail (including further references), i.e. explaining in more detail which types of short-lived glacial lakes you're looking at/focusing on. – I mean, supraglacial lakes are often also "short-lived", i.e. appear and disappear over the summer season, but processes are different from short-lived lakes in the glacier forefield where frozen moraine material plays a role as well... Not all small glacier lakes in the glacier forefield are short-lived and not all short-lived glacier lakes show the same formation and drainage processes as the ones you're specifically focusing on... I would appreciate if you could elaborate these issues a bit more clear, precise and complete/holistic in the introduction (this would for instance also include an explanation which nonstationary lakes are short-lived and which have a longer lifetime and why (cf. Lns 39f). Partly this is already done further below in the introduction... maybe you could rearrange the different sections of the introduction a bit in order to avoid confusion about what you mean by "small lakes" or "short-lived lakes"...

[Response] We improved it. "A certain number of these small lakes are "short-lived" or "unstable" that they grow rapidly and drain within a few months (Narama et al., 2010,

2018; Daiyrov et al., 2018). These short-lived lakes appear on depressions of ice-cored moraine complexes at glacier fronts. Such lakes drain through an outlet ice-tunnel (subsurface channel) within an ice-cored moraine complex (Popov, 1987; Narama et al., 2010; 2018), and are also called nonstationary lakes (Erokhin et al., 2017), though this term also includes lakes with a long lifetime. Among nonstationary glacial lakes, a short-lived glacial lake fill periodically and quickly within one year (Erokhin et al., 2017). However, some of short-lived glacial lakes have a longer lifetime which develop within 2-3 years until drain."

[Comment 26] Lns 40f: "A short-lived lake can be a severe hazard...", next sentence "The short-lived lakes are a major hazard..." → somehow a repetition, I think you could easily merge these two statements to avoid this

[Response] We improved our revised text. "A short-lived lake can be a severe hazard for local residents in northern Tien Shan because it appears suddenly yet can cause large debris flows."

[Comment 27] Lns 40f: About the use of the term "hazard" or "natural hazard". Please be aware that there is a difference between "hazard" and "risk"! As far as I understand these terms, "hazards" or "natural hazards" are just geomorphological processes which take place (naturally) and which can potentially be dangerous for people, infrastructure and goods. Important factors defining "hazards" are, for instance, duration, intensity/magnitude, spatial extent and return period of the respective geomorphological processes. "Risk" refers to the combination of the probability of occurrence and the damage potential of an event (whereas damage potential is a function of exposition and vulnerability). I was not always quite sure if you actually refer to "hazard" or "risk" whenever you use the term "hazard" in your manuscript. Please check if you use these terms correctly everywhere.

[Response] We do not focus on risk and damage, we focused on understanding of hazard as glacial lake in this paper. We checked all.

[Comment 28] Ln 41: If you write "The short-lived lakes are a major hazard in this region...", this means "all short-lived lakes in the area" are concerned. I would just write "Short-lived lakes are a major hazard..."Moreover I would be precise again and replace "in this region" with "in the Teskey Range" or with "in northern Tien Shan".

[Response] We already improved in [Comment 26].

[Comment 29] Lns 41f: "..., and differs from the outburst which caused by..." → sentence is not fully correct (English). I don't know if you refer to the differences in characteristics of lake drainage (i.e. processes) here or if you refer to the different damage potential/risk of the different lake types and outburst mechanisms here... Please clarify, write more precisely what you mean

[Response] We improved text. "The outburst of short-lived type in northern Tien Shan is differs from the outburst (damage potential and outburst mechanism) which caused by moraine-dam failure in Himalaya and Andes (Costa and Schuster, 1988; Richardson and Reynolds, 2000; Shreshta 2010; Emmer and Cochachin, 2013; Neupane et al.; 2019). For example, a mass-movement triggers are the main factors for dam failures of the glacial lakes in the Himalayas and Andes (Emmer and Cochachin, 2013; Neupane et al.; 2019)."

[Comment 30] Ln 44: "As such glacial lakes" → which glacial lakes? I know which ones you mean but I think it's important here to precisely write which types of glacial lakes you're talking about ( → "small and short-lived proglacial lakes which are dammed by (partially) frozen moraine material/sediments" would be clear, wouldn't it?)

[Response] We changed it.

[Comment 31] Ln 44: "Small and short-lived proglacial lakes which are dammed by (partially) frozen moraine material/sediments drain through a subsurface outlet ice-tunnel. These lakes can expand rapidly when the outlet ice-tunnel...." I think it's better and clearer like this

[Response] We changed it in [Comment 31].

[Comment 32] Ln 46: "Some short-lived glacial lakes" → clearer if you write "Some of these/the aforementioned short-lived glacial lakes..."

[Response] We changed it.

[Comment 33] Lns 46f: "..., which is behavior they share with supraglacial lakes on a debris-covered glacier" → not only the case for supraglacial lakes on debris-coverd glaciers. See for instance "Gornersee" or "Lac de Faverges" on the (quasi debris-free) "Gorner-/Grenzgletscher" or "Glacier de la Plaine Morte" in the Swiss Alps in recent years... (e.g. Huss et al. 2007 in Journal of Glaciology (doi: 10.3189/172756507782202784) or Huss et al. 2013 in Geographica Helvetica (doi:10.5194/gh-68-227-2013) I would just delete the "on a debris-covered glacier"

[Response] We changed it

[Comment 34] Lns 47ff: "Several studies have examined the relationship between supraglacial lakes and englacial conduit on a debris-covered glacier (Benn et al., 2000, 2017; Miles et al., 2016; Watson et al., 2016; Narama et al., 2017), but this relationship has seen little study for glacial lakes." → To me it's not clear what you want to say with this sentence... Please rephrase the sentence in order to be clear

[Response] We improved the text. "Several studies reported the formation and drainage supraglacial lakes are related to connectivity of englacial conduits on a debris-covered glacier (Benn et al., 2000, 2017; Miles et al., 2016; Watson et al., 2016; Narama et al., 2017). However, the variations of the short-lived glacial lakes are derived from open and closure in ice-tunnel inside of the ice-cored moraine complex at glacier fronts."

[Comment 35] Ln 50: "Short-lived glacial lakes" → again, it has to be clear which types of short-lived glacial lakes you're talking about... You could refer to the definition above (it's about "Small and short-lived proglacial lakes which are dammed by (partially) frozen moraine material/sediments"), maybe you could introduce an abbreviation for the types of short-lived glacial lakes you're investigating and use this abbreviation after having precisely introduced it in order to facilitate smooth reading and avoid misunderstandings, increase clarity. . .

[Response] We already defined short-lived glacial lake in [Comment 34].

[Comment 36] Ln 50: delete "either"

[Response] We deleted it.

[Comment 37] Ln 51: better ". . . or on a depression formed by a surging glacier"?

[Response] In this case, lake appeared on a surging glacier. We did not change it.

[Comment 38] Lns 52f: better and clearer → "Narama et al. (2018) showed that such short-lived glacial lakes typically exist where the three following geomorphological conditions apply: 1). . ., 2). . ., 3). . ."

[Response] We chanted it.

[Comment 39] Ln 55: maybe more precise if you write ". . .the existence of a subsurface outlet ice-tunnel."

[Response] We added one sentence. "(lake water is discharged through ice tunnel inside of moraine complex)."

[Comment 40] Ln 56: ". . .the recent expansion. . ." do you refer to the increase in number of glacial lakes or the growth of already existing lakes? – I guess rather the former, right? – If so you have to reformulate in order to be clear

[Response] We improved text. "the recent increase in number and area of glacial lakes . . ."

[Comment 41] Ln 58: "the large variability of glacial lakes" → large variability in terms of what? – Lake type? size? Formation/evolution/drainage? → please clarify

[Response] We improved text. "In addition, Daiyrov et al. (2018) showed that the large variability in the number and distribution of glacial lake types in the Issyk-Kul Basin (northern Tien Shan) is not..."

[Comment 42] Ln 58: "...was not only related to..."→ "...is not only related to...

[Response] We changed it.

[Comment 43] Ln 58: I would write "... of glacial lakes in the Issyk-Kul basin (Tien Shan) is not only related to the local climate conditions, but also..."

[Response] We changed it.

[Comment 44] Ln 59: In my opinion it is less misleading if you delete the "regional" here

[Response] We changed it.

[Comment 45] Ln 59: As you stated this before referring to the three geomorphological conditions for the existence of these short-lived glacial lakes (cf. Narama et al., 2018), I would rephrase the sentence as follows: "..., but also to geomorphological conditions in the glacier forefield as described above (cf. Narama et al., 2018)", I think this is easier to understand and clearer because it's not only about the closure and opening of an outlet ice-tunnel...

[Response] We changed it.

[Comment 46] Ln 60: "such complex" → "such complexes"

[Response] We changed it.

[Comment 47] Lns 60f: "Ice degradation within such complexes results in moraine formation"??? – I am not sure whether I agree here, I mean, you write about "ice-cored moraine complexes", which are also already moraine structures in my understanding (the word "moraine" is even already included...), just that they contain ice... I would

delete this sentence and instead write what changes in terms of surface dynamics and landform processes when there is no more ice in the morainic material...

[Response] We deleted it, and we added one sentence. "Melting of ground ice in moraine complexes results to the development of various landforms (Kääb and Haeberli, 2001)."

[Comment 48] Lns 61f: "...were confirmed in the Jeruy Glacier front..." you mean "...were observed in the forefield of Jeruy Glacier..." change accordingly

[Response] We changed it.

[Comment 49] Lns 62f: "..., and such changes likely affect the outlet ice tunnel and formation of the depressions." ok, can you briefly state/write how? Seems important to me here...

[Response] To clarify this point we have provided additional information in the revised MS. "Surface changes on an ice-cored moraine complex were observed in the forefield of Jeruy Glacier front between 1979 and 2016 (Daiyrov et al., 2018), and such changes likely affect the expansion of the outlet ice tunnel and formation of the depressions."

[Comment 50] Ln 64: "As changes can occur..." new section, please specify what changes you're referring to here!

[Response] We improved it. "As changes of glacial lakes can occur over very large areas and volumes in a short period of time, drainage features and flood discharge (rates) are extremely unpredictable (Erokhin et al., 2017)."

[Comment 51] Ln 64: See comment above, I think using the term "flood scale" is not very precise here... do you refer to discharge (rates), range of the flood, or what? please rephrase in order to be clear

[Response] We changed all.

[Comment 52] Lns 65f: "...are confirmed in recent years in the northern Tien Shan" →

"...have been observed in the northern Tien Shan in recent years, ..."

[Response] We changed it.

[Comment 53] Ln 66: "...difference of..."→"...difference in..." or "...difference between..."

[Response] We changed it.

[Comment 54] Lns 66f: "flood scale" see comment just above

[Response] We changed all.

[Comment 55] Lns 67f: "A lake's fate depends on..." this is a very general statement and not true as such you again have to be precise about which lakes you mean, because, for instance, a rock dammed lake doesn't depend on the existence of ice or permafrost when it comes to GLOF's at all...., here again you talk about a very specific glacial lake and dam type!

[Response] We improved it. "A short-lived lake's fate depends on whether the dam contains ice (Mergili et al., 2013),"

[Comment 56] Ln 68: "Such hazards" → "Hazards from abruptly changing discharge of glacial lakes can.. ."

[Response] We changed it.

[Comment 57] Ln 69: I think "investigate" is more suited than "predict" here

[Response] We changed it.

[Comment 58] Ln 70: I would add "...at the Korumdu lake (Teskey Range, Tien Shan, Kyrgyz Republic)..." Ln 70: "...reason of..."→ "...reason for..." Ln 70: "flood scales" see comment above Ln 70: "shot lived lakes" → "short lived lakes"

[Response] We changed all.

[Comment 59] Ln 71: "These new knowledges are important for glacier disaster mitigation" → "Findings from our study are relevant for glacier-related hazard mitigation."

[Response] We changed it.

[Comment 60] Ln 72: "The paper is organized as following" → "The paper is organized as follows: ..." Ln 72: "To understand the closure and drainage..."→ The lake per se doesn't close! → "To understand the formation and drainage mechanism of..."

[Response] We changed all.

[Comment 61] Ln 74: "To clear the reason how..." → "To find out how.../To investigate how..."

[Response] We changed it.

[Comment 62] Ln 75: The outlet ice-tunnel is not "in" the Korumdu lake →rephrase

[Response] We improved it."

[Comment 63] Ln 75: "...we examined the surface changes around the Korumdu lake in field survey." → "... we surveyed surface elevation changes around Korumdu lake in the field"

[Response] We changed it.

[Comment 64] Lns 75f: "To clarify how the other short-lived lakes..."→ which other short-lived lakes/where be clear, precise and rephrase accordingly

[Response] We added the location as "the Teskey Range".

[Comment 65] Lns 76f: "...we investigated the timing of appearance of short-lived lakes for the other lakes in this region were studied in 2015–2019" → "...we investigated their timing of appearance during summer months between 2015 and 2019 using..."

[Response] We changed it.

[Figure]

[Comment 66] Lns 78f: "...we discussed..."→ "...we discuss..."; "...the reason of..."→ "...the causes of..."; "...at Korumdu lake including other lakes..."→ "...for Korumdu lake and other lakes of the same type in the study area.", then start with a new sentence and rephrase as follows: "We also examine the relationship between outlet tunnel size and lake drainage rate under the influence of increasing air temperature."

[Response] We changed all.

[Study area] [Comment 67] Lns 81f: "The study area is in the northern part of the Teskey Range and near the south shoreline of the Issyk-Kul Basin, Kyrgyz Republic (Fig. 1)." → "The study area is situated in the northern part of the Teskey Range south of Lake Issyk-Kul (Fig. 1)."

[Response] We changed it.

[Comment 68] Lns 82f: "The glacier distribution in the western part of the range (3700–4200 m) is lower than the distribution in the eastern part (3800–4500 m)." → "There are less glaciers in the western part of the Teskey Range (3700–4200 m) than in the eastern part (3800–4500 m)."

[Response] Original is correct. We did not change it.

[Comment 69] Ln 83: "The difference is..." → "This difference is..." Ln 85: "...of the western part is 255 mm, ..."→ "...of the western part was 255 mm, ..." Ln 85: "...whereas that at the..." delete "that" Ln 85: "...of the central part is 378 mm, ..." → "...of the central part it was 378 mm, ..." Lns 85f: "..., and that at the..."→ "..., and at the Cong-Ashu station (2788 m) of the eastern part it was 550 mm (Podrezov and Ryskal, 2019; Fig. 1)."

[Response] We changed all.

[Comment 70] Lns 86f: "Their annual average temperatures are 0.1°C (1961–1988), –6.28°C (1995–2011; Kuzmichenok, 2013), and 0.27°C (1995–2005), respectively." →

"Mean annual air temperature was 0.1°C (1961–1988) for Kara Kujur, –6.28°C (1995–2011; Kuzmichenok, 2013) for Tien Shan, and 0.27°C (1995–2005) for Chong-Ashu." Moreover, unlike for mean annual precipitation, which you compare for the same reference period for the three weather stations (but unfortunately only over 9 years), you compare mean annual air temperatures for three completely different time periods for the three weather stations. In my opinion this makes not much sense (climate change and increasing temperatures!). If possible, you should take (both for mean annual precipitation and air temperature) reference periods of ca. 30 years (statistically significant) and you should compare values of mean annual precipitation and mean annual air temperature over the same reference periods for all weather stations!

[Response] We don't have data of 30 years. Absolutely your suggestion is correct. However, here is just basic information.

[Comment 71] Ln 88: "...has been smaller in the western than the eastern part..."→ "...was less pronounced in the western than in the eastern part..."

[Response] We changed it.

[Comment 72] Ln 90: "The glacier-moraine zones..." → This is not a technical term, or at least I haven't heard or read about "glacier-moraine zones"... What do you exactly mean here? Please rephrase and clarify, using correct technical terms

[Response] We changed to "moraine complexes".

[Comment 73] Lns 91f: "...the ice-cored moraine complex (debris landform including ice) at the glacier front developed during the Little Ice Age (Dikih, 1982; Shatravin, 2007)." → "...ice-cored moraine complexes (debris landforms including ice) at the glacier front developed during the Little Ice Age (Dikih, 1982; Shatravin, 2007)."

[Response] We changed it.

[Comment 74] Ln 93: "recessions" → "retreat"

[Response] We changed it.

[Comment 75] Lns 93ff: "Four large drainages occurred from short-lived glacial lakes that appeared on the ice-cored moraine complex; specifically, from Kashkasuu (2006), west Zyndan (2008), Jeruy (2013), and Karateke (2014) (Narama et al., 2010, 2018)." → "Four large drainage events occurred from short-lived glacial lakes that formed on ice-cored moraine complexes (Kashkasuu (2006), west Zyndan (2008), Jeruy (2013), and Karateke (2014) (Narama et al., 2010; 2018))."

[Response] We changed it.

[Comment 76] Lns 96f: "The Korumdu catchment forms the largest tributary in the Tong River Basin." → "The Korumdu catchment gives source to the largest tributary in the Tong River Basin." Moreover, from your figures I cannot really distinguish Tong River... If you name it (if this is important at all), you should indicate it on the map in Fig. 1...

[Response] We changed it and added "Tong River" in Fig. 1.

[Comment 77] Ln 97: "The Korumdu glacier occupies an area of 2.35 km2" → can you add in parentheses when the glacier covered 2.35 km2? Does this number come from a glacier inventory? – Source/Reference?

[Response] This data is our original. We added "based on Sentinel-2 satellite image in 2019."

[Comment 78] Lns 98f: "In addition, we investigated the timing of appearance for 160 short-lived lakes in this region (Fig. 1)." → see similar comments above, you should be more clear here and write if all these lakes are of the same type as Korumdu and also say how you chose these lakes (data source?, criteria?)

[Response] We improved it. "In addition, we investigated the timing of appearance for 160 short-lived lakes in the Teskey Range during 2013–2018 (Fig. 1) using Landsat-7/8, Sentinel-2, and PlanetScope satellite images.

Methods [Comment 79] Ln 100: "Method" → "Methods"

[Response] We changed it.

[Comment 80] Ln 102: I would place "(Fig. 1,2)" after "...at Korumdu lake..."

[Response] We changed it.

[Comment 81] Ln 103: "Aug" → "August"

[Response] We changed all.

[Comment 82] Ln 104: "set" → "placed/installed": "water-level" → "water level" (everywhere in the manuscript, be consistent how you write it, see respective comment above); "...and ground levels..."→ I guess you mean "at water table level" here? – "ground levels" is not very clear to me..., maybe you can combine that in a short and easily understandable way with information given in the following sentence (Ln 105). Moreover, I think it's a bit misleading to start the sentence with "We also placed/installed water level data loggers..." because just above you already mention that, and then it's confusing to read "We also placed/installed..."→ please rephrase accordingly

[Response] We improved the sentence. "ground surface on moraine (atmospheric pressure)"

[Comment 83] Ln 105: "Water-level" "Water level"; Moreover, technically it is not very clear to me how you did this correction, because – depending on daily weather conditions – atmospheric pressure which you measured and used to correct your water level data varies as well (i.e. is not a constant)... maybe you can add some information oder reference here

[Response] We changed it.

[Comment 84] Ln 106: "...was also set with an interval of 1 day." → "...was installed as well and took one oblique image of the area per day"

[Figure]

[Response] We changed it.

[Comment 85] Lns 108f: "Aug" → "August" (change everywhere)

[Response] We changed all.

[Comment 86] Ln 110: "…with ground control points…"→ "… and ground control points…" (because GCPs are logically independent of the Software you use but of course necessary to produce orthoimages) Lns 110f: "We obtained the GCPs…"→ "We collected/surveyed the GCPs…"

[Response] We changed it.

[Comment 87] Ln 111: "…using the Trimble GeoExplore 6000…→ "…using a Trimble GeoExplore 6000" Lns 111f: "The absolute positions were accurate to 30-40 cm at GCPs positions by post-processing with data from the Kyrgyz GNSS reference station…" → "The absolute positions of GCPs were corrected during post-processing using data from the Kyrgyz GNSS reference station and had an accuracy of 30–40 cm." Lns 112ff: "We also investigated the surface changes in an ice-cored moraine complex around the lake by comparing DSMs obtained in 2015 and 2016 on ArcGIS 10.5." → "Surface elevation changes of an ice-cored moraine complex surrounding the lake were computed in ArcGIS 10.5 by comparing DSMs from 2015 and 2016." Ln 115: "…in the summers…" → "…during the summers…" Ln 116: "…on ArcGIS 10.5" → "…in ArcGIS 10.5"

[Response] We changed all.

[Comment 88] Ln 116: "The daily volume…" → "The daily lake volume…"; "…(without water), combined with…" → delete the comma here

[Response] We changed it.

[Comment 89] Lns 116ff: It is not clear to me how daily time-lapse images were processed and combined with DSMs of different summers/years to compute daily lake

volumes. Did you somehow orthorectify the (oblique) time-lapse images? This is important in my opinion but not clear at all from your explanations here... → please add/write how you combined DSMs with daily time-lapse images to compute daily lake volumes

[Response] We changed the sentence more correct. We used satellite data to extract water level. "In addition, we investigated whether satellite remote sensing data could (completely) replace in situ water level logger data to calculate lake water levels using the combined DSMs. We found that the water level logger measurements agreed with the water levels that were reconstructed from satellite data and combined UAV-derived DSMs. For example, we confirmed the position of the water level by comparing UAV orthorectify image and satellite data on 1 m counter lines extracted by the combined UAV-derived DSMs. Finally, we obtained the water level and water area based on satellite data. Using this method, we reconstructed the water level data between August 4 and 31 based on 10 satellite images from PlanetScope, Landsat-8/Operational Land Imager (OLI), and Sentinel-2, because we do not have water level data after our last visiting on 4 August 2019."

[Comment 90] Ln 117: You write "(including amount of glacier recession)" → It is clear to me that you can compute glacier surface elevation changes and terminus retreat from DEM/DSM differencing, fair enough, but it is not clear to me how you included (annual, i.e. from summer to summer, i.e. from one UAV/SfM-derived DSM to the next) glacier surface elevation changes into daily lake volume calculations... or do you mean you just looked at glacier surface elevation changes as well and just wanted to mention that? – This is not 100% clear to me, please rephrase accordingly

[Response] We deleted it. We improved the text in [Comment 89].

[Comment 91] Ln 117: "water-level" → "water level"

[Response] We changed it.

[Comment 92] Ln 118: "...time-lapse camera data based on UAV DSMs" → don't you mean "...time-lapse camera data and/combined with UAV DSMs"?

[Response] We changed it.

[Comment 93] Lns 118ff: "Using the same method, we also reconstructed..." → again, it's not clear at all in my opinion how satellite images were "combined" with UAV DSMs to reconstruct water levels of Korumdu lake (see respective comment above) → you have to write how you did this Moreover, what is the benefit of using satellite imagery (of different resolution and quality) to reconstruct lake water levels compared to using your time-lapse camera images? – is it just to compare two or more different data sources to get the same results and compare the latter? – or is it to extend the temporal resolution of you lake water level data? – or does it have something to do with what you write below ("because we visited at the lake on 4 August 2019")? – or is it to investigate whether satellite remote sensing data could (completely) replace in situ time-lapse camera data to calculate lake water levels using DSMs? – interesting, but not really clear to me here → can you add something on that, be clearer on that point?

[Response] We improved text about the same method in [Comment 89]. As the reason of using of satellite data, we do not have the water level data after 4th August 2019, because our last visiting was 4 Augnsug 2019. We just want to extend a water level data between 4 August and 31 August 2019. We also investigated whether satellite remote sensing data could (completely) replace in situ data of lake water levels using DSMs.

[Comment 94] Ln 120: "...because we visited at the lake on 4 August 2019." → delete "at"

[Response] We changed it.

[Comment 95] Ln 121: "We also investigated the changes in lake area during 2017–2019 using PlanetScope images." → ok, I guess You just manually digitized lake areas

from the satellite imagery and then compared lake areas? – maybe it would be clearer to write this Moreover, I think it would be really good and more transparent for people who read your paper if you could add a table containing all different sorts and sources of satellite imagery you used, including columns with "resolution", "acquisition date", and information on how the different satellite imagery were used (i.e. for which type of analyses described in your methodology section you used which satellite images)

[Response] We added Table 1.

[Comment 96] Ln 122: I think "meteorological" is the right term to use here instead of "climatic"

[Response] We changed it.

[Comment 97] Ln 123: delete "resolution"

[Response] We changed it.

[Comment 98] Ln 124: "...were calculated for 2016–2017" → whole year round or only during summer months? → please clarify

[Response] We improved it. "Mean annual air temperature (MAAT) between 2015 and 2017 and mean annual ground surface temperature (MAGST) during in 2015–2019 were calculated."

[Comment 99] Ln 126: "...were identified using satellite images on ArcGIS 10.5" → "were identified in ArcGIS 10.5 using satellite images."

[Response] We changed it.

[Comment 100] Lns 126ff: "In particular, 91 images from Landsat- 7/Enhanced Thematic Mapper Plus (ETM+, SLC-off) and Landsat-8/OLI, 31 images from Sentinel-2, and 16 images from PlanetScope acquired during 2013–2018. The resolutions of these images are 15 m (pan-sharpened images of Landsat-7/8), 10 m (Sentinel-2), and 3 m (PlanetScope)." → referring to my comment above, I think it would be very

good to make a table with all satellite imagery data you used (including columns with "resolution", "acquisition date", and information on how the different satellite imagery were used (i.e. for which type of analyses described in your methodology section you used which satellite images)); because you worked with many different satellite images from different source dates, with different resolution, for various analyses etc. If you do that then you don't have to write all this information in Lns 126ff, but you can refer to the table and can rephrase these lines in a more descriptive manner...

[Response] Satellite data which used in this study was summarized in Table 1. Please see [Comment 95].

[Comment 101] Ln 129f: "As a definition of short-lived lake, we use that in Daiyrov et al. (2018), which is based..."→ "We used the definition by Daiyrov et al. (2018) for short-lived lakes, which is based on seasonal changes in lake area over the summer months of each year"

[Response] We changed it.

[Comment 102] Ln 131: "...that appears or doubles in area" → "...that suddenly appears and/or increases substantially in area" (I would write this like this because these lakes don't necessarily have to double in area, you just want to express that they often increase substantially in area)

[Response] We changed it.

[Comment 103] Lns 131f: "We counted the number that appeared..." → "We counted the number of lakes that appeared..." Ln 132: "In addition, the number was tracked..." → "In addition, the number of lakes was tracked..." Ln 133: "extracted" → "digitized"

[Response] We changed it.

Results [Comment 104] Ln 137: "It sits in a basin formed during glacier recession." → better: "It developed in a depression that formed during the retreat of the glacier." Ln 138: "The basin developed..." → "The lake basin developed..." Lns 137f: "At the front

of the Korumdu glacier lies the Korumdu glacial lake (Fig. 2). It developed in a depression that formed during the retreat of the glacier. The lake basin developed inside an ice-cored moraine complex." → In my opinion these three sentences belong to the "Study area" section and it is not necessary to write that again here at the beginning of the results section. . .

[Response] These sentences were derived from our satellite data analysis. So, we remains here, but we improved the text depend on your changes.

[Comment 105] Ln 138: Why "Although"? – Makes more sense to delete this

[Response] We improved the sentence. "Most of the lake basin area had been covered by the Korumdu glacier, based on ALOS/AVNIR-2 data taken on 17 September 2007., The lake basin developed in a depression that formed during the retreat of the glacier. The UAV ortho-images indicated a lake basin length of 360 m, a width of 110 m, with and a total area of the lake basin of 0.062 km2 in 2019."

[Comment 106] Ln 138: ". . .most of the basin area. . ." → ". . .most of the lake basin. . ."? Ln 139: ". . .glacier, based on. . ." → delete the comma there Ln 139: ". . .a basin length. . ." → ". . .a lake basin length. . ." Ln 140: ". . .with total area of. . ." → ". . .and a total area of the lake basin of. . ." Ln 140: "The basin volume. . ." → "The lake basin volume. . ."; ". . .from 264,000 m3 in 2017 to 330,000. . ." → add "m3" after "330,000" Ln 142: "basin" → "lake basin" (2x)

[Response] We changed all.

[Comment 107] Ln 143: ". . ., but we found an outlet point. . ." → ". . ., but we found an outlet point where meltwater from the lake emerges from a subsurface ice-tunnel within the frozen moraine complex which is connected to the lake (Fig. 2)." → Then you can delete the whole next sentence as it's clear enough (i.e. delete "The existence of the outlet shows that lake water flows through an outlet ice-tunnel from the lake.")

[Response] We changed it.

[Comment 108] Ln 145: "basin" → "lake basin" Ln 145: "Drainage water was observed at the outlet point in 2015, 2017 and 2019, but not 2016 and 2018." → "Leakage of meltwater was observed at the outlet point in 2015, 2017 and 2019, but not in 2016 and 2018."

[Response] We changed it.

[Comment 109] Ln 146: "...becoming large on 30 July" → do you mean "...reached its maximum on 30 July..."?

[Response] We do not have a data about maximum level in 2015.

[Comment 110] Lns 147f: "...of the changes appears in the images in Fig. 4..." → better "...of changes in lake size is shown with a sequence of PlanetScope satellite images in Fig. 4." (then you can delete "..., which are based on PlanetScope satellite data."

[Response] We changed it.

[Comment 111] Ln 152: "...and reached its maximum level.." → "...and reached its maximum size..."

[Response] We changed it.

[Comment 112] Lns 152f: "In contrast, the lake area did not change dynamically in 2016." ok, but what does that exactly mean? – the lake appeared but did not expand substantially? – not clear to me at all. Moreover, do you know this from field surveys, the on-site time-lapse cameras or satellite data? – not clear, either... → please clarify here

[Response] We improved it. "according to the water level logger data and on-site time-lapse camera images, ..."

[Comment 113] Lns 153f: "Based on Landsat-8/OLI data, we also found that the lake appeared in 2014 (May 5, June 27, and September 10)." → ok, does that mean it

appeared (and thus also drained) three times or does that mean that you just found the lake on May 5, June 27, and September 10 2014 on the satellite imagery but do not know what happened in between? – this is not clear here → please rephrase and clarify, see also my comment just below

[Response] We improved it. "we also found that the lake existed on 5 May, 27 June, and 10 September in 2014. . ."

[Comment 114] Lns 154f: "Although these images show rapid drainage, we did not find evidence that the drainage caused flooding during the survey period" → my comment here goes a bit in the same direction as my comment just above: How can you see on three single (and "point in time") satellite images that the lake showed rapid drainage? – in my opinion you cannot! – If you have field evidence or know from other sources that the lake drained ok, but then you have to write this more clearly → please rephrase and clarify Also, you write ". . .we did not find evidence that the drainage caused flooding. . ." → evidence like what? – can you maybe be more concrete/precise here?

[Response] We deleted this sentence.

[Comment 115] Lns 155f: "According to data in Narama et al. (2018), drainage from Korumdu lake is the flood-wave type in the downstream region because the water stream flows on a gentle slope." → ok, can you please connect this sentence a bit better to the precedent one in terms of logic and context? Also, ". . .drainage from Korumdu lake is the flood-wave type. . ." → ok, but from the sentence just above (when you write ". . .we did not find evidence that the drainage caused flooding. . .") it seems that the lake can also drain without flooding, so these two statements are a bit contradictory. . . → can you please rephrase and clarify?

[Response] We moved this sentence to "study area".

[Comment 116] Ln 159: I would delete the first sentence ("Consider the properties of Korumdu lake from 2017 to 2019"), and then write. . .. (see comment just below)

[Figure]

Lns 159f: ... "Figure 6 shows the measured water level, lake area and volume, and inflow- outflow rate of Korumdu lake from 2017 to 2019" (→ I would not use the term "inflow-outflow discharge" because "inflow is not a discharge" → I recommend using "inflow-outflow rate", you would also need to change this accordingly in Fig. 6d)

[Response] We changed it.

[Comment 117] Lns 160f: "For 2017, we also show the water temperature (Fig. 6a). We also reconstructed the water level data between August 4 and 31 based on 10 satellite images (yellow points in Fig. 6a)." → better and clearer to shorten and write everything in one sentence: "For 2017, water temperature data were also recorded (Fig. 6a), and water level data between 4 and 31 August were reconstructed based on 10 satellite images (yellow points in Fig. 6a)."

[Response] We just showed the water temp data in 2017 in Fig. 6a although we recorded water temp in 2016-2019. So, we improved the sentence based on your suggestion.

[Comment 118] Lns 161f: "We calculated volume and discharge using the water levels and the UAV DSMs." → "Lake volume and discharge were calculated based on the water level data and the UAV DSMs." Ln 163: "...August 3..." → "...3 August..." Lns 163f: "..., and then vanishes on 19 August (Fig. 6a)." → "..., and then the lake is empty on 19 August (Fig. 6a)." Ln 164: Why "In the first 29 days..."? → write "Within 29 days, ..."

[Response] We changed all.

[Comment 119] Ln 165: "The resulting rate of volume increase was 8,070 m3/day." → "The resulting rate of lake volume increase was 8,1 m3 per day." (I think the uncertainty in the applied methodology to derive daily lake volumes is too high to write three decimal places → round to one decimal place)

[Response] This volume is 8,070 m3 per day, not 8.070 m3. So, we remains it.

[Comment 120] Ln 165: "During discharge, 234,000 m3 of water drains in 17 days, ..." → "During the emptying of the lake, 234,000 m3 of water drain in 17 days, ..." Ln 166: insert comma after "(Fig. 6b)" Ln 167: "August 3" → "3 August"

[Response] We changed all.

[Comment 121] Lns 167f: In my opinion it's a bit a pitty to only mention the average recorded lake temperature here (referring to Fig. 6a). From your graph one can clearly see that if the water level is low, lake temperature is high, and vice versa, which makes absolutely sense (heating of shallower lake with less water by solar irradiance is stronger than cooling from inflowing ice meltwater) → I think it would be worth it to add some words about the observed variability in recorded lake temperatures (as well as about the reasons for these measured lake temperatures)

[Response] We added one sentence. "The temperature varies largely with lower water level, because the heating of shallower lake with less water by solar irradiance is stronger than cooling from inflowing ice meltwater."

[Comment 122] Ln 168: Insert "(Fig. 6a)" after "1°C" Ln 169: "The first, on 25 July, reaches 3.5 m and a volume of 21,000 m3 (Fig. 6a, b)." → "The first peak on 25 July, showing a lake depth of 3.5 m and a volume of 21,000 m3 (Fig. 6a, b)." Ln 170: "The second, the yearly maximum, on 11 August, reaches 6 m and a volume of 53,000 m3." → "The second on 11 August, with a lake depth of 6 m and a volume of 53,000 m3, which corresponds to the maximum values in 2018." Lns 170f: "Finally, the third peak, on 17 August, reaches a level of 5 m and a volume of 39,000 m3." → "The third peak occurs on 17 August, showing a lake depth of 5 m and a volume of 39,000 m3." Ln 172: "Compared to the case in 2017, ..." → "Compared to 2018, ..." Lns 172f: "However, like that in 2017, the inflow rate is also intermittent in 2018." → "Similar to 2017, the inflow rate also clearly varies over time in 2018." Lns 173f: "The three peaks indicate that closure of the tunnel occurred several times during the 1-month period." → "The three peaks in water level, area and volume of Korumdu lake indicate that closure of

the ice-tunnel occurred several times during the one-month period." Ln 175: "In 2019, the water level goes up and down until 22 July, when it rises sharply (Fig. 6a)." → "In 2019, the lake water level rises and falls before 22 July, when it rises sharply (Fig. 6a)." Lns 175ff: "Then the level has a local maximum on 30–31 July, reaching 5 m and a volume of 53,000 m3, followed by a yearly maximum on 11 August, reaching 6.5 m and 74,000 m3 (Fig. 6 a, b)." → "Then, the water level shows an intermittent maximum around 30 – 31 July, reaching a lake depth of 5 m and a volume of 53,000 m3. 2019 maximum values were recorded on 11 August, with a lake depth of 6.5 m and a corresponding volume of 74,000 m3 (Fig. 6a, b)."

[Response] We changed all.

[Comment 123] Ln 178: Delete "Over these years, . . ." and start with "Other differences include. . ." Lns 179f: ". . .small discharge rates. . ."→ ". . .small lake discharge rates. . .

[Response] We changed it.

[Comment 124] Ln 180: ". . ., consistent with the lack of reported flooding." → ". . ., which is consistent with the absence of reported flooding." Ln 181f: "Concerning fluc-tuations, according to the water level data for 2017–2019, the level increased with repeated storage-drainage cycles." → clearer: "According to the water level data of 2017–2019, the lake level rose and fell several times, indicating repeated storage-drainage cycles." Ln 182: ". . .small increases of water level. . ." → ". . .small increases in water level. . ."; ". . .with the level. . ."→ ". . .with the lake level. . ." Ln 185: "We ob-served drainage water at an outlet point. . ." → "We observed lake water leakage at an outlet point. . ." Lns 185f: "The reason we argue is due to the relative elevations." → "We argue that this might be due to the difference in relative elevations between the lake level and the outlet ice-tunnel entrance." Lns 186f: "The water levels were 3,810 m on 21 Aug 2015, 3,816 m on 6 Aug 2017, and 3,810 m on 4 Aug 2019, all of which are higher than the outlet point at the basin." → "The water levels were at 3,810 m a.s.l on 21 August 2015, 3,816 m a.s.l on 6 August 2017, and 3,810 m a.s.l on 4 August

2019, thus always higher than the outlet ice-tunnel entrance at approximately 3,807.5 m a.s.l." Lns 187f: "However, we did not observe water drainage in 2016 and 2018 because the water levels were 3,806.5 and 3,807.5 m, respectively (Fig. 8a, c)." → rephrase to avoid repetition of what is written in Ln 185 → "In 2016 and 2018, lake water levels were at 3,806.5 m a.s.l and 3,807.5 m a.s.l, respectively, thus always lower than the outlet ice-tunnel entrance at approximately 3,807.5 m a.s.l (Fig 8a, c). Therefore, no lake water leakage was observed at the outlet point of the ice-tunnel in 2016 and 2018."

[Response] We improved all. But we change two sentences. "The water levels were at 3,810 m a.s.l on 21 August 2015, 3,816 m a.s.l on 6 August 2017, and 3,810 m a.s.l on 4 August 2019, thus always higher than the outlet ice-tunnel entrance at approximately 3,807.5 m a.s.l." "Always" is not correct, because this level was just during our visiting. So, we used "the water levels higher than. . .."

[Comment 125] Ln 188f: "These results indicate that. . ." → following my rephrasing of the paragraph above, you can delete this whole last sentence!

[Response] We deleted it.

[Comment 126] Ln 191: "4.3 Surface changes around Korumdu lake" → more correct to write "4.3 Surface elevation changes around Korumdu lake"

[Response] We changed it.

[Comment 127] Lns 192f: "Over a period of one year, how does the region near the entrance of the outlet ice-tunnel change? To answer this question, we compared UAV orthoimages with DSM data in 2015 and 2016 (Fig. 9)." → "To investigate annual surface elevation changes near the entrance of the outlet ice-tunnel, we compared UAV-derived orthoimages with DSMs from 2015 and 2016 (Fig. 9)." Lns 193f: "A comparison of Fig. 9a, b shows debris sliding, with horizontal backwasting of an exposed ice ridge by 7m." → "Debris sliding and horizontal backwasting by 7 m of an exposed ice ridge between

2015 and 2016 appear from the comparison of the orthophotos." Lns 194f: "The back-wasting indicates melting occurred, which is supported by the UAV-derived DSMs in Fig. 9c." → "The backwasting indicates that melting of debris-covered ice occurred, which is supported by comparing the UAV-derived DSMs from both years (Fig. 9c)." Lns 195f: "In particular, along the profile (a–a'; Fig. 9b) of the landform between 2015 and 2016, the surface elevation decreases by about 5 m." → "For instance, along a cross-sectional profile (see a–a' in Fig. 9b), the surface elevation decreased by about 5 m (Fig. 9c)."

[Response] We changed all.

[Comment 128] Lns 196f: "These results indicate that the surface motion and deposition of debris can cause closure of the outlet ice-tunnel during summer." → I don't think that this is really clear here. Do you have real evidence that debris from the surface of melting ice blocked the entrance of the outlet ice-tunnel? What is the diameter of the ice-tunnel at its entrance (i.e. can it easily/quickly be blocked by mobilized sediments?) And how are these sediments transported from the ice margin to the entrance of the outlet-ice tunnel where they cause blocking of the ice-tunnel? How close are the features shown in Fig. 9 to the entrance of the ice-tunnel? Are these processes really directly linked? And what is the grain size distribution of the debris (I mean to block the tunnel entrance you need to have a mix of finer and coarser material so that lake water doesn't leak through the deposited debris material anymore)? And how does reopening of the tunnel entrance work after closure by debris deposition? I think you could very well be right with what you're saying but this is all not very clear from the information I have now from the text and the figures of the manuscript... → Please clarify, add on this

[Response] We improved the sentence. "These results indicate that the surface motion and ice-debris deposition due to ice-melting "might" cause closure in the outlet ice-tunnel during summer."

"These results indicate that the surface motion and ice-debris deposition due to ice-melting might cause closure in the outlet ice-tunnel during summer. In the fieldwork 2016, we observed the entrance of ice-tunnel and water flow to the entrance of ice-tunnel. After two or three hours, we confirmed the increase of lake levels in field (Fig. 7), indicating that closure of ice-tunnel."

[Comment 129] Ln 198: delete "mountain" or write "…discontinuous mountain permafrost…"; and write "3,100–3,200 m a.s.l"

[Response] We changed it.

[Comment 130] Ln 199: "…in 2015–2017…" → "…between 2015 and 2017…" Ln 200: "…in 2015–2019…" → "… during 2015–2019…"

[Response] We changed all.

[Comment 131] Lns 201f: "…such as that for a supraglacial lake on a debris-covered glacier…" → "…comparable to supraglacial lakes on debris-covered glaciers…" Moreover, I think that this sentence is kind of a repetition from the paragraph just above in the manuscript (Lns 192-197) → Maybe better to delete the sentence here and include the references in the paragraph above (Lns 192-197), would fit better…

[Response] We have deleted all sentence.

[Comment 132] Ln 205: "4.4 Comparison to other short-lived lakes in the area" → "4.4 Comparison to other short-lived glacial lakes of the Teskey Range"

[Response] We changed it.

[Comment 133] Ln 206: "…had relatively little drainage…" → "…showed relatively little drainage during emptying…" Ln 206: "…, whereas four other short-lived lakes…" → I would add the names of these lakes in parentheses here Ln 207: "…caused large drainages…" → "…caused larger drainage…"

[Response] We changed all.

[Comment 134] Ln 208: "...these appearance times..." → do you mean the (compared to Korumdu lake) earlier appearance times here → rephrase in order to be clear

[Response] We improved the sentence. "how common these appearance times of other short-lived glacial lakes are, ..."

[Comment 135] Ln 209: "...short-lived lakes..." → "...short-lived glacial lakes..."

[Response] We changed it.

[Comment 136] Lns 209f: You had a look at the months from June to September 2013–2018 to determine the appearance times of other short-lived glacial lakes, ok, but just above you refer to four lakes described by Narama et al. (2010, 2018) appeared from May to June... So why didn't you also consider May 2013–2018?

[Response] According to satellite data, there are too much snow in May through the 2013-2018. It is difficult to check all lakes. We focused from June to September during 2013-2018.

[Comment 137] Lns 211f: "...such short-lived lakes during 2013–2018 (the total includes re-appearances of the same lake in different years) in the study area." → "...such short-lived glacial lakes in the northern Teskey Range during 2013–2018 (the total number of lakes includes reappearances of the same lake in different years)." Ln 212: "In Fig. 10, we classify these by month of appearance." → "A classification of these lakes by month of appearance is shown in Fig. 10 for the six year period."

[Response] We changed all.

[Comment 138] Lns 212f: "The appearance months with the most lakes are June, the snow-melt period, and July, the ice-melt period; specifically, 43 lakes in June and 90 in July." → "Most lakes appeared in June (43 lakes), the snow-melt period, and July (90 lakes), the ice-melt period." Moreover, you can also have snow melt earlier or later than June, and ice melt earlier or later than July. Would it make sense to write "...June (43 lakes), the period of maximum snow-melt, and July (90 lakes), the period of maximum

ice-melt."?

[Response] We changed it.

[Comment 139] Ln 214: "...in these two periods..." → "...for these two periods..."

[Response] We changed it.

[Comment 140] Lns 214f: Would it be right to write "This large variability is related to different meteorological conditions during summer months of 2013–2018."? I think this would be clearer.

[Response] We improve the sentence. "This large variability is related to geomorphological conditions such as drainage through ice tunnel inside of moraine complex, and it was not directly related to local climate change (Daiyrov et al., 2018).

[Comment 141] Lns 216f: "Concerning re-appearances, 81 lakes appeared only once for 6 years. Of the remaining, 19 appeared twice, 7 appeared 3 times, 2 appeared 4 times, and 2 lakes appeared all 6 years." → "Concerning reappearances, 81 lakes appeared only once during six years. Of the remaining, 19 lakes appeared twice, 7 lakes appeared three times, 2 lakes appeared four times, and 2 lakes appeared every year."

[Response] We changed it.

[Comment 142] Ln 217: "indicating that tunnel closure occurred with a different month each year." → Does this refer to the two lakes that appeared every year? – not very clearly written, please rephrase

[Response] We improved the sentence. "Of the remaining, 19 lakes appeared twice, 7 lakes appeared three times, 2 lakes appeared four times, and 2 lakes appeared all six years. These results indicate that tunnel closure caused short-lived glacial lakes in the northern Teskey Range occurred with a different month each year."

[Comment 143] Ln 218: "Short-lived lakes that reappear many years likely have a tunnel condition in which closure occurs easily." → "Short-lived glacial lakes that reappear during many years likely show geomorphological settings at the drainage tunnel entrance which favor tunnel closure and hence lake formation."

[Response] We changed it.

Discussion [Comment 144] Ln 221: "5.1 Cause of outlet ice-tunnel closure at Korumdu lake" → "5.1 Causes of outlet ice-tunnel closure at Korumdu lake"

[Response] We changed it.

[Comment 145] Lns 222f: "In the case of ice tunnel closure, the supraglacial lakes on the debris-covered Inylchek Glacier in April–May are likely to appear due to the closure of englacial conduits when stored water freezes (Narama et al., 2017)." → "In the context of ice-tunnel closure, Narama et al. (2017) report that the supraglacial lakes on the debris-covered Inylcheck Glacier appear in April–May due to the closure of englacial conduits by freezing of stored water."

[Response] We changed it.

[Comment 146] Ln 228: I would replace "...in the study region..." with "...in the northern Teskey Range..." Ln 229: "...or by blockage by collapsing with deposition of mixed debris and ice..." → "...or by blockage with depositions of ice-debris mixture after roof collapsing..." Lns 231f: "The short-lived lakes here that caused the four large drainages (2006, 2008, 2013, and 2014) appeared in May–June and expanded in June–July (Narama et al., 2010, 2018)." → "Four short-lived glacial lakes of the Teskey Range that caused four large drainage events (2006, 2008, 2013, and 2014) appeared between May and June and expanded in area until June–July (Narama et al., 2010; 2018)." Ln 232: "The timing suggests a closure that..." → "The timing of lake appearance suggests an ice-tunnel closure that..."

[Response] We changed all.

[Comment 147] Ln 233: Do you mean "We call this the deposition-freezing type of

ice-tunnel closure."? → I would rephrase this accordingly...

[Response] We changed it.

[Comment 148] Ln 234: Do you mean "However, for none of the case studies investigated by Narama et al. (2010, 2018), neither geomorphological behavior of the ice-tunnel nor water level fluctuations were studied in detail." → I would rephrase this accordingly...

[Response] We changed it.

[Comment 149] Ln 235: "...excluding the case of no expansion..." → "...excluding the case of no lake expansion..."

[Response] We changed it.

[Comment 150] Ln 236: I would replace "based on water-level of a data logger and time-lapse camera images." by "based on our field surveys." (smoother and the reader knows your survey methods from the parts of the manuscript further above)

[Response] We changed it.

[Comment 151] Ln 236: "...changes in the basin..." → "...changes in the lake basin..." Ln 238: "...likely was caused by..." → "...were likely caused by..." (plural because you write about "the blockages") Ln 239: "Further evidence that Korumdu lake forms by the deposition process comes from consideration of water-level fluctuations." → "Looking at water level fluctuations of Korumdu lake gives further evidence for lake formation by deposition of ice and debris."

[Response] We changed all.

[Comment 152] Ln 240: "The fluctuations of water level, such as spikes, reveal changes in the tunnel condition (Fig. 6d)." → I'd suggest "The fluctuations of lake water level and discharge spikes reveal changes in the ice-tunnel morphology (Fig. 6d)."

[Response] We changed it.

[Comment 153] Ln 241: "...the water increase was..." → "...the water level increase was..." Ln 242f: "...ice tunnel..." → "...ice-tunnel..." Ln 243: delete "also" Ln 244: "...to the water pressure..." → "...to changes in water pressure..."; I suggest replacing "...or thermal erosion." with "...or deposition of ice-debris mixture through melting processes." Lns 245ff: "In 2017, the trend of water volume increase consisted of two parts: 5 to 25 July and 26 July to 3 August (Fig. 6b). The first period had sporadic fluctuations, indicating incomplete closure of the tunnel, but the second period had a smooth increase, indicating complete closure." → "In 2017, there were two periods of varying patterns of lake water volume increase (Fig. 6b). The first period (5 to 25 July) revealed sporadic fluctuations in increasing water volume, indicating incomplete closure of the ice-tunnel. However, the second period (26 July to 3 August) showed a continuous and rapid increase in water volume, indicating complete closure of the ice-tunnel." Ln 247: delete "value" Ln 248: "Longer closure periods..." → "Longer periods of tunnel closure..."; "...larger short-lived lakes..." → "...larger short-lived glacial lakes..." Ln 249: "Thus, the period of closure might be determined by the condition of the tunnel." → "Thus, the period of closure is likely determined by the morphology of the ice-tunnel." Lns 250f: "Many of the other short-lived lakes that also appear in the ice-melting period are likely to be the deposition-closure type, for the same reasons we applied to Korumdu lake." → "As for Korumdu lake, many of the other short-lived glacial lakes in the northern Teskey Range which were detected based on satellite imagery are likely to belong to the deposition-closure type as well."

[Response] We changed all.

[Comment 154] Lns 251f: "For example, in Fig. 12, we show surface changes in the outlet ice-tunnel at the Jeruy glacial lake between 2014 and 2016." → the observed surface changes are rather around the lake or above the ice-outlet channel I think, and not in the ice-tunnel itself → I would rephrase as follows: "For example, Figure 12 shows changes in surface elevation and the outlet ice-tunnel of the Jeruy glacial lake

between 2014 and 2016."

[Response] We changed it.

[Comment 155] Ln 252: I would replace "large" with "distinct" Ln 253: ". . .making closure likely" → ". . ., which likely led to tunnel closure." Lns 253: "Thus, the surface condition always changes. . ." → better: "Thus, morphology and surface characteristics of an ice-cored moraine complex within the mountain permafrost zone are prone to frequent changes, and. . ." Ln 254: ". . .and the deposition-closure type is the major type in this region." → clearer to write: ". . .and the deposition-closure type is likely the main type for drainage tunnel blockage and hence formation of short-lived glacial lakes in the northern Teskey Range."

[Response] We changed all.

[Comment 156] Lns 254f: "Thus, the appearance of a short-lived glacial lake is inevitable in summer when the melting rate is high." → following my suggestions to rephrase the subsequent sentence, I would rephrase this one as follows to make things clear: "If deposition-closure processes occur in summer when the melting rate is high, the formation of a short-lived glacial lake is highly likely."

[Response] We changed it.

[Comment 157] Lns 255f: "The characteristics of this lake disaster might be shown in another Asian mountain permafrost region." → This sentence makes not much sense to me and is quite misleading, as you primarily write about geomorphological processes linked to short-lived glacial lake formation and drainage! You hardly say anything about risks or disasters related to the emptying of the studied lake type! I really think it's better to delete this whole sentence.

[Response] We deleted it.

[Comment 158] Ln 258: "5.2 Relationship between outlet tunnel size and drainage scale" → "5.2 Relationship between outlet tunnel size and lake drainage" Lns 259f:

".. .had large drainages." → ".. .showed considerable drainage."

[Response] We changed all.

[Comment 159] Ln 261: ".. .with Jeruy's outlet cross-section being.. ." → ".. .with a cross-section measuring 8m2 at Jeruy.. ."; ".. .4 x 2 m2.. ." → was the tunnel width 2 m and the height 4 m? So a cross-section of 8 m2 in area? → clarify

[Response] We improved it. "These lakes had relatively large outlet tunnels, with a cross-section measuring about 8 m2 in area at one location of Jeruy (Fig. 12a,b) and a cross-section of about the same size or larger at Karateke."

[Comment 160] Ln 262: ".. .and Karateke's about the same or larger (not shown)." → ".. .and a cross-section of about the same size or larger at Karateke."

[Response] We changed it.

[Comment 161] Lns 262f: "Earlier, back in 2008, the w-Zyndan lake of 437,000 m3 had a discharge rate of.. ." → "In 2008, the w-Zyndan lake (437,000 m3) emptied at a discharge rate of.. ." Ln 264: ".. .did not have a large drainage during.. ." → ".. .did not show as high drainage rates during.. ." Ln 265: ".. .than those of.. ." → ".. .than those at.. ." Ln 266: ".. .of the two large drainages.. ." → ".. .of the two large drainage events.. ." Ln 267: ".. .with Korumdu lake.. ." → ".. .for Korumdu lake.. ." Lns 267f: ".. ., which is behavior consistent with closure of a small channel caused by deposition." → ".. ., which was related to closure of the small outlet ice-tunnel caused by deposition of and blockage by debris." Lns 268f: ".. .ensured a slower discharge even when it became full (300,000 m3)." → ".. .resulted in slower lake discharge even when lake volume reached its maximum (300,000 m3)."

[Response] We changed all.

[Comment 162] Lns 270f: "These results show that the lake size and the dimensions of the outlet ice-tunnel are related to the scale of discharge." → But above you just say that there was no clear relation between lake size and discharge rate for Korumdu

lake during 2017-2019! → "These results show that, at least for Korumdu lake, the dimensions of the outlet ice-tunnel were the dominant factor controlling lake discharge rates."

[Response] We changed it.

[Comment 163] Lns 273f: I would place "...enlarging the outlet ice-tunnel..." before the references here

[Response] We changed it.

[Comment 164] Ln 274: "...although basin-size changes depend on the particular glacier landforms, the basin area in the case of Korumdu lake has increased each year due to glacier recession." → "...although lake basin-size changes depend on the particular glacial landforms, the basin area of Korumdu lake has increased each year due to glacier retreat." Moreover, I am not very sure what you exactly mean here... how do which glacial landforms exactly influence lake basin-size? → please clarify

[Response] We improved the sentence. "In addition, although lake basin-size changes depend on the particular thermal erosion on ice-cored moraine complex, the basin area of Korumdu lake has increased each year due to glacier retreat."

[Comment 165] Ln 276: "...in this region,..." → "...in the Teskey Range,..."; "...during their discharge may increase in the future..." → "...during their discharge may become more frequent in the future..."

[Response] We changed it.

Conclusions [Comment 166] Ln 278: "6 Conclusion" → "6 Conclusions"?!

[Response] We changed it.

[Comment 167] Ln 279: "Our field survey found..." → "From our field survey we found that..."

[Response] We changed it.

[Comment 168] Ln 280: delete "Later, ..."

[Response] We changed it.

[Comment 169] Ln 281: "...the draining process was relatively slow..." → "The lake drainage was always relatively slow..."; "outlet ice tunnel" → "outlet ice-tunnel"; "...scale of discharge..." → "...discharge rate..." Ln 282: "...sizes..." → "...size..." Lns 282f: "We argued that predicting the scale of a drainage requires knowledge of the outlet ice-tunnel dimensions and the lake's depression size." → "We argue that predicting drainage rates requires knowledge about the dimensions of the outlet ice-tunnel and the size of the lake basin."

[Response] We changed all.

[Comment 170] Lns 283f: "Our research method of combination between water-level data and UAV DSMs could estimate the discharge and the approximate dimensions of the tunnel." → "By combining water level data and UAV-derived DSMs from consecutive years, we were able to estimate daily lake discharge and approximate dimensions of the outlet tunnel." I am not sure whether it is clear enough from the manuscript that (and how) you could estimate the dimensions of the outlet ice-tunnel from DSM differencing, i.e. from interpreting surface elevation changes. – In my opinion you rather wrote about ice melt, changing surfaces of the ice-cored moraine, backwasting and debris sliding... I did not read really much about how you estimated approximate dimensions of the outlet tunnel in the results section (4.3). If you write this in the conclusions, you would have to elaborate and talk about that in the results chapter... You cannot bring up "new results" only in the conclusions section! → please add some text in section 4.3 accordingly

[Response] We added some sentence in results section (4.3). The relative dimention size was also estimated by the fluctuations of lake water level and discharge spikes.

The ice-tunnel of large dimension size does not show small fluctuations of lake levels due to ice melting.

[Comment 171] Ln 285: "During 2013–2018, satellite data showed this region to have 160 short-lived glacial lakes, ..." → "Based on satellite images from 2013–2018, 160 short-lived glacial lakes were detected in the northern Teskey Range, ..." Lns 286f: "Four lakes that appeared a month earlier had large drainages, the only cases of large drainage in the study." → "Four lakes that appeared a month earlier showed drainage rates which were significantly higher compared to the rest of the lakes." Lns 287f: "Nevertheless, with a warming climate, any short-lived glacial lake might cause large flooding if the outlet ice tunnel and basin size sufficiently enlarge." → better and more specific: "However, with a warming climate resulting in enlarging outlet ice-tunnels and lake basin sizes, also other short-lived glacial lakes of the northern Teskey Range might cause large flood events." Lns 289f: "The glacial lake outburst floods (GLOFs) which caused by moraine-dam failure such as Himalaya and Andes are minor cases in this region." → "Glacial lake outburst floods (GLOFs) caused by moraine-dam failure, as frequently observed in the Himalayas or the Andes, rather rarely occur in the northern Teskey Range." Lns 290ff: "Short-lived lakes which caused by closure and opening of an outlet ice-tunnel in moraine complex are a major hazard in this region, because the short-lived lake exists on an ice-cored moraine complex within geomorphological and climate conditions of the mountain permafrost zone." → "Short-lived glacial lakes that form on ice-cored moraine complexes within the mountain permafrost zone through closure and opening of subsurface outlet ice-tunnels are a major hazard in the northern Teskey Range." Lns 293f: "These new knowledges are useful to understand the phenomena and behavior of the short-lived lakes and consider glacier hazard mitigation in the mountain permafrost regions of Asian high mountains." → "Insights from monitoring short-lived glacial lakes in permafrost zones are useful to better understand their characteristics and behavior, and therefore important for mitigation of glacier-related hazards in high-mountain areas of Central Asia."

[Response] We changed all.

[Comment 172] Lns 294f: "A threat of the short-lived lakes increases for the residents since 2000s. This hazard case might be major in Asian high mountains in present." → Apart from the fact that those two sentences would have to be rephrased in order to be clear, I would directly delete them because in your study you don't really address risks from glacier-related hazards and you have not shown that such threats have increased for residents since 2000. Hence, this would be something totally new at the end of the paper, and in the conclusions you should not come up with something new!

[Response] We deleted it.

Figures [Comment 173] Figure 1: I would enlarge the figure if possible and also add names of specific glacial lakes or rivers that you mention by name in the manuscript (see also corresponding comments above). Figure Caption: "...located on the south shoreline of Issyk-Kul Lake,..." → "...south of Lake Issyk-Kul,..."; "Red circles are..." → "Red circles indicate locations of..."; "Green squares with checks are..." → "Green squares with checks show short-lived glacial lakes..."; "...caused large drainages..." → "...caused large drainage events..."

[Response] We changed it.

[Comment 174] Figure 2: I think it would be beneficial for the reader of the paper to show locations of your time-lapse camera, water level measurement logger, water temperature logger, air and ground temperature loggers in Fig. 2 (see respective comments above referring to the methods section). Moreover, in my opinion, this is not really a "geomorphological map" (i.e. not really a map of landforms and processes) → I would just write "Overview of the Korumdu glacier front" in the figure caption instead of "Geomorphological map of the Korumdu glacier front".

[Response] We changed it.

[Comment 175] Figure 4, figure caption: "...of the Korumdu lake area during 2017–

2019." → "...showing the evolution of Korumdu lake in (a) 2017, (b) 2018, and (c) 2019."

[Response] We changed it.

[Comment 176] Figure 5, figure caption: I would rephrase as follows: "Evolution of Korumdu lake during 2017–2019 based on time-lapse camera images acquired in the field."

[Response] We changed it.

[Comment 177] Figure 6: I would write "air temperature" for the labeling of the right y-axe in Fig. 6a; Labeling of the y-axes: write "lake volume (m3)" and "lake area (m2)" (Fig. 6b, c); labeling of the x-axes; Fig. 6d: labeling of the y-axe: better write "inflow-outflow rate (m3/s)"? See my comments thereupon further above... Figure caption: I would delete the first sentence and rephrase the figure caption as follows: "(a) Water level of Korumdu lake in 2017–2019 and air temperature in 2017. (b) Lake volume. (c) Lake surface area. (d) Inflow-outflow rate. These data from 2017 to 2019 are based on water level logger data, UAV DSMs, time-lapse camera images, and PlanetScope satellite images."

[Response] We changed it.

[Comment 178] Figure 7: You have to add "(c)" and "(d)" to the right images and "(b)" has to be replaced to the second image I guess (otherwise it doesn't make sense with what you write in the figure caption. Figure caption: I would write "...increase in water level of Korumdu lake."

[Response] We changed it.

[Comment 179] Figure 8: Better write "maximum lake extension/area" in Fig. 8a instead of "basin line"; I would write "m a.s.l" instead of "m" wherever you refer to "elevation values" Figure caption: write out the name of the months ("August", "July"); in addition, write "Orthoimages were acquired from UAV surveys."

[Response] We changed it.

[Comment 180] Figure 9: write "Elevation (m a.s.l)" in the labeling of the y-axis of Fig. 9c; Figure caption: better write "...of the debris-covered stagnant ice/dead ice at the entrance of..."?; write "...based on UAV orthoimages."; "(a) On 21 August 2015."; delete "line" after "exposed ice edge"; "(b) Same as (a) except on 12 August 2016."; "...show the new positions of the respective surface features after one year."

[Response] We changed it.

[Comment 181] Figure 10: I would write out the names of the months in the legend of the graph. Figure caption: I would add "...derived by Landsat-7/8, Sentinel-2, and PlanetScope satellite images." at the end of the sentence!

[Response] We changed it.

[Comment 182] Figure 11: With the dotted black line you show the level of the lake-dam crest; I would write "level of lake-dam crest" in Fig. 11a instead of just "lake-dam" Figure caption: write "The two types of ice-tunnel closure occurring in the northern Teskey Range."; better: "(a) Deposition-freezing type of closure in case of an outlet ice-tunnel being blocked by freezing of stored water or deposition of debris and ice."; "(b) Deposition-closure type of closure in case of an outlet ice-tunnel being blocked by deposition of debris and ice by thermal erosion (ice melt)."

[Response] We changed it.

[Comment 183] Figure 12: I would add information about the width and height of the ice tunnel shown in Fig. 12b (2 x 4 m). Figure caption: write "...which drained on 15 August 2013."; write "(a) Lake basin of Jeruy glacial lake on 9 August 2014. The white arrow shows the direction of lake drainage. (b) Insight into the outlet ice-tunnel on 9 August 2014. (c) The outlet ice-tunnel area on 9 August 2014. The white circles in (c) and (d) show the same location. (d) Same as (c) except on 9 August 2016."

[Response] We changed it.

---

## Author Comment (AC2) · 19 Jan 2021

We thank you for their valuable comments, all of which helped to improve the manuscript. Our response to each comment is below. We showed our replay for reviewer comments.

General comments: This is an interesting study about ice tunnels of short-lived lakes in parts of the Tien Shan. A main problem with the paper is that it becomes not very clear over large parts what its focus is: Is it ice tunnels, is it Korumdu lake? Is Korumdu lake an example, or a main focus? Why Korumdu lake? The authors should at the beginning develop and explain the purpose of the paper, and then relate to this purpose

throughout the paper more clearly.

[Response] Our paper focused on short-lived type of glacial lakes in the Teskey Range of northern Tien Shan, Kyrgyz Republic. The Korumdu lake is good example of short-lived type which we could collect field survey data in detail, because we cannot know which lake appear every year. For none of the case studies investigated by previous studies, neither geomorphological behavior of the ice-tunnel nor water level fluctuations were studied in detail. We succeeded to get water level data directly based on field survey.

We added the sentences about reason to choose the Korumdu lake in "Study area". "As the reason why this lake was selected as a research site, (i) the lake is a short-lived type which appears every year, (ii) it is easy to access the field, and (iii) this lake is located at the Tong region where four large outburst floods occurred in the past."

Specific comments: Abstract: [Comment 1] The abstract is unclear. Needs to be rewritten thoroughly. What is the relation of Korumdu lake with respect to the other lakes, not mentioned by name? Only later in the text it becomes clear that the paper is about Korumdu lake.

[Response] We improved abstract. Please see response comment for General comments.

Introduction [Comment 2] Well written, but I recommend an additional paragraph summarizing the previous findings from a number of papers of the authors about short-lived lakes, and how this paper relates and adds to these previous papers. Is it a new outburst, not covered in the previous papers? Why was it not covered in the previous overview papers? Something special with this lake? What new knowledge is expected compared to the previous papers? Is there a special focus of this study (on ice tunnels?), not covered in the other studies? Etc.

[Response] We added references and explanation in detail. We also showed subjects

in this study based on previous studies. In this study, the changes of water level were clarified by direct observation in field work. This is first results about short-lived lake.

Study area [Comment 3] This section suffers from the lack of clarity in the paper focus. You introduce the study region, not the lake Korumdu, but then you start investigating one specific lake. You need to introduce the region and the specific lake, and make clear why you investigate in detail lake Korumdu. What makes this lake particularly useful or necessary for ice tunnel investigations in addition to the ones studied earlier?

[Response] We added reason to choose the Korumdu lake in "Study area".

Results [Comment 4] L189: too low for drainage? Do you mean the lake did not run over in 2016 and 2018? Where did the melt water from the basin go then?

[Response] The lake did not appear in 2016. Lake was empty at our visiting in 2018, But lake appeared in this year as shown by lake level data and time-lapse camera images (Figs. 4-6). This sentence is written about lake level and drainage during our visiting. We improved the sentences.

"During the fieldwork, we observed lake water leakage at an outlet point in 2015, 2017, and 2019, but not in 2016 and 2018. We argue that this might be due to the difference in relative elevations between the lake level and the outlet ice-tunnel entrance. The water levels were at 3,810 m a.s.l on 21 August 2015, 3,816 m a.s.l on 6 August 2017, and 3,810 m a.s.l on 4 August 2019, thus the water levels higher than the outlet ice-tunnel entrance at approximately 3,807.5 m a.s.l. In 2016 and 2018, lake water levels were at 3,806.5 m a.s.l and 3,807.5 m a.s.l, respectively, thus the water levels lower than the outlet ice-tunnel entrance at approximately 3,807.5 m a.s.l (Fig. 8a, c). Therefore, no lake water leakage was observed at the outlet point of the ice-tunnel in 2016 and 2018 during our visiting."

Discussion [Comment 5] L250: deposition-closure type? This term/type was not introduced before. What is the difference to the deposition-freezing type?

[Response] We explained for these types in text.

Popov (1987) and Narama et al (2010a, 2018) have reported about the past drainages of short-lived lakes in the northern Tien Shan. Narama et al., (2010, 2018) documented that four short-lived glacial lakes which caused four large drainages appeared between May and June due to closure of ice-tunnel by freezing of stored water during winter or ice-debris deposition. This is "deposition-freezing type". Our present study found many short-lived lakes including Korumdu lake appeared in July–August. The freezing is not reason in this season. During July-August, thermal erosion caused deposition of ice and debris which blockaded the outlet ice-tunnel at its entrance or interior. Therefore, we call this type as "deposition-closure type".

"The timing of lake appearance suggests an ice-tunnel closure that is caused by the freezing of stored water during winter or deposition of ice-debris mixture (Fig. 11a). We call this the deposition–freezing type of ice-tunnel closure. However, for none of the case studies investigated by Narama et al. (2010; 2018), neither geomorphological behavior of the ice-tunnel nor water level fluctuations were studied in detail. In the case of Korumdu lake, the tunnel closed in July–August of every year since 2014 (excluding the case of no lake expansion in 2016) based on our field surveys. As we observed changes in the lake basin on the ice-cored moraine complex caused by subsidence or downwasting (Fig. 9), the blockages of the outlet ice-tunnel at its entrance or interior were likely caused by deposition of ice-debris mixture due to thermal erosion. This type of blockage (deposition-closure type) is sketched in Fig. 11b. Looking at water level fluctuations of Korumdu lake gives further evidence for lake formation by deposition of ice-debris mixture."

[Comment 6] L254-256: I don't understand these sentences. "inevitable"? Do you say every moraine complex will lead to a short-lived lake? I don't get the purpose of the last sentence.

[Response] Narama et al. (2018) and Daiyrov et al. (2018) found that short-lived

glacial lakes formed on the ice-cored moraine complexes under mountain permafrost zone. Because ice-tunnel has developed inside of ice-cored moraine complex. Not all moraine complexes lead to short-lived lakes in study area. However, our previous studies showed that four past drained short-lived lakes formed on these ice-cored moraines. Because these ice-cored moraine complexes within the mountain permafrost zone include stagnant ice, surface changes are caused the expansion of the outlet ice tunnel and formation of the depressions.

[Comment 7] L270: But what causes what? Does more discharge lead to larger tunnels, or do larger tunnels enable more discharge? I would expect that large discharge melts the tunnel walls and enlarges the tunnel. But you seem to argue that other way round? Further: what influence has the drainage catchment size and the amount of melt water available? Could it be that larger catchments produce more water which then causes larger tunnels?

[Response] This is drainage case of ice-tunnel. Dimension of ice-tunnel does not change suddenly. In the Korumdu lake, lake volume (234,000 m3) was larger than past large drainage cases in 2006, 2013 and 2014. However, this lake never caused hazardous floods before, because tunnel size related to discharge scale was too small. Our paper shows larger dimension of tunnel causes large discharge.

Discussion [Comment 8] Why don't you summarize your two types of tunnel closures? I think these are important to understand your conclusions of paragraphs 1 and 2. Last paragraph needs rewriting. Especially the conclusions regarding hazards are not well discussed and backed-up in the text.

[Response] We improved the discussion part more clearly.

Technical corrections [Comment 9] Line 49: "but this relationship was so far little studied with regard to proglacial lakes as of concern in this study." Or something like that.

[Response] We changed it.

[Comment 10] L54: supply "from"?

[Response] We changed it.

[Comment 11] L55: from the "lake" depression.

[Response] We changed it.

[Comment 12] L64: As changes "related to short-lived lakes" can occur...

[Response] We changed it.

[Comment 13] L74: clear -> clarify.

[Response] We changed it.

[Comment 14] L110: SA) Structure from Motion. Remove "of"

[Response] We deleted it.

[Comment 15] L131: does the lake need to "double"??? or just increase in area?

[Response] The lake increase in area.

[Comment 16] L155: lake is of flood-wave type... Do you want to say that the slope is too low that the flood would incorporate debris and become a debris flow?

[Response]

That is right. As shown in Narama et al. (2018), drainage water from short-lived lake is clean water without debris. So, after drainage, water is flood wave type (Huggel et al., 2004). However, most stream from glacial lakes changed the flow type to debris flows. The flood wave without moraine deposits can transform into a debris flow where the channel gets steeper and the wall-soft material erodible. The change occurs because banks of the channels are often composed of loose material (Haeberli, 1983; Clague and Evans, 1994; Breien et al., 2008; Evans and Delaney, 2015). When a steep slope starts at the end of a flat valley, the flood wave is able to gather debris, transforms into

a debris flow. In the case of the Korumdu lake, the water stream from lake flows on a gentle slope. This is flood wave type. It is important to understand which type of flow comes.

[Comment 17] Fig 5: can you remove the blue tone of the photos by improving the colour balance?

[Response] We already arranged color balance.

[Comment 18] Fig 7: there are no panels c and d as indicated in the caption, and I guess b is placed wrong.

[Response] We changed it.

[Comment 19] Fig 11: indicate that dark blue in the tunnel is frozen

[Response] We changed it.

---

## Referee Report (RR1)

**Review by Mauro Fischer (Glaciologist at the Institute of Geography, University of Bern, Switzerland, mauro.fischer@giub.unibe.ch) of the revised manuscript**

**«Formation, evolution and drainage of short-lived glacial lakes in permafrost environments of the northern Teskey Range, Central Asia"**

**By Mirlan Daiyrov and Chiyuki Narama**

**General Comments**

Dear authors, dear editor, I appreciate the authors' efforts made concerning corrections and improvements of the first submitted version of their manuscript. In general, implementations of reviewers' comments to the first submitted version of the manuscript are good. The paper is now much clearer, easier to read, very nicely illustrated, and of interest to the scientific community.

On behalf of the editor, I reviewed the second submitted version of the manuscript again in detail. I would ask the authors to consider my comments on specific content/scientific and technical issues listed below and implement these comments as much as possible for their next version of the paper. This should be quite easy and won't take so much time again as for the first submitted version of the manuscript.

My main point of criticism to the second version of the paper is that interpretation/discussion/findings of/from the analysis of the other 160 short-lived glacial lakes in the Teskey Range detected using satellite imagery is still rather (if not too) speculative, even if I can totally follow your rationale here and also guess that you're not that wrong... It mainly concerns chapters 4.4, 5.1 (Lns 252-259) and some corresponding remarks in the conclusions. I think it does make sense that you use the well-studied formation and drainage processes of Korumdu lake (and of the four other lakes) as an analogy for other short-lived glacial lakes of the same type… but I argue that you would need to investigate the other 160 short-lived glacial lakes a bit in more detail to say, for instance, if really 73% are of the same type as Korumdu, how they form and drain etc…. It makes sense to me that you take appearance dates to estimate which of the different possible formation processes belong to the lakes detected using satellite imagery. – But, in my opinion, you should be able to "give some more hard facts rather than only educated guesses" here… For instance, you could try to analyse based on the satellite imagery whether the other lakes have a (visible) surface or subsurface (outlet ice-tunnel) drainage channel (at least from the imagery with highest resolution) or not, how the geomorphological setting of the short-lived lake is (i.e. are there ice-cored moraine complexes visible or not), etc. etc…. this would give you valuable additional information to check if you're right with your statement that ¾ of the other short-lived glacial lakes show similar formation processes as for instance Korumdu lake… I hope you see my point here…

In my opinion, the manuscript now is subject to minor revisions, but can be accepted for publication after implementation of these issues by the authors. Many thanks and all the best, kind regards

**Abstract**
Add reason how formation of short-lived glacial lakes due to closure of outlet ice-debris tunnel can be detected from satellite imagery

Ln 23: *"…in rates…"* → "…in discharge rates…"

Lns 23f: *"…outlet ice tunnels."* → "…outlet ice-tunnels."

Lns 24f: *"For the 160 other short-lived glacial lakes, we argue that 117 formed mainly due to tunnel closure from deposition of ice ice-debris mixture, though increased glacial melt also likey contributed."* → delete one "ice" ("…ice ice-debris…"); add "…to lake formation." at the end of the sentence?; moreover, I would appreciate if you could add some words here in this sentence about the reason(s) why you argue that 117 of the detected short-lived glacial lakes formed mainly due to tunnel closure (how did you do that/how could you define that based on the satellite imagery?)

**Introduction**
Short-lived proglacial lakes? Unterscheidung zu anderen Gruppen von "glacial lakes» nicht immer klar, überall überprüfen

Ln 41: *"Most short-lived glacial lake fill …"* → "…glacial lakes fill…"

Ln 44: *"…because they appears suddenly…"* → "…because they appear suddenly…"

Lns 44f: *"Such an outburst (outburst mechanism and damage potential differs from those that are caused by…"* → I would delete "such an outburst" at the beginning of the sentence here, much clearer in my opinion to just write: "Outburst mechanism and damage potential of short-lived glacial lakes in the northern Tien Shan differ from those that are caused by…"

Ln 47: *"… a mass-movement trigger is the main cause of dam failures of the glacial lakes in the Himalayas and Andes…"* → I think the wording "mass-movement trigger" is misleading here, what do you exactly mean? – be precise and rephrase accordingly…

Ln 53: *"Several studies reported the formation and drainage supraglacial lakes are related to…"* → "Several studies reported that formation and drainage of supraglacial lakes are related to…"

Ln 55: *"…of the short-lived glacial lakes…"* → be precise what glacial lakes you are referring to here, I would write: "…of short-lived glacial lakes in northern Tien Shan…"

Ln 56: *"Short-lived glacial lakes appear…"* → again, I would define which glacial lakes you are referring to here (seems important to me to be very clear on that in the introduction already), thus I would write: "Short-lived glacial lakes in northern Tien Shan appear…"

Ln 56: *"…when glacier recedes…"* → "…when a glacier retreats…"?

Lns 58f: *"…and on a surging glacier (Richardson and Reynolds, 2000; Kääb et al., 2004)."* → I would argue that it might be clearer or less misleading here if you'd delete this subordinate clause because as far as I know surging glaciers (or short-lived glacial lakes formed by glacier surges) are not really frequent in the northern Teskey Range, in northern Tien Shan (contrary to for example the Pamir), right? And in this paragraph you really focus on short-lived glacial lakes in northern Tien Shan or on short-lived glacial lakes in permafrost environment, i.e. the type of short-lived glacial lakes your paper is all about…

Ln 60: *"…water supply on an…"* → "water supply to/towards an…"?

Ln 66: *"…but also to above three conditions in the glacier forefield as described above."* → "…but also to the three conditions in the glacier forefield described above." (otherwise there is a repetition in this sentence…)

Lns 67f: *"They can change over large areas and volumes in a short period of time…"* → I think what you mean here is: "They can change in area and volume over a short period of time…", please change accordingly

Ln 74: I would delete the "flood" here, as lake discharge also occurs if there isn't a flood…

Ln 75: *"…store and drain water…"* → clearer to write "…form and drain…"?

Lns 78f: *"Findings from our study are relevant for glacier-related hazard mitigation."* → In my opinion, this sentence is "a bit lost" or "out of context" here… I think your study is first about process understanding of the formation and drainage of short-lived glacial lakes in permafrost environments of the Teskey Range, and in a second step it may serve as valuable information/resource for mitigating hazards related to rapid drainage of such lakes…. But your study is not really dealing with how to mitigate such hazards directly… I would consider if it's not clearer to delete this sentence here and add something on that point to the discussion of the paper…

**Study area**

In this section, but also in the whole manuscript, I would maybe use "m a.s.l" instead of just "m" for information or numbers related to altitude

Lns 86f: One cannot really compare the mean annual air temperature data between the western, central and eastern part of the Teskey Range based on the data you give here (recorded time periods, length of recorded periods, and elevation of the three stations are significantly different), the same actually also applies to some extent for mean annual precipitation, as there is more precipitation for higher elevations and the three stations are not at the same altitude at all (Lns 83ff)... Maybe it would be more clear to compare climatic data over the same time period and for an equal elevation to describe climatic differences between the western, central and eastern part of the Teskey Range...

Lns 87f: *"The western part of the range had less glacier shrinkage than that in the eastern part..."* → do you refer to absolute or relative values here? (as the glaciers in the eastern Teskey Range are generally larger I guess you mean "...showed less relative glacier shrinkage than the eastern part..."? → please check and implement accordingly...

Lns 92f: *"...due to ice and debris stagnated by several glacier advances during the glacier shrinkage process..."* → not very well and not very much understandably written... please rephrase in order to be clear what you exactly mean here

Ln 95: *"...satellite image in 2019..."* → "...satellite image of 2019..."

Lns 97f: *"We selected this lake for research because..."* → "We selected this lake for field surveys because..."

Lns 100f: *"...drainage from Korumdu lake is the flood-wave type in the downstream region because the water stream flows on a gentle slope."* → In my opinion, the reason if drainage of a glacial lake occurs as a flood wave (with a clear "front", i.e. rapidly increasing discharge) or if drainage of a glacial lake takes place by steadily (but more slowly) increasing discharge does not depend on the slope downstream of the lake but on the outburst or discharge processes (sudden outburst vs. steadily increasing channels)! → Please correct and rephrase accordingly...

Ln 102: *"Supplemental Table 1"* → "Supplementary Table 1"

**Methods**

Lns 119f: *"For the water volume at the lake bottom..."* → You should delete "at the lake bottom" here because "at the lake bottom" there is no "water volume", and it becomes clear what you mean from the following subordinate clause...

Ln 121: *"…an in situ water level logger data…"* → delete "an" here (as data is plural); *"in situ"* → "in-situ"?

Ln 123: *"…and combined UAV-derived DSMs."* → clearer: "…and therefore combined UAV-derived DSMs with satellite imagery."!?

Lns 123ff: *"For example, we confirmed the position of the water level by comparing a UAV orthorectify image and satellite data on 1-m counter lines extracted by the combined UAV-derived DSMs."* → I don't really understand this sentence, cannot really follow how you did this… Rephrase in order to be clearer or delete this sentence?

Ln 125: *"…water area…"* → "…lake area…"

Ln 126: *"…between August 4 and 31…"* → of which year(s)? not very clear to me…

Ln 127: *"…after our last visit…"* → "…after our last field survey…"

Ln 131: *"…during in 2015-2019…"* → delete the "in" here

Ln 134: *"Supplemental Table 1"* → "Supplementary Table 1"

Section 3.2: It is not very clear to me from this paragraph if, based on the numerous satellite data (Supplementary Table 1) you mapped/digitized short-lived glacial lakes only once or several times during summer months (i.e. if you were also looking at area changes of detected short-lived glacial lakes during summer months of the individual years or not…) → if yes please add something on that here…

**Results**

Ln 145: I would add "…in recent years." after *"…that formed during the retreat of the glacier…"*; *"The UAV ortho-images in 2019…"* → "The UAV ortho-images of 2019…"

Ln 151: *"…we observed melt draining…"* → "…we observed melt water draining…"

Ln 152: *"…but not in 12 August 2016…"* → "…but not on 12 August 2016…"

Ln 154: *"…, the lake area did not appear."* → "…, the lake did not form."

Ln 155: *"…we had more images of the area…"* → "…more images of the area could be acquired…"

Lns 155f: *"…a more precise sequence of changes in lake size is shown with a sequence of PlanetScope satellite images…"* → repetition "sequence of", I would rephrase as: "…a more detailed evolution of changes in lake size is shown with a sequence of PlanetScope satellite images…"

Lns 160ff: *"In addition, we checked Landsat-8/OLI data in 2014, finding that the lake existed on 5 May, 27 June, and 10 September in 2014. Thus, the satellite data demonstrate that the lake is a short-lived glacial lake."* → ok, all fine, but I have two remarks here: i) I totally agree with you that Korumdu lake is a short-lived glacial lake, no doubt about that, but if you refer to three points in time when the lake is visible from satellite imagery it does, in sensu stricto not imply that the lake is a short-lived one (to "prove" your statement in the second sentence here you'd also have needed satellite imagery on which the lake basin is empty (not filled with melt water)), so my comment here relates to the logic of the sentences…; ii) you also detected the lake on 5 May 2014, rather early to the reported timing of lake appearance in other years… do you have any explanation why the lake already existed on 5 May 2014 (did it possibly not drain in summer 2013 and therefore still existed in spring 2014?)… would be interesting to add something on that here…

Chapter 4.2 is mainly written in present tense; shouldn't it be written in past tense?

Ln 165: *"Consider the trends in Korumdu lake during the three summers of 2017–2019."* → "Changes in water level, area, volume and discharge of Korumdu lake were studied in detail during the three summers of 2017–2019."

Ln 170: *"…the net outflow is relatively smooth."* → do you mean "low" here instead of "smooth"? – I would change that accordingly…

Ln 171: *"The temperature fluctuates more…"* → "The water temperature fluctuates more…"

Ln 173: *"The same figures show the cases for 2018 and 2019."* → I would delete this entire sentence here (it is clear from what you write afterwards and like this you can circumvent the issue of the rather fuzzy formulation of "…show the cases for…")…

Lns 179f: *"…shows a local maximum…"* → why "local"? – the whole chapter 4.2 is about very local processes… I would delete "local" here…

Ln 196: Replace "visit" by "field survey"!?

Ln 202f: *"…are consistent with closure in the outlet ice-tunnel **during being duet to** surface motion and ice-debris deposition."* → there is something wrong here (words in bold) but I don't know during which period you mean, please correct and rephrase in order that it corresponds clearly to what you mean…

Lns 203f: *"During our fieldwork in 2016, we observed the entrance of an ice-tunnel and water flow to its entrance. After two or three hours, the lake level increased (Fig. 7), consistent with the cause being closure of the ice-tunnel."* → In the first sentence, there is a repetition of "entrance". The subordinate clause of the second sentence here is not clear yet, please rephrase and change accordingly in order that one can easily understand what is meant here…

Lns 212f: *"To help determine when other short-lived glacial lakes form, we used satellite images during 2013–2018 to identify and examined 160 such short-lived glacial lakes in the northern Teskey Range."* → "To determine when other short-lived glacial lakes in the northern Teskey Range formed, we used satellite images of 2013–2018 (cf. Supplementary Table 1). Based on the satellite imagery, a total of 160 short-lived glacial lakes could be identified."

Lns 216f: *"Such variability has been argued to be related to geomorphological conditions such as drainage through ice tunnel inside of ice-cored moraine complex (Daiyrov et al., 2018)."* → Ok, but this does not directly or not clear enough relate to the detected variability in lake formation (number of lakes in different years) and appearance date (proportion of lakes forming during snow-melt period vs. during ice-melt period). You have to clarify and rephrase here and write exactly what you mean in order to be clear…

Ln 220: *"…likely has…"* → "…likely have…"

**Discussion**
Ln 227: Why not add years of appearance for the four glacial lakes in addition to the months here? – or did they reappear several times? – not very clear (one would have to go into the paper you refer to here)…

Ln 235: *"…supports…"* → "…support…"

Lns 241ff: This paragraph about the Toguz-Bulak glacial lake is very interesting, but, as you also write, the process of lake drainage and main drivers of lake formation is (very) different compared to the short-lived glacial lakes forming through blockage of a subsurface outlet ice-tunnel… Indeed, water level and lake volume in such cases (lakes with surface drainage channel) is predominantly influenced by the morphology of the lake depression and the elevation of the surface drainage channel, as well as by the glacier mass balance. In my opinion you're talking about an existing and well interpreted but other type of short-lived glacial lakes here… and the last sentence of the paragraph *("Thus, in addition to the closure of deposition, the lake-area changes during summer is also likely influenced by changes in the rate of incoming meltwater.")* is not really appropriate here (as short-lived glacial lakes are rather of the one type (with surface drainage channel) or the other (with subsurface drainage and temporal blockage of the outlet ice-tunnel). Thus, what you write is not wrong but in my

opinion you should more clearly differentiate between these lake types and maybe write: "Thus, as for short-lived glacial lakes with surface drainage channels like Toguz-Bulak, the evolution of the glacier mass balance during summer (amount of snow and ice melt flowing into the lake) also plays an important role for the formation and evolution of short-lived glacial lakes having a subsurface outlet ice-tunnel."

Ln 245: You're talking about Korumdu lake again here → write: "In 2017, there were two trends in water volume of Korumdu lake…"

Ln 251: *"…and deposition condition of tunnel-closure point."* → I can guess what you mean here but I think it would be worth adding some key words in parentheses at the end of this sentence to clarify what you exactly mean here…

Ln 266: *"…emptied at the higher discharge rate…"* → "…emptied at a higher discharge rate…"

Ln 270: *"…which we argued…"* → "…which we argue…"

Ln 273: *"…yet the discharge rates were nearly the same every year…"* → From figure 6 I wouldn't say that they were "nearly the same", maybe write "…yet discharge rates were in the same order of magnitude every year…"

Ln 278: *"If these apply…"* → "If these conditions and evolution also apply…"

**Conclusions**
Ln 283: I would add the years of field surveys here and some specifications about the location of Korumdu lake in parentheses

Ln 284: *"argued"* → "argue"

Ln 287: *"we were able to estimate daily lake discharge and approximate the tunnel dimensions at much less…"* → "…we were able to study the temporal evolution (lake area, volume) and daily lake discharge and approximate the tunnel dimensions to much less…"

Ln 288: Delete *"…were…"*

Ln 289: *"Kromudu"* → "Korumdu"

Ln 293: *"…flood via…"* → «…drain through…"

Ln 294: Delete *"…can be…"* (or replace "are" with "can be" in Ln 293)

**Acknowledgements**

Ln 303: *"Thanks to an editor Margreth Keiler and two revisers (M. Fisher and anonymous)…."*
→ "Thanks to the editor (Margreth Keiler) and two reviewers (Mauro Fischer and anonymous)…"; please spell my name correctly ;-) but many thanks for the acknowledgement, I appreciate that!

**Figures**

Figure 1: In the legend of figure 1 (right of the green square with check) you have to change *"Large drainage of short-live lake"* into "Large drainage of short-lived lake" (a "d" is missing)

Figure 3: figure caption: *"The width at the lake middle is about 85 m (left image) and 40 m (right)."* → "Maximum lake width is about 85 m (left image) and 40 m (right image)."; maybe you could add this directly in the figure (similar as you have done in Figure 12 b for the outlet ice-tunnel height and width)…

Figure 5: figure caption: *"Korumdu lake during 2017–2019 from on time-lapse camera images acquired in the field."* →"Korumdu lake during 2017–2019 from on-site time-lapse camera images acquired in the field."

Figure 6: figure caption: *"Water levels of Korumdu lake in 2017–2019. (a) and air/water temperature in 2017. (b) Lake volume. (c) Lake surface area. (d) Inflow–outflow rate."* → "(a) Water levels and air/water temperature, (b) lake volume, (c) lake area, and (d) inflow-outflow rate of Korumdu lake during summer months of 2017–2019."; moreover, I don't really understand why you write "air/water temperature"… As this is not the same… is this because the temperature logger is first above the water level and then later in summer below (i.e. in the water)?... or are you really just showing water temperature data here (in Fig. 6a (as also explained in chapter 4.2)? should you therefore delete "air/" in the figure caption here? please clarify and add something on that here or in the text…

Figure 7: I would suggest to delete the "small" in the figure caption but add the amount of lake level increase (in centimeters or decimeters) directly in Fig. 7a) and 7c) (with arrows showing the lake level difference and numbers indicating the amount of lake level changes)…

Figure 8: I would delete "…on the listed dates" in the figure caption as this becomes clear right after…; in the figure I would add somewhere (a), (b), (c) or (d) to which date the "basin line" corresponds (and I would change "basin line" into "lake level on XX.XX.20XX")…

Figure 9: In figure 9 a) and b), the red lines refer to feature positions in 2015, and the blue lines to the new position of the same features in 2016. In figure 9 c), blue is taken for the 2015

surface and red for the 2016 surface. – This is misleading. – I would switch the colours in figure 9 c) in order to be consistent with figure 9 a) and b) and increase clarity…

---

## Author Response (AR2)

Dear Editor and Referees,

We would like to thank the editors of NHESS and two Referees for revised manuscript.

We highly appreciate the reviewers for their time spent on reviewing our manuscript and their valuable comments helping us improving the article. Following the two reviewers' suggestions and comments, we carefully revised our manuscript.

Following your instructions and suggestions, this document was prepared and we resubmit it. Below, the authors have tried to answer the questions and reply to the referee comments, point by point. We showed our replay for reviewer comments as red color sentences.

Thank you again for your time and efforts on our manuscript.

Yours sincerely,

Mirlan Daiyrov and Chiyuki Narama

**Response to comments from Referee #1**

*[General comments] The paper has much improved. I recommend acceptance subject to the following minor revisions that should be checked by the editor.*
*The paper will need considerable language editing, I list below only remarks concerning the scientific content.*
**[Response]** We thanks a lot for your careful reading and valuable comments! We appreciate it.

Line 10: Proposed text change: Short-lived glacial lakes in the Teskey Range grow rapidly …
**Response**: We changed it.

L28: Proposed text change: Moreover, increasing temperatures …
**Response**: We changed it.

L47: Proposed text change: A mass-movement trigger is often the main cause of …
**Response**: We changed it.

L81: glacier distribution? Do you mean "size of glaciers", "number of glaciers", "density of glaciers", "glacier area"?
**Response**: This shows elevation range. So, we improved the text.

L92: Proposed text change: …due to ice and debris stagnating during glacier shrinkage after several glacier advances.
**Response**: We changed it.

L100: it is unclear what the "flood-wave type" is and how it relates to gentle slopes?
**Response**: When the channel steeps and the wall-material is erodible, the flood wave can gather debris, transforming into a debris flow. However, the channel slope is not steep in Korumdu, and it means flood-wave without debris. We improved it as "According to data in Narama et al. (2018), outburst drainage from Korumdu lake is the flood-wave type in the downstream region because the water stream flows on a gentle slope which the flow hardly acquire any debris by erosion."

L147: what is "ice ridging"?
**Response**: We changed to ice-exposed ridge.
This is a small ice-exposed ridge at lake water side caused by a melting process on ice-cored moraines.

L165: Proposed text change: In the following we consider the trends …
**Response**: We improved it.

L199: Proposed text change: … exposed an ice ridge of up to 7 m height (Fig. 9).
**Response**: We changed it.

L198, 4.3. It is not clear how this section is on surface elevation changes. Please clarify. Only the first sentences relate to elevation.
**Response**: We changed it as "surface changes on ice-cored moraine complex around Korumdu lake.

L234: The term "deposition-closure type" is not very good as also the "deposition-freezing type" tunnels actually close. So this term does not well discern the two types. What about "deposition-collapse type" or so? "Closure" of ice tunnels is often used to describe the slow closure due to ice overburden.
**Response**: We have changed it.

L274: wouldn't one expect that the flowing water melts the tunnel walls and increases the tunnel capacity? Could also some debris/rock features in the tunnel limit this effect and stabilize drainage at low discharge values?
**Response**: We improved the text as "due to thermal erosion and flowing water,,,"

**Response to comments from Referee #2**

«Formation, evolution and drainage of short-lived glacial lakes in permafrost environments of the northern Teskey Range, Central Asia" By Mirlan Daiyrov and Chiyuki Narama

***[General comments]*** *Dear authors, dear editor, I appreciate the authors' efforts made concerning corrections and improvements of the first submitted version of their manuscript. In general, implementations of reviewers' comments to the first submitted version of the manuscript are good. The paper is now much clearer, easier to read, very nicely illustrated, and of interest to the scientific community. On behalf of the editor, I reviewed the second submitted version of the manuscript again in detail. I would ask the authors to consider my comments on specific content/scientific and technical issues listed below and implement these comments as much as possible for their next version of the paper. This should be quite easy and won't take so much time again as for the first submitted version of the manuscript.*
*My main point of criticism to the second version of the paper is that interpretation/discussion/findings of/from the analysis of the other 160 short-lived glacial lakes in the Teskey Range detected using satellite imagery is still rather (if not too) speculative, even if I can totally follow your rationale here and also guess that you're not that wrong... It mainly concerns chapters 4.4, 5.1 (Lns 252-259) and some corresponding remarks in the conclusions. I think it does make sense that you use the well-studied formation and drainage processes of Korumdu lake (and of the four other lakes) as an analogy for other short-lived glacial lakes of the same type… but I argue that you would need to investigate the other 160 short-lived glacial lakes a bit in more detail to say, for instance, if really 73% are of the same type as Korumdu, how they form and drain etc…. It makes sense to me that you take appearance dates to estimate which of the different possible formation processes belong to the lakes detected using satellite imagery. – But, in my opinion, you should be able to "give some more hard facts rather than only educated guesses" here… For instance, you could try to analyse based on the satellite imagery whether the other lakes have a (visible) surface or subsurface (outlet ice-tunnel) drainage channel (at least from the imagery with highest resolution) or not, how the geomorphological setting of the short-lived lake is (i.e. are there ice-cored moraine complexes visible or not), etc. etc…. this would give you valuable additional information to check if you're right with your statement that ¾ of the other short-lived glacial lakes show similar formation processes as for instance Korumdu lake… I hope you see my point here…*
*In my opinion, the manuscript now is subject to minor revisions, but can be accepted for publication after implementation of these issues by the authors. Many thanks and all the best, kind regards*
**[Response]** We thanks a lot for your careful reading and valuable comments! We appreciate it.

**Abstract**
Add reason how formation of short-lived glacial lakes due to closure of outlet ice-debris tunnel can be detected from satellite imagery
**Response**: We improved the sentence as "For the 160 other short-lived glacial lakes, we found that 117 formed during the ice-melt period from July to September. This timing and our findings for Korumdu lake show that these 117 lakes likely formed primarily because deposition of an ice-debris mixture blocked the outlet tunnel, though increased glacial melt would also have contributed."

Ln 23: "…in rates…" "…in discharge rates…"
**Response**: We changed it.

Lns 23f: "…outlet ice tunnels." "…outlet ice-tunnels."
**Response**: We changed it.

Lns 24f: "For the 160 other short-lived glacial lakes, we argue that 117 formed mainly due to tunnel closure from deposition of ice ice-debris mixture, though increased glacial melt also likey contributed." delete one "ice" ("…ice ice-debris…"); add "…to lake formation." at the end of the sentence?; moreover, I would appreciate if you could add some words here in this sentence about the reason(s) why you argue that 117 of the detected short-lived glacial lakes formed mainly due to tunnel closure (how did you do that/how could you define that based on the satellite imagery?)
**Response**: We detected short-live lake due to tunnel closure from deposition of ice-debris mixture based on the timing of lake appearance. 27% lakes appeared in June and 73% lakes appeared in July–September. 73 % lakes are the similar timing of Korumdu lake appearance. In addition, many lakes do not have surface channels. So, we wrote "primarily" in the sentence.
We improved it as "For the 160 other short-lived glacial lakes, we found that 117 formed during the ice-melt period from July to September. This timing and our findings for Korumdu lake show that these 117 lakes likely formed primarily because deposition of an ice-debris mixture blocked the outlet tunnel, though increased glacial melt would also have contributed."

**Introduction**
Short-lived proglacial lakes? Unterscheidung zu anderen Gruppen von "glacial lakes» nicht immer klar, überall überprüfen
**Response**: Short-lived lakes are correct.

Ln 41: "Most short-lived glacial lake fill …" "…glacial lakes fill…"
**Response**: We changed it.

Ln 44: "…because they appears suddenly…" "…because they appear suddenly…"
**Response**: We changed it.

Lns 44f: "Such an outburst (outburst mechanism and damage potential differs from those that are caused by…" I would delete "such an outburst" at the beginning of the sentence here, much clearer in my opinion to just write: "Outburst mechanism and damage potential of short-lived glacial lakes in the northern Tien Shan differ from those that are caused by…"
**Response**: We changed it.

Ln 47: "… a mass-movement trigger is the main cause of dam failures of the glacial lakes in the Himalayas and Andes…" I think the wording "mass-movement trigger" is misleading here, what do you exactly mean? – be precise and rephrase accordingly…
**Response**: We improved as "A mass-movement such as an ice avalanche or landslide is often the main cause of dam failures of the glacial lakes in the Himalayas and Andes (Emmer and Cochachin, 2013; Neupane et al.; 2019)."

Ln 53: "Several studies reported the formation and drainage supraglacial lakes are related to…" "Several studies reported that formation and drainage of supraglacial lakes are related to…"
**Response**: We changed it.

Ln 55: "…of the short-lived glacial lakes…" be precise what glacial lakes you are referring to here, I would write: "…of short-lived glacial lakes in northern Tien Shan…"
**Response**: We changed it.

Ln 56: "Short-lived glacial lakes appear…" again, I would define which glacial lakes you are referring to here (seems important to me to be very clear on that in the introduction already), thus I would write: "Short-lived glacial lakes in northern Tien Shan appear…"

**Response**: We changed it.

Ln 56: "…when glacier recedes…" "…when a glacier retreats…"?
**Response**: We changed it.

Lns 58f: "…and on a surging glacier (Richardson and Reynolds, 2000; Kääb et al., 2004)." I would argue that it might be clearer or less misleading here if you'd delete this subordinate clause because as far as I know surging glaciers (or short-lived glacial lakes formed by glacier surges) are not really frequent in the northern Teskey Range, in northern Tien Shan (contrary to for example the Pamir), right? And in this paragraph you really focus on short-lived glacial lakes in northern Tien Shan or on short-lived glacial lakes in permafrost environment, i.e. the type of short-lived glacial lakes your paper is all about…
**Response:** We deleted last sentence of "and on a surging glacier (Richardson and Reynolds, 2004)."

Ln 60: "…water supply on an…" "water supply to/towards an…"?
**Response**: We changed it.

Ln 66: "…but also to above three conditions in the glacier forefield as described above." "···but also to the three conditions in the glacier forefield described above." (otherwise there is a repetition in this sentence…)
**Response**: We changed it.

Lns 67f: "They can change over large areas and volumes in a short period of time…" I think what you mean here is: "They can change in area and volume over a short period of time…", please change accordingly
**Response**: We changed it.

Ln 74: I would delete the "flood" here, as lake discharge also occurs if there isn't a flood…
**Response**: We changed it.

Ln 75: "…store and drain water…" clearer to write "…form and drain…"?
**Response**: We changed it.

Lns 78f: "Findings from our study are relevant for glacier-related hazard mitigation." In my opinion, this sentence is "a bit lost" or "out of context" here… I think your study is first about process understanding of the formation and drainage of short-lived glacial lakes in permafrost environments of the Teskey Range, and in a second step it may serve as valuable information/resource for mitigating hazards related to rapid drainage of such lakes…. But your study is not really dealing with how to mitigate such hazards directly… I would consider if it's not clearer to delete this sentence here and add something on that point to the discussion of the paper…
**Response**: We deleted it.

**Study area**
In this section, but also in the whole manuscript, I would maybe use "m a.s.l" instead of just "m" for information or numbers related to altitude
**Response**: we changed all in the manuscript.

Lns 86f: One cannot really compare the mean annual air temperature data between the western, central and eastern part of the Teskey Range based on the data you give here (recorded time periods, length of recorded periods, and elevation of the three stations are significantly different), the same actually also applies to some extent for mean annual precipitation, as there is more precipitation for higher elevations and the three stations are not at the same altitude at all (Lns 83ff)… Maybe it would be more clear to compare climatic data over the same time period and for an equal elevation to describe climatic differences between the western, central and eastern part of the Teskey Range…
**Response**: Unfortunately, there are no meteorological observation records for the same period. Kara-Kujur station is Soviet times.

Lns 87f: "The western part of the range had less glacier shrinkage than that in the eastern part…" do you refer

to absolute or relative values here? (as the glaciers in the eastern Teskey Range are generally larger I guess you mean "…showed less relative glacier shrinkage than the eastern part…"? please check and implement accordingly…

**Response**: We changed it.

Lns 92f: "…due to ice and debris stagnated by several glacier advances during the glacier shrinkage process…" not very well and not very much understandably written… please rephrase in order to be clear what you exactly mean here

**Response**: We changed it as "due to ice and debris stagnating during glacier shrinkage after several glacier advances (Iwata et al., 2005)."

Ln 95: "…satellite image in 2019…" "…satellite image of 2019…"

**Response**: We changed it.

Lns 97f: "We selected this lake for research because…" "We selected this lake for field surveys because…"

**Response**: We changed it.

Lns 100f: "…drainage from Korumdu lake is the flood-wave type in the downstream region because the water stream flows on a gentle slope." In my opinion, the reason if drainage of a glacial lake occurs as a flood wave (with a clear "front", i.e. rapidly increasing discharge) or if drainage of a glacial lake takes place by steadily (but more slowly) increasing discharge does not depend on the slope downstream of the lake but on the outburst or discharge processes (sudden outburst vs. steadily increasing channels)! Please correct and rephrase accordingly…

**Response**: We improved the sentence as "According to data in Narama et al. (2018), outburst drainage from Korumdu lake is the flood-wave type in the downstream region because the water stream flows on a gentle slope which the flow hardly acquire any debris by erosion."

Ln 102: "Supplemental Table 1" "Supplementary Table 1"

**Response**: We changed it.

**Methods**

Lns 119f: "For the water volume at the lake bottom…" You should delete "at the lake bottom" here because "at the lake bottom" there is no "water volume", and it becomes clear what you mean from the following subordinate clause…

**Response**: We changed to "at the lake bottom layer"

Ln 121: "…an in situ water level logger data…" delete "an" here (as data is plural); "in situ" "in-situ"?

**Response**: We changed it.

Ln 123: "…and combined UAV-derived DSMs." clearer: "…and therefore combined UAVderived DSMs with satellite imagery."!?

**Response**: We changed it.

Lns 123ff: "For example, we confirmed the position of the water level by comparing a UAV orthorectify image and satellite data on 1-m counter lines extracted by the combined UAV-derived DSMs." I don't really understand this sentence, cannot really follow how you did this… Rephrase in order to be clearer or delete this sentence?

**Response**: We improved the sentence as "For example, we confirmed the position of the water level by comparing a UAV orthorectify image or satellite data with 1-m contour lines from the combined UAV-derived DSMs."

Ln 125: "…water area…" "…lake area…"

**Response**: We changed it.

Ln 126: "…between August 4 and 31…" of which year(s)? not very clear to me…

**Response**: We changed to August 4 and 31 2019.

Ln 127: "…after our last visit…" "…after our last field survey…"
**Response**: We changed it.

Ln 131: "…during in 2015-2019…" delete the "in" here
**Response**: We deleted it.

Ln 134: "Supplemental Table 1" "Supplementary Table 1"
**Response**: We changed it.

Section 3.2: It is not very clear to me from this paragraph if, based on the numerous satellite data (Supplementary Table 1) you mapped/digitized short-lived glacial lakes only once or several times during summer months (i.e. if you were also looking at area changes of detected short-lived glacial lakes during summer months of the individual years or not…) if yes please add something on that here…
**Response**: We added the sentence as "We also investigated the area changes of short-lived glacial lakes during summer months in a given year."

**Results**
Ln 145: I would add "…in recent years." after "…that formed during the retreat of the glacier…"; "The UAV ortho-images in 2019…" "The UAV ortho-images of 2019…"
**Response**: We changed it.

Ln 151: "…we observed melt draining…" "…we observed melt water draining…"
**Response**: We changed it.

Ln 152: "…but not in 12 August 2016…" "…but not on 12 August 2016…"
**Response**: We changed it.

Ln 154: "…, the lake area did not appear." "…, the lake did not form."
**Response**: We changed it.

Ln 155: "…we had more images of the area…" "…more images of the area could be acquired…"
**Response**: We changed it.

Lns 155f: "…a more precise sequence of changes in lake size is shown with a sequence of PlanetScope satellite images…" repetition "sequence of", I would rephrase as: "…a more detailed evolution of changes in lake size is shown with a sequence of PlanetScope satellite images…"
**Response**: We changed it.

Lns 160ff: "In addition, we checked Landsat-8/OLI data in 2014, finding that the lake existed on 5 May, 27 June, and 10 September in 2014. Thus, the satellite data demonstrate that the lake is a short-lived glacial lake." ok, all fine, but I have two remarks here: i) I totally agree with you that Korumdu lake is a short-lived glacial lake, no doubt about that, but if you refer to three points in time when the lake is visible from satellite imagery it does, in sense strict not imply that the lake is a short-lived one (to "prove" your statement in the second sentence here you'd also have needed satellite imagery on which the lake basin is empty (not filled with melt water)), so my comment here relates to the logic of the sentences…; ii) you also detected the lake on 5 May 2014, rather early to the reported timing of lake appearance in other years… do you have any explanation why the lake already existed on 5 May 2014 (did it possibly not drain in summer 2013 and therefore still existed in spring 2014?)… would be interesting to add something on that here…
**Response**: We deleted this sentence.

**Chapter 4.2** is mainly written in present tense; shouldn't it be written in past tense?
**Response**: We imported sentences in this chapter.

Ln 165: "Consider the trends in Korumdu lake during the three summers of 2017–2019."

"Changes in water level, area, volume and discharge of Korumdu lake were studied in detail during the three summers of 2017–2019."
**Response**: We changed it.

Ln 170: "…the net outflow is relatively smooth." do you mean "low" here instead of "smooth"? – I would change that accordingly…
**Response**: We changed it.

Ln 171: "The temperature fluctuates more…" "The water temperature fluctuates more…"
**Response**: We changed it.

Ln 173: "The same figures show the cases for 2018 and 2019." I would delete this entire sentence here (it is clear from what you write afterwards and like this you can circumvent the issue of the rather fuzzy formulation of "…show the cases for…")…
**Response**: We deleted it.

Lns 179f: "…shows a local maximum…" why "local"? – the whole chapter 4.2 is about very local processes… I would delete "local" here…
**Response**: We deleted "local".

Ln 196: Replace "visit" by "field survey"!?
**Response**: We replaced "visit" into "field survey".

Ln 202f: "…are consistent with closure in the outlet ice-tunnel **during being due to** surface motion and ice-debris deposition." there is something wrong here (words in bold) but I don't know during which period you mean, please correct and rephrase in order that it corresponds clearly to what you mean…
**Response**: We changed to during ice-melt period due to surface motion and ice-debris deposition.

Lns 203f: "During our fieldwork in 2016, we observed the entrance of an ice-tunnel and water flow to its entrance. After two or three hours, the lake level increased (Fig. 7), consistent with the cause being closure of the ice-tunnel." In the first sentence, there is a repetition of "entrance". The subordinate clause of the second sentence here is not clear yet, please rephrase and change accordingly in order that one can easily understand what is meant here…
**Response**: We improved the first sentence as "During our fieldwork in 2016, we observed water flow at the entrance of an ice-tunnel."

Lns 212f: "To help determine when other short-lived glacial lakes form, we used satellite images during 2013–2018 to identify and examined 160 such short-lived glacial lakes in the northern Teskey Range." "To determine when other short-lived glacial lakes in the northern Teskey Range formed, we used satellite images of 2013–2018 (cf. Supplementary Table 1). Based on the satellite imagery, a total of 160 short-lived glacial lakes could be identified."
**Response**: We changed it.

Lns 216f: "Such variability has been argued to be related to geomorphological conditions such as drainage through ice tunnel inside of ice-cored moraine complex (Daiyrov et al., 2018)." Ok, but this does not directly or not clear enough relate to the detected variability in lake formation (number of lakes in different years) and appearance date (proportion of lakes forming during snow-melt period vs. during ice-melt period). You have to clarify and rephrase here and write exactly what you mean in order to be clear…
**Response**: We added the sentence as "The number of lakes vary greatly by year and by appearance date, indicating that the formation of these short-lived glacial lakes can not be explained solely by an increase of melt water during summer."

Ln 220: "…likely has…" "…likely have…"
**Response**: We changed it.

**Discussion**

Ln 227: Why not add years of appearance for the four glacial lakes in addition to the months here? – or did they reappear several times? – not very clear (one would have to go into the paper you refer to here)…
**Response**: We added the years.

Ln 235: "…supports…" "…support…"
**Response**: We changed it.

Lns 241ff: This paragraph about the Toguz-Bulak glacial lake is very interesting, but, as you also write, the process of lake drainage and main drivers of lake formation is (very) different compared to the short-lived glacial lakes forming through blockage of a subsurface outlet icetunnel… Indeed, water level and lake volume in such cases (lakes with surface drainage channel) is predominantly influenced by the morphology of the lake depression and the elevation of the surface drainage channel, as well as by the glacier mass balance. In my opinion you're talking about an existing and well interpreted but other type of short-lived glacial lakes here… and the last sentence of the paragraph ("Thus, in addition to the closure of deposition, the lake-area changes during summer is also likely influenced by changes in the rate of incoming meltwater.") is not really appropriate here (as short-lived glacial lakes are rather of the one type (with surface drainage channel) or the other (with subsurface drainage and temporal blockage of the outlet ice-tunnel). Thus, what you write is not wrong but in my opinion you should more clearly differentiate between these lake types and maybe write:

"Thus, as for short-lived glacial lakes with surface drainage channels like Toguz-Bulak, the evolution of the glacier mass balance during summer (amount of snow and ice melt flowing into the lake) also plays an important role for the formation and evolution of short-lived glacial lakes having a subsurface outlet ice-tunnel."
**Response**: We improved the sentence.

Ln 245: You're talking about Korumdu lake again here write: "In 2017, there were two trends in water volume of Korumdu lake…"
**Response**: We changed it.

Ln 251: "…and deposition condition of tunnel-closure point." I can guess what you mean here but I think it would be worth adding some key words in parentheses at the end of this sentence to clarify what you exactly mean here…
**Response**: We added some words as "and deposition condition of tunnel-closure point (e.g., when melting can open the blocked region)."

Ln 266: "…emptied at the higher discharge rate…" "…emptied at a higher discharge rate…"
**Response**: We changed it.

Ln 270: "…which we argued…" "…which we argue…"
**Response**: We changed it.

Ln 273: "…yet the discharge rates were nearly the same every year…" From figure 6 I wouldn't say that they were "nearly the same", maybe write "…yet discharge rates were in the same order of magnitude every year…"
**Response**: We changed it.

Ln 278: "If these apply…" "If these conditions and evolution also apply…"
**Response**: We changed it.

**Conclusions**

Ln 283: I would add the years of field surveys here and some specifications about the location
of Korumdu lake in parentheses
**Response**: We added years of field surveys and the location.

Ln 284: "argued" "argue"
**Response**: We changed it.

Ln 287: "we were able to estimate daily lake discharge and approximate the tunnel dimensions at much less…"
"…we were able to study the temporal evolution (lake area, volume) and daily lake discharge and approximate the tunnel dimensions to much less…"
**Response**: We changed it.

Ln 288: Delete "…were…"
**Response**: We deleted it.

Ln 289: "Kromudu" "Korumdu"
**Response**: We corrected it.

Ln 293: "…flood via…" «…drain through…"
**Response**: We changed it.

Ln 294: Delete "…can be…" (or replace "are" with "can be" in Ln 293)
**Response**: We replaced "are" with "can be" in Ln 293 and deleted "can be".

**Acknowledgements**

Ln 303: "Thanks to an editor Margreth Keiler and two revisers (M. Fisher and anonymous)…." "Thanks to the editor (Margreth Keiler) and two reviewers (Mauro Fischer and anonymous)…"; please spell my name correctly ;-) but many thanks for the acknowledgement, I appreciate that!
**Response**: Thank you for your comments. We changed it.

**Figures**

Figure 1: In the legend of figure 1 (right of the green square with check) you have to change "Large drainage of short-live lake" into "Large drainage of short-lived lake" (a "d" is missing)
**Response**: We changed it.

Figure 3: figure caption: "The width at the lake middle is about 85 m (left image) and 40 m (right)." "Maximum lake width is about 85 m (left image) and 40 m (right image)."; maybe you could add this directly in the figure (similar as you have done in Figure 12 b for the outlet ice-tunnel height and width)…
**Response**: We changed it.

Figure 5: figure caption: "Korumdu lake during 2017–2019 from on time-lapse camera images acquired in the field." "Korumdu lake during 2017–2019 from on-site time-lapse camera images acquired in the field."
**Response**: We changed it.

Figure 6: figure caption: "Water levels of Korumdu lake in 2017–2019. (a) and air/water temperature in 2017. (b) Lake volume. (c) Lake surface area. (d) Inflow–outflow rate." "(a) Water levels and water temperature, (b) lake volume, (c) lake area, and (d) inflow-outflow rate of Korumdu lake during summer months of 2017–2019.";
moreover, I don't really understand why you write "air/water temperature"… As this is not the same… is this because the temperature logger is first above the water level and then later in summer below (i.e. in the water)?... or are you really just showing water temperature data here (in Fig. 6a (as also explained in chapter 4.2)? should you therefore delete "air/" in the figure caption here? Please clarify and add something on that here or in the text…
**Response**: We deleted "air/" from text.

Figure 7: I would suggest to delete the "small" in the figure caption but add the amount of lake level increase (in centimeters or decimeters) directly in Fig. 7a) and 7c) (with arrows showing the lake level difference and numbers indicating the amount of lake level changes)…
**Response**: We deleted "small" and added the amount of lake level increase in Fig. 7a and 7c.

Figure 8: I would delete "…on the listed dates" in the figure caption as this becomes clear right after…; in the

figure I would add somewhere (a), (b), (c) or (d) to which date the "basin line" corresponds (and I would change "basin line" into "lake level on XX.XX.20XX")…

**Response**: We deleted "…on the listed dates". Red line shows outline of lake basin, so we remain the name as outline of basin in the figure.

Figure 9: In figure 9 a) and b), the red lines refer to feature positions in 2015, and the blue lines to the new position of the same features in 2016. In figure 9 c), blue is taken for the 2015 surface and red for the 2016 surface. – This is misleading. – I would switch the colours in figure 9 c) in order to be consistent with figure 9 a) and b) and increase clarity…

**Response**: We changed colors in Fig. 9 c (red color for 2015 and blue color for 2016) as suggested.

---

## Author Response (AR3)

Dear NHESS

We would like to thank the editor and two Referees for revised manuscript.

We are glad to accept our manuscript.

We send our materials of manuscript.

If you need more high resolution data, please inform us it.

Thank you again for your time and efforts on our manuscript.

Yours sincerely,

Chiyuki Narama